# Is Artificial Intelligence Generated Image Detection a Solved Problem?

**Ziqiang Li**[1]    **Jiazhen Yan**[1]    **Ziwen He**[1]    **Kai Zeng**[2]
**Weiwei Jiang**[1]    **Lizhi Xiong**[1]    **Zhangjie Fu**[1][*]
[1]School of Computer Science, Nanjing University of Information Science and Technology
[2]University of Siena, Siena, Italy
iceli@mail.ustc.edu.cn, 247918horizon@gmail.com, kai.zeng@unisi.it
{ziwen.he,weiwei.jiang,fzj}@nuist.edu.cn, lzxiong16@163.com

## Abstract

The rapid advancement of generative models, such as GANs and Diffusion models, has enabled the creation of highly realistic synthetic images, raising serious concerns about misinformation, deepfakes, and copyright infringement. Although numerous Artificial Intelligence Generated Image (AIGI) detectors have been proposed, often reporting high accuracy, their effectiveness in real-world scenarios remains questionable. To bridge this gap, we introduce AIGIBench, a comprehensive benchmark designed to rigorously evaluate the robustness and generalization capabilities of state-of-the-art AIGI detectors. AIGIBench simulates real-world challenges through four core tasks: multi-source generalization, robustness to image degradation, sensitivity to data augmentation, and impact of test-time preprocessing. It includes 23 diverse fake image subsets that span both advanced and widely adopted image generation techniques, along with real-world samples collected from social media and AI art platforms. Extensive experiments on 11 advanced detectors demonstrate that, despite their high reported accuracy in controlled settings, these detectors suffer significant performance drops on real-world data, limited benefits from common augmentations, and nuanced effects of preprocessing, highlighting the need for more robust detection strategies. By providing a unified and realistic evaluation framework, AIGIBench offers valuable insights to guide future research toward dependable and generalizable AIGI detection[2].

## 1    Introduction

Recent advancements in generative models, such as GANs [1–6] and diffusion models [7–10], have demonstrated remarkable capabilities in synthesizing high-quality images that closely resemble real-world scenes. While these technologies have enabled various applications, including personalized portraits [11], virtual try-on, and content creation [12, 13], they also raise significant concerns regarding the authenticity of visual content and its potential misuse in misinformation dissemination [14, 15], deepfake generation [16], and copyright infringement [17, 8].

To address these concerns, Artificial Intelligence Generated Image (AIGI) detection technologies have been developed to differentiate synthetic images from real ones. Most existing approaches [18–28] rely on training binary classifiers to distinguish between real and generated content, with some studies reporting near-perfect performance. In light of reported detection accuracies exceeding 95%, a critical question emerges: *Is Artificial Intelligence Generated Image detection a solved problem?*

---

[*]Corresponding author
[2]Data and code are publicly available at: https://github.com/HorizonTEL/AIGIBench

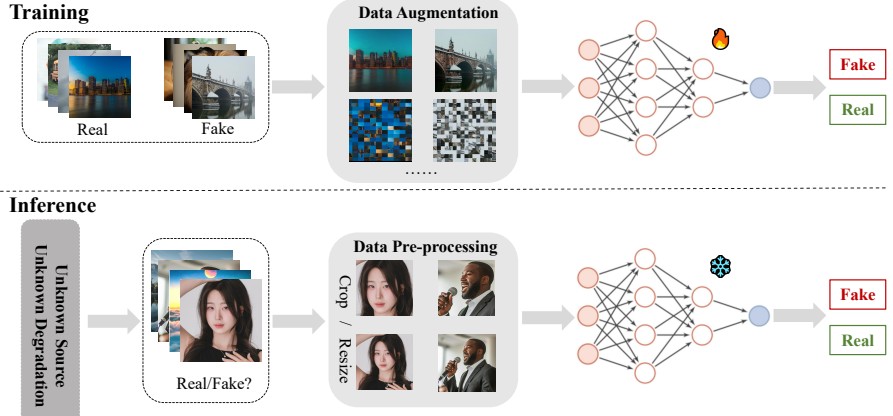

Figure 1: The AIGI Detection Pipeline. In the training phase, both real and AI-generated images are augmented to improve the model's robustness against diverse and previously unseen test distributions. The model is trained to distinguish real images from synthetic ones generated by a variety of unknown sources. During inference, test images potentially affected by unknown degradations or generation pipelines are pre-processed using cropping or resizing to align with the training conditions. The pre-processed images are then evaluated by the trained detector to assess their authenticity.

To answer this question, we propose a novel and comprehensive benchmark, termed AIGIBench, for AIGI detection. As illustrated in Figure 1, we present an end-to-end pipeline designed to assess AIGI detection in real-world scenarios. During the training phase, a diverse collection of real and synthetic images is assembled, and various data augmentation techniques are applied to improve detector robustness. In the testing phase, the detector is evaluated on its ability to identify the authenticity of images originating from unknown sources and subjected to unknown degradations. Prior to detection, each image undergoes pre-processing, such as cropping or resizing to ensure compatibility with the trained detector. The detector then outputs a binary classification indicating whether the image is real or AI-generated. Accordingly, to systematically evaluate the real-world performance of state-of-the-art AIGI detectors, AIGIBench defines four core tasks that mirror practical challenges often overlooked in idealized test environments: **i) Generalization Assessment: Multi-source**, **ii) Robustness Assessment: Multi-degradation**, **iii) Data Augmentation Variation Assessment: Identifying the Most Effective Augmentation**, and **iv) Test Data Pre-processing Assessment: Identifying the Most Effective Pre-processing**. These task variations reflect common challenges in practical applications but are often overlooked in idealized test environments where most existing detectors are developed and evaluated.

The construction of AIGIBench effectively achieves our goal of providing a rigorous and realistic evaluation framework for AIGI detection. Experimental results demonstrate that existing detection methods encounter significant challenges when evaluated on AIGIBench. Key insights are summarized as follows: **i)** Despite recent progress, all detectors suffer notable performance degradation on real-world manipulations such as DeepFakes and in-the-wild content. Furthermore, no single method consistently outperforms others across all generative scenarios, underscoring the difficulty of developing generalizable detectors. **ii)** While most detectors maintain high R.Acc. under perturbations, their F.Acc. drops sharply, indicating reduced detection reliability in practical settings. **iii)** Common data augmentation strategies provide limited benefits in improving detector performance and may even introduce performance trade-offs. **iv)** Although prior study suggests that applying a crop operation during test phase enhances the ability to capture fine-grained artifacts, thereby improving overall detection accuracy, our analysis indicates that these improvements are primarily driven by gains in R.Acc., while F.Acc. often remains unaffected or even degrades.

**The Differences Between Our Benchmark and Others.** Several deepfake and AIGI detection benchmarks have been introduced in recent years, including GenImage (NeurIPS 2023) [29], AIGCDetection (arXiv 2023) [30], DeepfakeBench (NeurIPS 2023) [31], MPBench (NeurIPS 2023) [32], Diff-Forensics (ICCV 2023) [33], WildRF (arXiv 2024) [18], DF40 (NeurIPS 2024) [34], WildFake (AAAI 2025) [35], and Chameleon (ICLR 2025) [20]. As illustrated in Table 1 and Table 2, all of these benchmarks exhibit key differences compared to our proposed AIGIBench: *i) AIGIBench com-*

Table 1: Comparison with existing benchmarks on dataset.

| Benchmark ↓ Dataset → | Generative Methods | ~2022 | 2023 | 2024~ | General Content | GAN & Diffusion | Image-based & Noise-based | Social Networks | AI-painting Communities |
|---|---|---|---|---|---|---|---|---|---|
| GenImage [29] | 8 | 8 | 0 | 0 | ✓ | ✓ | ✗ | ✗ | ✗ |
| AIGCDetction [30] | 17 | 13 | 4 | 0 | ✓ | ✓ | ✗ | ✗ | ✗ |
| DeepfakeBench [31] | 9 | 9 | 0 | 0 | ✗ | ✗ | ✗ | ✗ | ✗ |
| MPBench [32] | 11 | 5 | 6 | 0 | ✓ | ✓ | ✗ | ✗ | ✗ |
| Diff-Forensics [33] | 7 | 7 | 0 | 0 | ✓ | ✗ | ✗ | ✗ | ✗ |
| WildRF [18] | - | - | - | - | ✓ | ✓ | ✗ | ✓ | ✗ |
| DF40 [34] | 40 | 27 | 10 | 3 | ✗ | ✓ | ✗ | ✗ | ✗ |
| WildFake [35] | 22 | 17 | 5 | 0 | ✓ | ✓ | ✓ | ✗ | ✗ |
| Chameleon [20] | - | - | - | - | ✓ | ✓ | ✗ | ✗ | ✓ |
| AIGIBench (Ours) | 25 | 9 | 5 | 11 | ✓ | ✓ | ✓ | ✓ | ✓ |

Table 2: Comparison with existing benchmarks on evaluation.

| Benchmark ↓ Evaluation → | Detection Methods | ~2022 | 2023 | 2024~ | Generalization | Robust | Data Augmentation | Test Data Processing |
|---|---|---|---|---|---|---|---|---|
| GenImage (NeurIPS 2023) [29] | 7 | 7 | 0 | 0 | ✓ | ✓ | ✗ | ✗ |
| AIGCDetction (arXiv 2023) [30] | 10 | 5 | 5 | 0 | ✓ | ✓ | ✗ | ✗ |
| DeepfakeBench (NeurIPS 2023) [31] | 34 | 30 | 3 | 1 | ✓ | ✓ | ✗ | ✗ |
| MPBench (NeurIPS 2023) [32] | 3 | 3 | 0 | 0 | ✓ | ✗ | ✗ | ✗ |
| Diff-Forensics (ICCV 2023) [33] | 6 | 5 | 1 | 0 | ✓ | ✗ | ✗ | ✗ |
| WildRF (arXiv 2024) [18] | 5 | 3 | 1 | 1 | ✓ | ✓ | ✗ | ✗ |
| DF40 (NeurIPS 2024) [34] | 7 | 7 | 0 | 0 | ✓ | ✗ | ✗ | ✗ |
| WildFake (AAAI 2025) [35] | 6 | 2 | 4 | 0 | ✓ | ✓ | ✗ | ✗ |
| Chameleon (ICLR 2025) [20] | 10 | 4 | 4 | 2 | ✓ | ✓ | ✗ | ✗ |
| AIGIBench (Ours) | 11 | 3 | 2 | 6 | ✓ | ✓ | ✓ | ✓ |

*prehensively simulates state-of-the-art image generation methods.* The AIGIBench test set consists of 23 subsets covering both advanced and widely adopted image generation techniques, including: (a) GAN-based noise-to-image generation (ProGAN [36], StyleGAN3 [37], StyleGAN-XL [38], StyleSwim [39], R3GAN [40], and WFIR [41]), (b) Diffusion for text-to-image generation (SD-XL [42], SD-3 [43], DALLE-3 [44], Midjourney-v6 [45], FLUX.1-dev [46], Imagen-3 [47], and GLIDE [48]), (c) GANs for deepfake (BlendFace [49], E4S [50], FaceSwap [51], InSwap [52], and SimSwap [53]), and (d) Diffusion for personalized generation (InstantID [54], Infinite-ID [8], PhotoMaker [16], BLIP-Diffusion [55], and IP-Adapter [56]). It also includes 2 general subsets featuring fake images collected from social media platforms such as X (Twitter), Facebook, and Reddit, as well as AI-painting communities like ArtStation, Civitai, and Liblib. *ii) AIGIBench evaluates nearly all state-of-the-art AIGI detection methods currently available.* In contrast to previous benchmarks that mainly focused on detection methods developed before 2022, AIGIBench incorporates 11 recent detection techniques, over half of which were published after 2024, reflecting the latest advancements in the field. Additionally, all methods are evaluated under a consistent and equitable experimental framework. *iii) AIGIBench is the first benchmark to conduct a comprehensive evaluation of four critical components in the AIGI detection pipeline.* Following the typical workflow of AIGI detection, we systematically assess state-of-the-art detection methods across four fundamental tasks: generalization to unseen sources, robustness to image degradations, sensitivity to data augmentation strategies, and impact of test-time pre- processing. This holistic evaluation framework provides new insights into the practical reliability and limitations of existing AIGI detectors.

## 2 AIGIBench

We enhance the existing dataset construction methodology to more closely reflect real-world detection scenarios. In this section, we present AIGIBench, a newly proposed benchmark designed to support comprehensive evaluation under such practical conditions.

### 2.1 Training Setting

**Training Dataset.** Existing studies on detecting AI-generated images [57, 23, 18, 58] primarily focus on a single-model training setting, where detectors are trained on images generated by a specific model—such as ProGAN [59] or Stable Diffusion [60], and evaluated on samples from various generative models. In contrast, our benchmark introduces two training dataset settings: **i) Setting-I:**

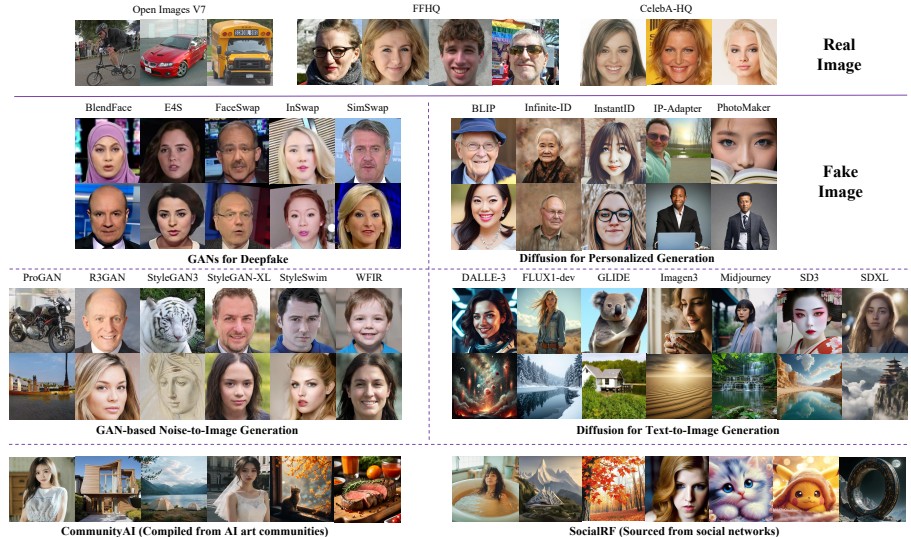

Figure 2: Visualizations of real and fake images from the evaluation datasets used in our AIGIBench.

Training on 72K images generated by ProGAN across four object categories—car, cat, chair, and horse. **ii) Setting-II:** Training on 144K images generated by both SD-v1.4 and ProGAN, covering the same four object categories.

**Implementation Details.** To ensure a fair comparison, we adopt the original hyperparameter settings provided by each method without any modifications. All methods are re-trained under the two training settings described above and evaluated on our proposed custom dataset.

## 2.2 Evaluation Datasets

To support various image generation methods, AIGIBench is structured into multiple subsets, as illustrated in Figure 2, with each subset containing an identical collection of real and fake images. The details of generation methods used in evaluation datasets can be found in Sec. A.6 of the Appendix.

**Fake Image Collection.** To ensure the quality and fairness of our dataset, we collect substantially more images than necessary through both generator and web API, and applied the following processing pipeline: i) To mitigate the presence of visually similar images, we use CLIP [61] image embeddings with a cosine similarity threshold of 0.98 to identify and remove near-duplicate instances, promoting diversity and fairness in detection. More details are shown in Sec. A.5 of the Apendix; ii) To enhance image quality and better reflect real-world social environments, we employ the CLIP-based aesthetic score predictor [62] to evaluate image aesthetics and discarded images with low scores; iii) Finally, manual screening is conducted to remove obviously fake images, further increasing the dataset's realism and challenge. We collect diverse fake images from various sources and organize them into five levels based on the generation pipelines of different models.

*i) GAN-based Noise-to-Image Generation:* To emphasize high-quality synthesis, we select six advanced GANs: ProGAN [36], StyleGAN3 [37], StyleGAN-XL [38], StyleSwim [39], R3GAN [40], and WFIR [41]. We randomly sample latent codes and generate diverse, high-quality images.

*ii) Diffusion for Text-to-Image Generation:* Given the rapid advancements in diffusion models, our benchmark includes seven representative models: SD-XL [42], SD-3 [43], DALLE-3 [44], Midjourney-v6 [45], FLUX.1-dev [46], Imagen-3 [47], and GLIDE [48]. These text-to-image diffusion models generate images by iteratively denoising random noise under the guidance of textual prompts, making the quality and diversity of input text crucial for dataset construction. To this end, we leverage the Gemini [63] API with carefully designed prompts to synthesize diverse and high-quality image descriptions. For instance, a prompt for generating realistic human descriptions might read: "To create varied descriptions of people in photo or realistic style, I need 1000 distinct sentences, each 20–25 words, please help me. Example: 'woman wearing a red dress in the park,

Disney cartoon style.'". We generate 1500 sentences for each of four categories—people, animals, objects, and landscapes—to ensure broad coverage and content diversity.

*iii) GANs for Deepfake:* Deepfake techniques are often misused to manipulate a person's identity or control facial expressions and movements in portrait images. We include five representative deepfake methods: BlendFace [49], E4S [50], FaceSwap [51], InSwap [52], and SimSwap [53]. We randomly select source and target face images and generate manipulated images using each method accordingly.

*iv) Diffusion for Personalized Generation:* Recently, personalized text-to-image generation based on diffusion models has seen significant progress. These methods [56, 16, 8] enable faithful preservation of a specific identity across novel scenes, actions, and styles, often guided by one or more reference images. In our benchmark, we select five representative personalized generation techniques: InstantID [54], Infinite-ID [8], PhotoMaker [16], BLIP-Diffusion [55], and IP-Adapter [56]. We sample 1500 face images from the FFHQ dataset as identity references and use the Gemini API [63] to generate diverse, high-quality textual descriptions. Based on these inputs, we synthesize realistic and diverse images that combine varying face identities with a wide range of text prompts.

*v) Open-source Platforms:* To better simulate AIGI detection in real-world scenarios, our benchmark additionally collects data from open-source platforms and constructs two representative subsets. The first subset, SocialRF, is sourced from social networks such as X (formerly Twitter), Facebook, and Reddit. We retrieve fake images using common hashtags like #aiart, #aigenerated, and #fakephoto. The second subset, CommunityAI, is compiled from AI art communities including ArtStation, Civitai, and Liblib. These images are often highly realistic, generated by diverse models, and reflect the current landscape of AI-generated content.

**Real Image Collection.** The real images in our benchmark are sampled from three sources: FFHQ, CelebA-HQ, and Open Images V7. FFHQ and CelebA-HQ offer high-resolution facial images, while Open Images V7 introduces greater diversity across ten object categories, including Car, Taxi, and Ambulance. We randomly select an equal number of images from each dataset and merge them to construct the real image set. This set is curated to ensure a one-to-one correspondence with the number of fake images in each subset, enabling balanced evaluation.

## 2.3 Evaluation Metrics

Following the established evaluation paradigm [64], we adopt classification accuracy (Acc.) and average precision (A.P.) as our primary evaluation metrics. However, we observe that overall accuracy alone may not sufficiently capture performance across diverse image generation methods. Given that the ultimate objective of AIGI detection is to accurately identify fake images while minimizing false positives on real ones, we further decompose accuracy into two complementary components: R.Acc., the accuracy of detecting real images, and F.Acc., the accuracy of detecting fake images. This decomposition provides a more nuanced and informative evaluation of a detector's effectiveness.

## 2.4 Task Definition

Following the pipeline for AIGI detection in real-world scenarios, we evaluate the performance of AIGI detectors across four distinct tasks, as detailed below:

**Task 1: Generalization Assessment: Multi-source.** This task evaluates the generalization capability of detectors across different generative models, aiming to assess their effectiveness in diverse scenarios. It specifically emphasizes the detector's ability to handle samples from unknown distributions. The average performance score across 25 subsets is used as the evaluation metric.

**Task 2: Robustness Assessment: Multi-degradation.** This task assesses the baseline performance of detectors under a range of degradation strategies, including JPEG compression, noise interference, and up-and-down sampling. It aims to evaluate the robustness of detectors against unknown degradations that may arise in test data from unseen sources in real-world scenarios. The average performance score across these conditions is used as the evaluation metric.

**Task 3: Data Augmentation Variation Assessment.** AIGI detection is formulated as a binary classification task, making it particularly prone to overfitting to specific features. Data augmentation is a widely used strategy to mitigate overfitting; however, existing pipelines have not thoroughly investigated the impact of different augmentation on AIGI detection performance. This task sys-

tematically evaluates the baseline performance of detectors under a variety of data augmentation methods, including RandomRotation, Color-Jitter, and RandomMask. The goal is to identify which augmentation strategy is most effective for the AIGI detection task. The average performance score across 25 subsets is adopted as the evaluation metric.

**Task 4: Test Data Pre-processing Assessment.** Different detectors are trained under varying settings, and test images often come from diverse sources. As such, data pre-processing is crucial to ensure compatibility with each trained detector. A recent study [58] found that the Resize operation can unintentionally smooth local correlations in synthetic images, thereby weakening subtle discriminative artifacts in the low-level feature space. To address this, the authors proposed replacing Resize with Crop, which better preserves fine details and local structures in synthetic content. Building on this insight, our work systematically evaluates detector performance under both Cropping and Resizing strategies. The goal is to identify the most effective pre-processing method for the AIGI detection task. We use the average performance score across 25 subsets as the evaluation metric.

## 2.5 Detection Methods

To investigate whether AIGI detection is a solved problem, we evaluate 11 SOTA and popular detectors on our AIGIBench. These methods span recent advances in the field, including ResNet-50 (CVPR 2016) [65], CNNDetection (CVPR 2020) [64], Gram-Net (CVPR 2020) [66], LGrad (CVPR 2023) [67], CLIPDetection (CVPR 2023) [19], FreqNet (AAAI 2023) [57], NPR (CVPR 2024) [23], LaDeDa (arXiv 2024) [18], DFFreq (arXiv 2025) [68], AIDE (ICLR 2025) [20], and SAFE (KDD 2025) [58]. For a detailed description of each detector, please refer to Sec. A.7 of the Appendix.

# 3 Experiments and Discussion

Based on four distinct tasks outlined in Sec. 2.4, we organize experiments and analyze from following four perspectives: **i) Generalization Assessment:** We evaluate how well detectors perform on data distributions that differ from those seen during training. **ii) Robustness Assessment:** We examine the factors that affect detector robustness under various image degradation strategies. **iii) Data Augmentation Variation Assessment:** We analyze the impact of different data augmentation on AIGI detection performance. **iv) Test Data Pre-processing Assessment:** We assess the effectiveness of various pre-processing strategies applied during inference for the AIGI detection task. It is worth noting that we conduct experiments under two different settings: Setting-I (illustrated in the Appendix) and Setting-II (shown in this section). A summary of both settings is provided in Sec. 2.1.

## 3.1 Task 1: Generalization Assessment

We evaluate the performance of existing detectors on 25 testsets from our AIGIBench. As shown in Table 3 (F.Acc. and R.Acc. on training setting-II) and Table 7 (Acc. and A.P. on training setting-II), 8 (F.Acc. and R.Acc. on training setting-I), and 9 (Acc. and A.P. on training setting-I) of Sec. A.1, there are significant performance disparities among AI-generated image detection methods when tested across a broad spectrum of generative models and datasets. Several key observations emerge: **i) Overall Best Generalization.** SAFE [58] consistently achieves the highest overall accuracy (Setting-I: 79.2%, Setting-II: 79.9%), establishing itself as the most robust single-model detector with balanced F.Acc. and R.Acc. across diverse generators. AIDE [20] also performs competitively, attaining the highest mean average precision (A.P. = 82.7%) in Setting-II and the highest mean fake accuracy (F.Acc. = 69.0%) in Setting-I. **ii) Effect of Adding SD-v1.4 to Training (Setting II vs. I).** Including the SD-v1.4 dataset in training Setting-II significantly improves R.Acc.. However, this improvement often comes at the cost of reduced F.Acc., indicating a trade-off between sensitivity and precision. **iii) Traditional CNN-based detectors (ResNet-50 [65], CNNDetection [64], and Gram-net [66]).** These detectors achieve high R.Acc. on in-distribution data. However, they demonstrate poor generalization in terms of F.Acc., with a substantial performance decline when tested on datasets generated by unseen generative models. **iv) Advanced Detectors.** More recent methods, including CLIPDetection [19], DFFreq [68], LaDeDa [18], AIDE [20], and SAFE [58], achieve more consistent and robust performance across varied settings. Notably, while the CLIP-based detector CLIPDetection [19] and AIDE [20] show only moderate performance on certain datasets, they achieve the highest mean F.Acc. in Setting-II and Setting-I, respectively, indicating strong generalization ability. Furthermore, frequency-based methods such as FreqNet [57] and DFFreq [68]

Table 3: The generalization results (F.Acc. and R.Acc.) for different AIGI detectors, where the training dataset settings is **Setting-II:** Training on 144K images generated by both SD-v1.4 and ProGAN.

| Test Dataset → | ProGAN | | R3GAN | | StyleGAN3 | | StyleGAN-XL | | StyleSwim | | WFIR | | BlendFace | | E4S | | FaceSwap | |
|---|---|---|---|---|---|---|---|---|---|---|---|---|---|---|---|---|---|---|
| Detectors ↓ | R.Acc. | F.Acc. | R.Acc. | F.Acc. | R.Acc. | F.Acc. | R.Acc. | F.Acc. | R.Acc. | F.Acc. | R.Acc. | F.Acc. | R.Acc. | F.Acc. | R.Acc. | F.Acc. | R.Acc. | F.Acc. |
| Resnet-50 | 100.0 | 98.1 | 95.1 | 2.0 | 95.3 | 46.3 | 95.0 | 25.1 | 95.7 | 70.7 | 95.4 | 3.6 | 92.7 | 0.0 | 93.4 | 0.0 | 96.5 | 0.0 |
| CNNDetection | 99.9 | 95.3 | 98.6 | 2.3 | 99.3 | 9.1 | 98.2 | 0.7 | 98.3 | 6.9 | 99.4 | 0.2 | 98.7 | 6.2 | 98.1 | 4.1 | 98.1 | 1.4 |
| Gram-net | 99.8 | 97.2 | 89.6 | 6.1 | 91.1 | 40.1 | 89.8 | 56.0 | 90.3 | 60.7 | 78.4 | 10.7 | 84.6 | 0.0 | 85.0 | 0.0 | 93.0 | 0.0 |
| LGrad | 99.2 | 94.1 | 84.8 | 23.6 | 88.2 | 52.6 | 84.1 | 74.1 | 85.6 | 77.0 | 83.4 | 17.2 | 80.9 | 2.4 | 82.1 | 0.5 | 86.4 | 2.5 |
| CLIPDetection | 97.9 | 98.9 | 72.9 | 94.1 | 76.5 | 82.6 | 72.5 | 96.7 | 74.7 | 98.1 | 48.0 | 91.9 | 64.6 | 5.5 | 67.0 | 46.9 | 78.4 | 27.3 |
| FreqNet | 99.3 | 99.4 | 64.7 | 59.9 | 68.0 | 98.2 | 64.3 | 95.5 | 64.7 | 97.1 | 30.2 | 89.3 | 46.2 | 0.3 | 50.3 | 1.1 | 73.4 | 6.2 |
| NPR | 100.0 | 98.9 | 93.2 | 8.4 | 93.2 | 63.6 | 92.4 | 28.2 | 93.7 | 77.7 | 95.3 | 7.9 | 89.0 | 0.0 | 89.9 | 0.0 | 95.0 | 0.0 |
| DFFreq | 99.9 | 96.3 | 88.5 | 34.6 | 89.2 | 51.9 | 87.6 | 59.6 | 88.4 | 80.6 | 89.7 | 55.4 | 82.6 | 0.0 | 83.8 | 0.0 | 91.4 | 0.3 |
| LaDeDa | 100.0 | 99.7 | 90.2 | 19.5 | 91.6 | 93.2 | 90.5 | 80.5 | 90.8 | 97.3 | 97.8 | 19.2 | 84.8 | 0.0 | 85.9 | 0.0 | 93.2 | 0.0 |
| AIDE | 99.1 | 95.3 | 86.7 | 99.0 | 85.1 | 91.1 | 85.7 | 91.7 | 85.4 | 82.0 | 99.9 | 42.9 | 79.7 | 23.2 | 82.1 | 6.6 | 89.0 | 14.3 |
| SAFE | 100.0 | 99.9 | 96.6 | 91.2 | 96.6 | 92.9 | 96.5 | 89.7 | 96.3 | 99.3 | 100.0 | 20.7 | 93.8 | 0.8 | 94.8 | 28.9 | 97.1 | 3.3 |

| Test Dataset → | InSwap | | SimSwap | | FLUX1-dev | | Midjourney-V6 | | GLIDE | | DALLE-3 | | Imagen3 | | SD3 | | SDXL | |
|---|---|---|---|---|---|---|---|---|---|---|---|---|---|---|---|---|---|---|
| Detectors ↓ | R.Acc. | F.Acc. | R.Acc. | F.Acc. | R.Acc. | F.Acc. | R.Acc. | F.Acc. | R.Acc. | F.Acc. | R.Acc. | F.Acc. | R.Acc. | F.Acc. | R.Acc. | F.Acc. | R.Acc. | F.Acc. |
| Resnet-50 | 96.6 | 0.0 | 96.2 | 0.0 | 95.6 | 69.1 | 89.5 | 15.5 | 96.2 | 62.4 | 95.5 | 11.8 | 95.8 | 27.9 | 95.3 | 33.1 | 95.3 | 50.4 |
| CNNDetection | 98.3 | 9.7 | 98.0 | 6.2 | 98.5 | 16.3 | 98.9 | 5.8 | 97.6 | 4.6 | 98.1 | 9.8 | 98.2 | 4.2 | 98.4 | 13.3 | 98.4 | 7.3 |
| Gram-net | 92.9 | 0.3 | 93.3 | 0.1 | 89.9 | 39.0 | 78.2 | 9.6 | 92.3 | 50.8 | 90.5 | 16.4 | 90.7 | 10.5 | 91.0 | 14.0 | 91.0 | 36.6 |
| LGrad | 86.2 | 1.3 | 85.8 | 2.3 | 84.1 | 76.6 | 78.7 | 41.5 | 85.9 | 78.9 | 84.3 | 29.7 | 83.9 | 40.2 | 84.4 | 42.4 | 84.4 | 62.7 |
| CLIPDetection | 78.4 | 8.2 | 78.8 | 8.6 | 73.3 | 86.6 | 50.0 | 80.6 | 78.2 | 75.2 | 74.9 | 75.2 | 73.6 | 84.2 | 78.4 | 90.6 | 78.4 | 91.0 |
| FreqNet | 72.6 | 0.9 | 72.0 | 0.6 | 64.7 | 92.4 | 25.3 | 83.6 | 71.8 | 79.7 | 64.2 | 68.2 | 65.8 | 81.5 | 66.6 | 88.1 | 66.6 | 98.9 |
| NPR | 94.6 | 0.0 | 94.8 | 0.0 | 93.3 | 97.2 | 83.9 | 53.8 | 94.8 | 70.3 | 93.0 | 21.2 | 93.5 | 78.2 | 94.2 | 89.7 | 94.2 | 79.0 |
| DFFreq | 91.0 | 0.0 | 90.8 | 0.0 | 88.9 | 64.1 | 74.9 | 54.0 | 91.0 | 86.0 | 88.5 | 14.5 | 89.2 | 62.1 | 89.1 | 73.4 | 89.1 | 88.7 |
| LaDeDa | 93.0 | 0.0 | 92.6 | 0.0 | 90.0 | 99.3 | 77.2 | 83.4 | 92.4 | 81.8 | 90.5 | 9.7 | 90.5 | 92.6 | 91.1 | 99.0 | 91.1 | 98.3 |
| AIDE | 89.6 | 11.4 | 88.2 | 21.5 | 86.0 | 90.0 | 73.0 | 79.8 | 88.4 | 98.4 | 85.7 | 24.5 | 85.7 | 93.9 | 89.3 | 99.3 | 89.3 | 97.6 |
| SAFE | 97.7 | 56.8 | 97.0 | 1.1 | 96.3 | 99.8 | 91.0 | 97.2 | 97.2 | 87.8 | 96.1 | 1.8 | 96.4 | 97.0 | 96.6 | 91.7 | 96.6 | 99.9 |

| Test Dataset → | BLIP | | Infinite-ID | | InstantID | | IP-Adapter | | PhotoMaker | | SocialRF | | CommunityAI | | Mean | | |
|---|---|---|---|---|---|---|---|---|---|---|---|---|---|---|---|---|---|
| Detectors ↓ | R.Acc. | F.Acc. | R.Acc. | F.Acc. | R.Acc. | F.Acc. | R.Acc. | F.Acc. | R.Acc. | F.Acc. | R.Acc. | F.Acc. | R.Acc. | F.Acc. | R.Acc. | F.Acc. | Acc. | A.P. |
| Resnet-50 | 99.2 | 99.8 | 95.7 | 4.0 | 95.3 | 26.8 | 95.5 | 30.2 | 95.4 | 2.7 | 97.3 | 13.4 | 100.0 | 5.1 | 95.7 | 27.9 | 61.9 | 69.3 |
| CNNDetection | 98.0 | 56.5 | 98.3 | 1.1 | 98.3 | 8.1 | 97.9 | 6.0 | 98.4 | 1.7 | 94.7 | 7.5 | 97.3 | 5.3 | 98.2 | 11.6 | 54.9 | 67.0 |
| Gram-net | 98.0 | 99.2 | 90.7 | 10.6 | 90.6 | 59.6 | 90.4 | 18.7 | 90.1 | 10.0 | 92.6 | 11.5 | 99.0 | 6.2 | 90.5 | 26.6 | 58.6 | 62.4 |
| LGrad | 89.4 | 96.6 | 84.8 | 17.0 | 84.0 | 61.0 | 85.5 | 54.9 | 85.0 | 34.6 | 83.7 | 22.2 | 98.9 | 11.4 | 85.8 | 39.6 | 62.9 | 66.6 |
| CLIPDetection | 85.1 | 92.1 | 75.2 | 93.8 | 74.0 | 96.9 | 73.3 | 92.0 | 73.3 | 65.2 | 53.3 | 55.5 | 82.8 | 51.2 | 73.3 | 71.5 | 72.5 | 75.6 |
| FreqNet | 87.7 | 100.0 | 65.4 | 92.7 | 65.8 | 93.9 | 65.8 | 93.0 | 65.5 | 88.6 | 68.5 | 39.3 | 98.9 | 12.2 | 65.9 | 66.4 | 66.2 | 70.1 |
| NPR | 98.4 | 99.9 | 93.1 | 34.6 | 93.5 | 34.1 | 93.1 | 71.8 | 92.6 | 3.6 | 96.3 | 21.9 | 99.9 | 8.2 | 93.8 | 41.9 | 67.9 | 73.9 |
| DFFreq | 96.5 | 99.4 | 89.7 | 50.9 | 88.5 | 95.3 | 88.3 | 78.1 | 88.5 | 87.4 | 96.2 | 17.5 | 99.9 | 7.3 | 89.6 | 51.9 | 71.1 | 75.7 |
| LaDeDa | 98.1 | 100.0 | 90.8 | 32.2 | 90.7 | 82.4 | 91.0 | 90.6 | 90.2 | 66.7 | 97.8 | 19.4 | 100.0 | 9.0 | 91.7 | 54.9 | 73.4 | 79.3 |
| AIDE | 92.8 | 100.0 | 87.1 | 97.5 | 86.6 | 97.0 | 86.6 | 93.5 | 85.9 | 97.5 | 97.2 | 18.4 | 99.0 | 9.3 | 88.1 | 67.0 | 77.6 | 82.7 |
| SAFE | 99.4 | 100.0 | 96.3 | 99.8 | 96.5 | 99.9 | 95.9 | 89.8 | 96.0 | 98.0 | 99.6 | 16.4 | 100.0 | 8.5 | 96.8 | 63.0 | 79.9 | 82.6 |

also perform well, achieving advanced mean F.Acc.. These findings suggest that frequency features enhance the generalization capability of AIGI detectors. **v) Impact on Face-Swap and In-the-Wild Manipulations.** Despite their strong performance on GAN-based noise-to-image and diffusion-based text-to-image generation tasks, existing detectors exhibit substantial performance degradation on more challenging real-world datasets. The results illustrate that detectors struggle significantly on datasets such as DeepFake variants (*e.g.*, FaceSwap, SimSwap), DALLE-3, and social media content (*e.g.*, SocialRF, CommunityAI), often mis-classifying nearly all samples as real. This indicates a pronounced bias and a lack of robustness to distributional shifts introduced by identity-preserving manipulations and stylistically diverse generative content. These limitations highlight a critical gap in current detection models' ability to generalize beyond synthetic benchmarks. To address this, we recommend incorporating representative DeepFake-style manipulations during training, which can expose detectors to a wider range of generative patterns and improve their resilience against real-world forgeries.

In summary, our results highlight significant variability in detector performance and confirm that no single method consistently dominates across all scenarios. These findings underscore the importance of integrating complementary strategies, such as frequency analysis, self-supervised learning, and large-scale pre-training, to build more generalizable detectors for real-world applications.

## 3.2 Task 2: Robustness Assessment

In real-world applications, there is a strong demand for detectors that can effectively adapt to various types of image degradation. In this section, we further investigate this requirement by evaluating the robustness of different detectors under three common degradation types: JPEG compression, Gaussian noise, and Up-down sampling. Specifically, we apply JPEG compression with a quality

Table 4: The overall robust performance of AI-generated image detectors is demonstrated in real-world scenarios, where the training dataset follows Setting-II: training on 144K images generated by both SD-v1.4 and ProGAN. Notably, all reported results represent average values computed across 25 diverse test datasets.

| Detectors → | Resnet-50 | | | CNNDetection | | | Gram-net | | | LGrad | | | CLIPDetection | | | FreqNet | | |
|---|---|---|---|---|---|---|---|---|---|---|---|---|---|---|---|---|---|---|
| Robust Settings ↓ | R.Acc. | F.Acc. | A.P. | R.Acc. | F.Acc. | A.P. | R.Acc. | F.Acc. | A.P. | R.Acc. | F.Acc. | A.P. | R.Acc. | F.Acc. | A.P. | R.Acc. | F.Acc. | A.P. |
| Origin | 95.7 | 27.9 | 69.3 | 98.2 | 11.6 | 67.0 | 90.5 | 26.6 | 62.4 | 85.8 | 39.6 | 66.6 | 73.3 | 71.5 | 75.6 | 65.9 | 66.4 | 70.1 |
| JPEG Compression | 100.0 | 0.1 | 60.1 | 94.3 | 17.2 | 63.7 | 99.6 | 1.2 | 55.8 | 95.9 | 7.3 | 54.6 | 91.1 | 33.0 | 71.6 | 99.5 | 1.4 | 53.0 |
| Gaussian Noise | 98.8 | 4.2 | 66.1 | 97.7 | 2.6 | 47.0 | 95.4 | 10.6 | 60.5 | 91.9 | 17.5 | 60.0 | 78.3 | 58.7 | 72.2 | 73.7 | 48.5 | 66.2 |
| Up-down Sampling | 96.3 | 26.5 | 71.5 | 99.8 | 1.8 | 56.7 | 91.2 | 25.1 | 63.9 | 86.5 | 57.2 | 80.3 | 77.0 | 66.6 | 75.0 | 74.7 | 63.1 | 73.2 |
| Mean | 97.7 | 14.7 | 66.8 | 97.5 | 8.3 | 58.6 | 94.2 | 15.9 | 60.7 | 90.0 | 30.4 | 65.4 | 79.9 | 57.4 | 73.6 | 78.5 | 44.9 | 65.6 |

| Detectors → | NPR | | | DFFreq | | | LaDeDa | | | AIDE | | | SAFE | | |
|---|---|---|---|---|---|---|---|---|---|---|---|---|---|---|---|
| Robust Settings ↓ | R.Acc. | F.Acc. | A.P. | R.Acc. | F.Acc. | A.P. | R.Acc. | F.Acc. | A.P. | R.Acc. | F.Acc. | A.P. | R.Acc. | F.Acc. | A.P. |
| Origin | 93.8 | 41.9 | 73.9 | 89.6 | 51.9 | 75.7 | 91.7 | 54.9 | 79.3 | 88.1 | 67.0 | 82.7 | 96.8 | 63.0 | 82.6 |
| JPEG Compression | 100.0 | 0.2 | 59.2 | 100.0 | 0.1 | 58.8 | 100.0 | 0.0 | 61.6 | 98.9 | 1.5 | 50.3 | 100.0 | 0.0 | 48.7 |
| Gaussian Noise | 98.5 | 6.2 | 68.5 | 86.3 | 32.2 | 69.0 | 98.8 | 2.6 | 68.5 | 93.0 | 22.4 | 72.5 | 100.0 | 1.2 | 46.9 |
| Up-down Sampling | 94.8 | 34.3 | 81.0 | 91.8 | 41.9 | 75.3 | 92.2 | 46.6 | 84.5 | 74.8 | 27.4 | 55.1 | 100.0 | 16.2 | 73.5 |
| Mean | 96.8 | 20.7 | 70.7 | 91.9 | 31.5 | 69.7 | 95.7 | 26.0 | 73.5 | 88.7 | 29.6 | 65.2 | 99.2 | 20.1 | 62.9 |

Table 5: Evaluating the impact of different data augmentation on AIGI detectors under Setting-II, where the training dataset consists of 144K images generated by both SD-v1.4 and ProGAN.

| Data augmentation | | | CLIPDetection | | FreqNet | | NPR | | DFFreq | | SAFE | |
|---|---|---|---|---|---|---|---|---|---|---|---|---|
| Rotation | Jitter | Mask | R.Acc./F.Acc. | Acc./A.P. | R.Acc./F.Acc. | Acc./A.P. | R.Acc./F.Acc. | Acc./A.P. | R.Acc./F.Acc. | Acc./A.P. | R.Acc/F.Acc. | Acc./A.P. |
| | | | 73.3/**71.5** | **72.5**/75.6 | 65.9/**66.4** | 66.2/70.1 | 93.8/41.9 | 67.9/73.9 | 89.6/51.5 | 71.1/75.7 | 94.3/64.6 | 79.5/84.5 |
| ✓ | | | **86.1**/54.9 | 70.5/75.7 | 76.9/58.9 | **68.0**/**71.5** | 93.3/**44.0** | **68.7**/74.1 | **92.1**/52.6 | **72.7**/**77.4** | 82.7/**69.5** | 76.1/82.0 |
| | ✓ | | 79.1/63.7 | 71.4/75.6 | **89.7**/36.8 | 63.5/70.2 | 92.7/38.5 | 65.6/70.8 | 90.0/40.1 | 65.5/70.5 | **99.4**/50.1 | 74.8/**84.8** |
| | | ✓ | 72.8/64.1 | 68.4/73.5 | 74.4/59.4 | 66.9/70.2 | **96.4**/37.0 | 66.7/73.2 | 89.9/52.2 | 71.4/**76.2** | 96.4/60.9 | 78.7/84.5 |
| ✓ | ✓ | | 80.6/62.2 | 71.4/**76.6** | 76.4/51.8 | 64.2/67.7 | 94.3/36.8 | 65.6/70.6 | 86.4/45.3 | 66.2/71.2 | 93.5/65.0 | 79.3/81.5 |
| ✓ | ✓ | ✓ | 79.6/61.3 | 70.5/75.8 | 62.4/62.5 | 62.4/64.8 | **98.1**/32.5 | 65.3/**75.6** | 86.0/47.8 | 67.3/72.4 | 96.8/63.0 | **79.9**/82.6 |

Table 6: Evaluating the impact of different data pre-processing strategies on AI-generated image detectors under Setting-II, where the training dataset consists of 144K images generated by both SD-v1.4 and ProGAN. Note that Crop prep-rocessing primarily improves R.Acc. with limited or even negative impact on F.Acc. across several detectors.

| Detectors → | CLIPDetection | | FreqNet | | NPR | | DFFreq | | LaDeDa | | SAFE | |
|---|---|---|---|---|---|---|---|---|---|---|---|---|
| Process ↓ | R.Acc./F.Acc | Acc./A.P. | R.Acc./F.Acc | Acc./A.P. | R.Acc./F.Acc | Acc./A.P. | R.Acc./F.Acc | Acc./A.P. | R.Acc./F.Acc | Acc./A.P. | R.Acc./F.Acc | Acc./A.P. |
| Resize | 73.3/71.5 | 72.5/75.6 | 65.9/66.4 | 66.2/70.1 | 93.8/41.9 | 67.9/73.9 | 89.6/51.9 | 71.1/75.7 | 91.7/54.9 | 73.4/79.3 | 63.3/66.5 | 64.9/68.6 |
| Crop | 76.9/56.1 | 66.5/68.4 | 84.6/63.5 | 74.2/80.0 | 99.3/36.9 | 68.2/81.9 | 96.1/51.7 | 74.4/81.1 | 98.9/56.1 | 77.5/82.5 | 96.8/63.0 | 79.9/82.6 |

factor of 50 to simulate compression artifacts. To introduce noise artifacts, Gaussian noise with a standard deviation of $\sigma = 4$ is added. Additionally, to emulate typical sampling artifacts observed in practical scenarios, we apply down-sampling using the nearest neighbor algorithm to reduce the image size by half, followed by up-sampling back to the original resolution.

As shown in Table 4, our experimental results demonstrate that under the Origin setting (*i.e.*, clean test images), most detectors achieve high real image accuracy (R.Acc.) and reasonable fake image accuracy (F.Acc.). However, substantial performance degradation is observed under perturbation settings. Specifically: **i)** JPEG Compression and Gaussian Noise cause a dramatic decline in F.Acc. for all detectors, often approaching 0%, while R.Acc. remains artificially high (close to 100%). This indicates a strong bias toward predicting "real" under these perturbations, resulting in a failure to detect fake images. **ii)** Up-down Sampling induces a more moderate drop in F.Acc. and is relatively less harmful for some detectors, such as LaDeDa [18], CLIPDetection [19], and FreqNet [57]. **iii)** Among all methods, the CLIP-based CLIPDetection and frequency-aware FreqNet achieve the best overall trade-off between R.Acc. and F.Acc. across all robustness settings, suggesting superior generalization capabilities. In particular, CLIPDetection [19] performs binary classification in a feature space not explicitly trained for forgery detection. It utilizes fixed embeddings from large pre-trained CLIP-ViT models and applies a lightweight classification strategy based on nearest-neighbor lookup and linear probing over a feature library constructed from both real and fake images. By decoupling from model-specific cues, this method exhibits greater resilience to various types of

degradation. In contrast, FreqNet [57] operates in the frequency domain, allowing it to capture forgery patterns less sensitive to spatial perturbations, thus further enhancing its robustness in real-world scenarios. Additional results for Training Setting-I are provided in Table 10 in the Appendix.

Overall, the Mean row reveals that while most detectors consistently maintain high real image accuracy (R.Acc. $\geq 90\%$) under perturbations, their F.Acc. drops significantly (often below 35%), indicating compromised detection reliability in practical scenarios. This underscores the critical need for developing detectors that are robust to various types of image degradation. In this context, exploring robust real/fake discriminative features, either through large pre-trained models or frequency-domain representations, emerges as a promising and impactful research direction.

### 3.3 Task 3: Data Augmentation Variation Assessment

Data augmentation is a widely adopted strategy to mitigate overfitting and enhance the generalization ability of detectors. In this section, we investigate the impact of three common augmentation techniques, rotation, color-jitter, and masking, on five advanced detectors: CLIPDetection [19], FreqNet [57], NPR [23], DFFreq [68], and SAFE [58]. We evaluate five augmentation settings: Rotation, Color Jitter, Masking, Rotation + Color Jitter, and Rotation + Color Jitter + Masking.

The main results are summarized in Table 5, with detailed configurations reported in Table 11, Table 12, Table 13, Table 14, and Table 15, respectively. The findings reveal several insights: **i)** Data augmentation generally improves R.Acc. but may degrade F.Acc., particularly for CLIPDetection and FreqNet; **ii)** Combining all three augmentations offers no clear advantage and can impair performance consistency, especially for models sensitive to semantic or frequency cues such as FreqNet and DFFreq; **iii)** The effectiveness of each augmentation strategy is model-dependent, underscoring the importance of designing augmentation-aware training pipelines tailored to specific detectors. Overall, these results suggest that commonly used data augmentation strategies offer limited benefit for enhancing the performance of AIGI detectors, and in some cases, may even introduce trade-offs.

### 3.4 Task 4: Test Data Pre-processing Assessment

Data pre-processing is essential to ensure compatibility between input samples and the trained detectors. A recent study [58] demonstrated that the Resize operation can unintentionally smooth local correlations in synthetic images, thereby weakening subtle discriminative artifacts in the low-level feature space. To address this issue, it proposed replacing Resize with Crop, which better preserves intricate details and local structures in synthetic content. This enhances the detector's ability to capture fine-grained artifacts embedded in generated images. In this section, we investigate the impact of two common pre-processing strategies, Resize and Crop, on the performance of six representative and advanced AI-generated image detectors: CLIPDetection [19], FreqNet [57], NPR [23], DFFreq [68], LaDeDa [18], and SAFE [58].

Table 6 and Table 16 of Appendix highlight the critical impact of data pre-processing on detector performance, with the Crop strategy generally outperforming Resize across multiple advanced detectors. However, a closer examination of R.Acc. and F.Acc. reveals a key observation: the performance gains from Crop are predominantly attributed to improvements in R.Acc., while F.Acc. remains largely unchanged or even declines in certain cases. For instance, SAFE [58] and FreqNet [57] demonstrate substantial increases in R.Acc. (from 63.3% to 96.8% and 65.9% to 84.6%, respectively), yet their F.Acc. either drops or shows marginal improvement. This asymmetry can be attributed to the modality gap between real and synthetic images. Real images, typically captured by a limited range of devices and scenes, exhibit concentrated and consistent modalities, characterized by strong local structural similarity and statistical stability. Crop, by preserving key image regions and avoiding the blurring effects of Resize, retains high-frequency local features and texture details, thereby enhancing the model's ability to recognize real content—resulting in improved R.Acc. In contrast, fake images are generated by diverse models and processes, leading to high modality variance. While Crop may amplify certain generative artifacts, it can also remove distinctive cues (*e.g.*, boundary artifacts), resulting in selective enhancement or even information loss. This modality asymmetry explains why Crop consistently boosts real image detection, reducing false positives, but provides limited or inconsistent improvements in detecting synthetic content.

# 4  Conclusions, Limitations, and Board Impacts

**Conclusions.** This paper presents AIGIBench, a novel benchmark designed to evaluate the effectiveness of detectors in identifying AI-generated images. AIGIBench comprises 23 diverse subsets of synthetic images, encompassing both cutting-edge and widely adopted image generation techniques, as well as real-world samples sourced from social media and AI art platforms. To better reflect practical scenarios, AIGIBench systematically evaluates detector performance across four core tasks: multi-source generalization, robustness to image degradation, sensitivity to data augmentation, and impact of test-time pre-processing. Experimental results demonstrate that existing detectors suffer significant performance degradation under real-world conditions, underscoring that AIGI detection remains a formidable challenge. The comprehensive evaluations and in-depth analyses enabled by AIGIBench aim to foster new research directions and promote further progress in the field.

**Limitation and Future Works.** AIGIBench currently considers two widely adopted training settings: i) training on ProGAN, and ii) training on ProGAN combined with SD-v1.4. To further enhance the benchmark, we plan to introduce a leaderboard showcasing the performance of various detectors trained on larger datasets sampled from diverse sources. Additionally, we aim to incorporate a broader range of representative detectors and datasets, thereby providing a more comprehensive platform for evaluating detection performance. AIGIBench is designed as a continually evolving resource to support the research community and accelerate the development of advanced AIGI detectors.

**Board Impacts.** AIGIBench advances the detection of AIGI by providing a comprehensive and realistic evaluation framework. It reveals the limitations of existing detectors and enables standardized comparisons, thereby fostering the development of more robust and generalizable forensic methods. This benchmark plays a crucial role in mitigating the societal risks associated with synthetic media, including misinformation and digital fraud. Moreover, it serves as a valuable resource for both researchers and practitioners. Nonetheless, a potential ethical concern arises from the risk that malicious actors could exploit AIGIBench to enhance the evasiveness of generative models.

# 5  Acknowledgments

This work was supported the National Natural Science Foundation of China under grant U22B2062, 62502215, 62172233, 62472231, 62172232, 62402229; the Natural Science Foundation of Jiangsu under Grants BK20250735; the Jiangsu Provincial Science and Technology Major Project (No. BG2024042); the General Program of Natural Science Research in Universities of Jiangsu under Grants 25KJB520027.

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

# A Appendix

Table 7: The generalization results (Acc. and A.P.) for AI-generated image detectors in real-world scenarios, where the training dataset settings is **Setting-II:** Training on 144K images generated by both SD-v1.4 and ProGAN.

| Test Dataset → | ProGAN | | R3GAN | | StyleGAN3 | | StyleGAN-XL | | StyleSwim | | WFIR | | BlendFace | | E4S | | FaceSwap | |
|---|---|---|---|---|---|---|---|---|---|---|---|---|---|---|---|---|---|---|
| Detectors ↓ | Acc. | A.P. | Acc. | A.P. | Acc. | A.P. | Acc. | A.P. | Acc. | A.P. | Acc. | A.P. | Acc. | A.P. | Acc. | A.P. | Acc. | A.P. |
| Resnet-50 | 99.1 | 99.9 | 48.5 | 53.7 | 70.8 | 88.3 | 60.0 | 76.7 | 83.2 | 93.7 | 49.5 | 56.8 | 46.4 | 34.3 | 46.7 | 32.9 | 48.8 | 39.8 |
| CNNDetection | 97.6 | 99.9 | 50.4 | 52.7 | 55.8 | 73.1 | 52.8 | 64.2 | 52.6 | 76.5 | 49.8 | 50.0 | 52.4 | 73.4 | 51.1 | 68.9 | 50.3 | 58.7 |
| Gram-net | 98.5 | 100.0 | 47.9 | 52.5 | 65.6 | 77.9 | 72.9 | 83.7 | 75.5 | 86.0 | 44.5 | 43.9 | 42.3 | 33.3 | 42.5 | 32.5 | 47.0 | 34.6 |
| LGrad | 96.6 | 99.8 | 54.4 | 58.7 | 70.5 | 80.5 | 65.7 | 74.6 | 81.3 | 90.0 | 51.7 | 49.4 | 41.8 | 34.9 | 41.5 | 32.8 | 45.3 | 37.5 |
| CLIPDetection | 98.4 | 99.9 | 83.5 | 91.2 | 79.6 | 84.5 | 84.6 | 93.3 | 86.4 | 95.2 | 70.0 | 82.0 | 35.0 | 35.3 | 57.0 | 57.1 | 53.1 | 52.4 |
| FreqNet | 99.3 | 100.0 | 62.3 | 56.8 | 83.0 | 92.4 | 79.8 | 84.1 | 80.8 | 91.8 | 58.5 | 48.9 | 23.3 | 34.1 | 25.8 | 34.7 | 40.4 | 43.4 |
| NPR | 99.4 | 100.0 | 50.8 | 61.1 | 78.4 | 91.7 | 60.3 | 75.3 | 85.7 | 94.9 | 51.6 | 65.5 | 44.5 | 34.7 | 45.0 | 34.4 | 48.1 | 43.6 |
| DFFreq | 98.1 | 100.0 | 61.7 | 74.0 | 90.1 | 96.4 | 73.7 | 83.4 | 84.5 | 92.9 | 74.6 | 82.2 | 41.5 | 35.2 | 42.1 | 34.8 | 47.6 | 45.6 |
| LaDeDa | 99.8 | 100.0 | 54.8 | 72.6 | 92.4 | 96.9 | 94.7 | 98.5 | 94.0 | 98.5 | 58.5 | 86.9 | 42.4 | 42.1 | 42.9 | 49.3 | 47.1 | 40.9 |
| AIDE | 97.2 | 99.6 | 92.9 | 97.1 | 88.1 | 91.4 | 88.7 | 93.2 | 83.7 | 89.3 | 71.4 | 90.8 | 51.5 | 54.2 | 44.3 | 44.3 | 52.1 | 56.3 |
| SAFE | 100.0 | 100.0 | 93.9 | 98.2 | 89.7 | 97.6 | 93.1 | 97.6 | 97.8 | 99.6 | 60.4 | 81.8 | 47.3 | 45.6 | 47.6 | 46.0 | 50.7 | 45.7 |

| Test Dataset → | InSwap | | SimSwap | | FLUX1-dev | | Midjourney-V6 | | GLIDE | | DALLE-3 | | Imagen3 | | SD3 | | SDXL | |
|---|---|---|---|---|---|---|---|---|---|---|---|---|---|---|---|---|---|---|
| Detectors ↓ | Acc. | A.P. | Acc. | A.P. | Acc. | A.P. | Acc. | A.P. | Acc. | A.P. | Acc. | A.P. | Acc. | A.P. | Acc. | A.P. | Acc. | A.P. |
| Resnet-50 | 48.8 | 39.7 | 48.1 | 40.3 | 82.3 | 93.6 | 52.5 | 57.4 | 79.3 | 93.4 | 53.7 | 68.4 | 61.9 | 78.6 | 64.2 | 83.7 | 72.8 | 89.8 |
| CNNDetection | 54.5 | 77.9 | 52.1 | 70.0 | 57.4 | 72.3 | 52.3 | 59.8 | 51.1 | 60.0 | 53.9 | 68.6 | 51.2 | 57.4 | 55.8 | 73.1 | 52.8 | 64.2 |
| Gram-net | 47.1 | 38.0 | 46.7 | 36.4 | 64.5 | 75.1 | 43.9 | 41.6 | 71.6 | 84.3 | 53.5 | 61.1 | 50.6 | 54.7 | 52.5 | 60.0 | 63.8 | 76.9 |
| LGrad | 44.6 | 35.0 | 44.2 | 37.6 | 80.4 | 88.1 | 60.4 | 64.5 | 82.4 | 91.0 | 57.2 | 62.7 | 62.2 | 69.7 | 63.4 | 72.1 | 73.6 | 82.8 |
| CLIPDetection | 43.7 | 40.2 | 43.7 | 40.4 | 80.0 | 79.5 | 65.3 | 61.5 | 76.7 | 80.3 | 75.1 | 76.3 | 78.9 | 79.3 | 84.5 | 87.2 | 84.7 | 88.0 |
| FreqNet | 37.5 | 42.1 | 36.5 | 41.9 | 78.5 | 87.3 | 53.9 | 55.9 | 75.8 | 77.4 | 66.2 | 61.0 | 73.6 | 80.7 | 77.3 | 82.6 | 82.7 | 95.2 |
| NPR | 47.8 | 40.7 | 47.4 | 42.7 | 95.2 | 99.0 | 68.8 | 76.9 | 82.5 | 94.3 | 57.1 | 70.0 | 85.9 | 94.4 | 91.9 | 97.2 | 86.6 | 94.4 |
| DFFreq | 47.0 | 41.2 | 45.6 | 43.8 | 76.6 | 86.3 | 64.6 | 68.1 | 88.5 | 96.3 | 51.8 | 58.6 | 75.7 | 87.3 | 81.3 | 90.8 | 88.9 | 95.8 |
| LaDeDa | 47.0 | 47.4 | 46.3 | 42.3 | 94.6 | 98.7 | 80.3 | 86.9 | 87.1 | 95.0 | 50.1 | 59.8 | 91.6 | 97.2 | 95.1 | 98.7 | 94.7 | 98.5 |
| AIDE | 50.9 | 54.6 | 54.9 | 62.7 | 88.0 | 93.4 | 76.4 | 83.0 | 93.4 | 97.7 | 55.1 | 63.1 | 89.8 | 95.2 | 94.3 | 98.3 | 93.5 | 95.7 |
| SAFE | 49.7 | 49.9 | 49.0 | 49.5 | 98.1 | 99.7 | 94.1 | 98.4 | 92.5 | 97.9 | 49.0 | 45.8 | 96.7 | 98.8 | 94.1 | 98.8 | 98.3 | 99.7 |

| Test Dataset → | BLIP | | Infinite-ID | | InstantID | | IP-Adapter | | PhotoMaker | | SocialRF | | CommunityAI | | Mean | |
|---|---|---|---|---|---|---|---|---|---|---|---|---|---|---|---|---|
| Detectors ↓ | Acc. | A.P. | Acc. | A.P. | Acc. | A.P. | Acc. | A.P. | Acc. | A.P. | Acc. | A.P. | Acc. | A.P. | Acc. | A.P. |
| Resnet-50 | 99.5 | 100.0 | 49.9 | 62.6 | 61.0 | 80.7 | 62.8 | 83.1 | 49.1 | 58.7 | 55.3 | 61.5 | 52.5 | 64.2 | 61.9 | 69.3 |
| CNNDetection | 77.2 | 92.9 | 49.7 | 49.5 | 53.2 | 80.2 | 52.0 | 65.8 | 50.1 | 58.2 | 51.1 | 50.6 | 51.3 | 59.1 | 55.1 | 67.1 |
| Gram-net | 98.6 | 99.9 | 50.6 | 60.1 | 75.1 | 85.7 | 54.5 | 64.2 | 50.1 | 58.6 | 52.1 | 53.0 | 52.6 | 66.1 | 58.6 | 62.4 |
| LGrad | 93.0 | 97.4 | 50.9 | 54.6 | 72.6 | 81.5 | 70.3 | 78.3 | 59.9 | 67.2 | 53.5 | 54.9 | 55.5 | 69.4 | 62.9 | 66.6 |
| CLIPDetection | 88.6 | 95.8 | 84.5 | 89.6 | 85.4 | 93.5 | 82.6 | 87.3 | 69.3 | 72.3 | 54.4 | 55.2 | 67.0 | 73.2 | 72.5 | 75.6 |
| FreqNet | 93.8 | 100.0 | 79.0 | 74.5 | 79.8 | 86.3 | 78.8 | 79.9 | 77.0 | 74.9 | 54.2 | 58.1 | 55.9 | 69.7 | 66.2 | 70.1 |
| NPR | 99.2 | 100.0 | 63.9 | 80.4 | 63.8 | 79.2 | 82.4 | 91.7 | 48.1 | 43.6 | 59.1 | 68.4 | 54.0 | 62.9 | 67.9 | 73.9 |
| DFFreq | 97.9 | 99.6 | 70.4 | 82.7 | 91.9 | 97.2 | 83.2 | 91.3 | 88.0 | 94.0 | 57.6 | 63.3 | 54.5 | 52.1 | 71.1 | 75.7 |
| LaDeDa | 99.0 | 99.9 | 61.5 | 76.9 | 86.5 | 90.4 | 90.8 | 94.3 | 78.4 | 90.7 | 58.6 | 68.3 | 54.5 | 56.3 | 73.4 | 79.3 |
| AIDE | 96.4 | 95.5 | 92.2 | 94.7 | 91.8 | 96.3 | 90.0 | 95.4 | 91.7 | 95.6 | 57.8 | 65.0 | 54.1 | 61.0 | 77.6 | **82.7** |
| SAFE | 99.7 | 100.0 | 96.9 | 99.2 | 98.2 | 99.6 | 92.8 | 98.1 | 97.0 | 99.3 | 58.0 | 64.2 | 54.2 | 55.2 | **79.9** | 82.6 |

## A.1 More Results on Task 1: Generalization Assessment

In this section, we provide more quantitative result on Task 1: Generalization Assessment.

**Generalization results (Acc. and A.P.) on Training Setting-II.** Table 7 reports the generalization performance of various AI-generated image detectors under Setting-II, where models are trained on a mixture of SD-v1.4 and ProGAN images and evaluated on a diverse range of generative models and datasets. Performance is measured in terms of Accuracy (Acc.) and Average Precision (A.P.). The results reveal several key findings:

*i) Classical detectors tend to overfit the training distribution.* Traditional CNN-based detectors (*e.g.*, ResNet-50, Gram-net, CNNDetection, LGrad) achieve near-perfect performance on the training domain (ProGAN), with Acc. and A.P. exceeding 97%. However, their generalization to unseen generative models (*e.g.*, R3GAN, FaceSwap, StyleGAN-XL) is poor, often yielding accuracies below 55% and A.P. under 60%.

*ii) CLIP-based and frequency-based detectors exhibit improved generalization.* Detectors such as CLIPDetection and DFFreq leverage CLIP or frequency-domain features that are less sensitive to dataset-specific generation artifacts. These approaches achieve substantially better performance on

Table 8: The generalization results (F.Acc. and R.Acc.) for AI-generated image detectors in real-world scenarios, where the training dataset settings is **Setting-I:** Training on 72K images generated by ProGAN.

| Method | ProGAN | | R3GAN | | StyleGAN3 | | StyleGAN-XL | | StyleSwim | | WFIR | | BlendFace | | E4S | | FaceSwap | |
|---|---|---|---|---|---|---|---|---|---|---|---|---|---|---|---|---|---|---|
| | R.Acc. | F.Acc. | R.Acc. | F.Acc. | R.Acc. | F.Acc. | R.Acc. | F.Acc. | R.Acc. | F.Acc. | R.Acc. | F.Acc. | R.Acc. | F.Acc. | R.Acc. | F.Acc. | R.Acc. | F.Acc. |
| Resnet-50 | 99.9 | 98.6 | 64.0 | 83.2 | 64.7 | 82.6 | 64.3 | 91.0 | 63.7 | 90.8 | 33.4 | 90.9 | 49.4 | 0.2 | 51.5 | 0.0 | 70.2 | 0.1 |
| CNNDetection | 99.9 | 95.9 | 99.3 | 0.5 | 99.1 | 29.0 | 99.3 | 0.2 | 98.8 | 17.2 | 99.1 | 25.0 | 98.5 | 0.0 | 99.0 | 0.0 | 99.0 | 0.0 |
| Gram-net | 98.8 | 92.4 | 78.8 | 64.0 | 80.4 | 67.1 | 79.8 | 60.3 | 78.2 | 67.4 | 65.3 | 57.3 | 72.6 | 3.6 | 74.3 | 1.7 | 82.2 | 3.8 |
| LGrad | 99.3 | 96.7 | 49.6 | 81.7 | 54.4 | 65.4 | 49.3 | 73.2 | 49.6 | 86.9 | 66.2 | 44.1 | 55.3 | 21.8 | 54.9 | 23.4 | 48.1 | 7.5 |
| CLIPDetection | 99.9 | 99.1 | 90.4 | 95.6 | 88.6 | 48.1 | 90.5 | 83.7 | 90.6 | 98.3 | 88.0 | 94.0 | 89.2 | 5.1 | 89.6 | 55.1 | 91.0 | 47.1 |
| FreqNet | 100.0 | 99.5 | 81.2 | 48.5 | 76.0 | 66.9 | 80.2 | 33.7 | 80.0 | 71.8 | 81.1 | 12.1 | 75.7 | 14.7 | 76.0 | 22.1 | 81.7 | 8.4 |
| NPR | 99.9 | 99.7 | 77.0 | 84.6 | 77.2 | 90.9 | 76.5 | 72.4 | 76.6 | 92.8 | 76.8 | 78.9 | 67.6 | 0.1 | 67.9 | 0.1 | 80.9 | 0.3 |
| DFFreq | 100.0 | 99.5 | 70.7 | 96.0 | 68.8 | 98.9 | 70.3 | 92.8 | 70.5 | 89.4 | 60.0 | 97.6 | 57.3 | 8.0 | 60.1 | 10.0 | 77.4 | 11.1 |
| LaDeDa | 100.0 | 99.9 | 70.0 | 75.8 | 71.9 | 95.5 | 70.2 | 94.0 | 69.6 | 95.5 | 75.5 | 93.5 | 56.4 | 0.0 | 58.0 | 0.0 | 75.3 | 6.3 |
| AIDE | 99.6 | 96.4 | 78.6 | 99.3 | 72.5 | 85.8 | 78.0 | 90.9 | 78.1 | 92.2 | 93.7 | 86.8 | 72.2 | 14.6 | 73.6 | 5.1 | 81.8 | 10.7 |
| SAFE | 100.0 | 100.0 | 90.1 | 97.9 | 90.0 | 92.7 | 90.0 | 100.0 | 89.5 | 100.0 | 99.0 | 36.1 | 83.0 | 10.3 | 84.7 | 6.4 | 92.8 | 14.1 |

| Method | InSwap | | SimSwap | | FLUX1-dev | | Midjourney V6 | | GLIDE | | DALLE-3 | | Imagen3 | | SD3 | | SDXL | |
|---|---|---|---|---|---|---|---|---|---|---|---|---|---|---|---|---|---|---|
| | R.Acc. | F.Acc. | R.Acc. | F.Acc. | R.Acc. | F.Acc. | R.Acc. | F.Acc. | R.Acc. | F.Acc. | R.Acc. | F.Acc. | R.Acc. | F.Acc. | R.Acc. | F.Acc. | R.Acc. | F.Acc. |
| Resnet-50 | 71.4 | 0.7 | 69.4 | 0.4 | 63.6 | 65.8 | 31.4 | 63.7 | 70.5 | 47.0 | 65.8 | 57.8 | 65.0 | 52.4 | 64.6 | 52.0 | 64.6 | 87.6 |
| CNNDetection | 99.1 | 0.0 | 99.2 | 0.1 | 98.9 | 1.6 | 98.2 | 1.6 | 99.4 | 3.7 | 99.1 | 1.7 | 99.2 | 2.7 | 98.7 | 7.0 | 98.7 | 6.4 |
| Gram-net | 83.2 | 7.8 | 82.5 | 7.1 | 79.1 | 20.3 | 64.0 | 23.8 | 82.6 | 19.8 | 79.9 | 15.4 | 79.0 | 31.2 | 80.8 | 16.3 | 80.8 | 38.9 |
| LGrad | 48.2 | 26.7 | 47.8 | 24.2 | 50.4 | 79.1 | 55.6 | 62.9 | 49.0 | 64.4 | 50.2 | 48.8 | 49.4 | 73.4 | 54.2 | 78.3 | 54.2 | 78.3 |
| CLIPDetection | 90.7 | 6.2 | 90.8 | 9.4 | 90.1 | 2.4 | 86.8 | 2.7 | 91.8 | 47.0 | 91.2 | 2.5 | 90.7 | 5.5 | 91.9 | 6.0 | 91.9 | 9.6 |
| FreqNet | 82.9 | 20.0 | 83.3 | 14.8 | 81.1 | 65.4 | 71.3 | 43.4 | 81.9 | 53.4 | 80.6 | 13.3 | 80.0 | 79.9 | 80.2 | 75.0 | 80.2 | 92.7 |
| NPR | 80.9 | 0.2 | 80.9 | 0.1 | 75.8 | 98.5 | 55.3 | 84.1 | 80.1 | 76.4 | 76.6 | 38.6 | 77.2 | 92.1 | 75.9 | 96.9 | 75.9 | 98.1 |
| DFFreq | 76.1 | 15.5 | 76.3 | 7.0 | 70.0 | 85.9 | 42.6 | 81.3 | 75.9 | 75.7 | 70.1 | 29.3 | 71.4 | 86.7 | 70.6 | 91.4 | 70.6 | 93.2 |
| LaDeDa | 75.6 | 0.0 | 74.5 | 0.1 | 69.7 | 98.8 | 41.0 | 93.0 | 75.3 | 97.5 | 70.1 | 8.3 | 70.3 | 91.9 | 68.8 | 98.4 | 68.8 | 97.3 |
| AIDE | 82.1 | 9.3 | 81.8 | 12.6 | 77.7 | 92.2 | 65.0 | 90.7 | 81.2 | 98.9 | 77.5 | 36.3 | 78.6 | 94.2 | 79.6 | 99.7 | 79.6 | 98.2 |
| SAFE | 92.8 | 7.7 | 91.9 | 9.0 | 89.2 | 99.9 | 73.4 | 96.7 | 93.2 | 96.7 | 93.0 | 1.7 | 89.2 | 94.4 | 91.3 | 99.9 | 91.3 | 99.9 |

| Method | BLIP | | Infinite-ID | | InstantID | | IP-Adapter | | PhotoMaker | | SocialRF | | CommunityAI | | Mean | |
|---|---|---|---|---|---|---|---|---|---|---|---|---|---|---|---|---|
| | R.Acc. | F.Acc. | R.Acc. | F.Acc. | R.Acc. | F.Acc. | R.Acc. | F.Acc. | R.Acc. | F.Acc. | R.Acc. | F.Acc. | R.Acc. | F.Acc. | R.Acc. | F.Acc. |
| Resnet-50 | 84.4 | 97.6 | 64.1 | 55.3 | 64.8 | 80.1 | 64.4 | 59.0 | 65.4 | 22.1 | 69.0 | 28.0 | 99.8 | 10.1 | 65.6 | 52.7 |
| CNNDetection | 99.6 | 0.0 | 99.1 | 2.5 | 99.0 | 1.2 | 99.2 | 0.9 | 99.1 | 0.1 | 97.2 | 3.4 | 99.1 | 0.8 | **99.0** | 8.1 |
| Gram-net | 88.2 | 23.5 | 80.7 | 13.9 | 80.2 | 20.9 | 79.5 | 8.4 | 79.3 | 14.2 | 88.1 | 15.6 | 96.4 | 6.5 | 80.6 | 28.1 |
| LGrad | 43.8 | 69.2 | 53.8 | 8.8 | 49.9 | 57.9 | 50.3 | 40.7 | 49.3 | 14.6 | 60.5 | 38.3 | 95.4 | 16.7 | 55.5 | 51.1 |
| CLIPDetection | 92.5 | 23.1 | 91.1 | 53.8 | 90.6 | 44.2 | 91.2 | 33.4 | 90.4 | 3.6 | 91.2 | 7.2 | 97.3 | 2.4 | 91.1 | 35.4 |
| FreqNet | 86.7 | 33.1 | 80.3 | 6.2 | 81.4 | 29.0 | 80.1 | 31.6 | 80.9 | 2.5 | 75.5 | 22.6 | 95.1 | 7.6 | 81.3 | 38.7 |
| NPR | 89.3 | 98.7 | 76.1 | 49.6 | 77.8 | 64.8 | 76.9 | 81.2 | 76.4 | 16.3 | 83.0 | 26.1 | 99.9 | 8.9 | 78.3 | 58.0 |
| DFFreq | 87.6 | 98.9 | 70.7 | 66.8 | 70.9 | 90.7 | 71.7 | 75.6 | 70.0 | 89.6 | 85.6 | 24.8 | 98.5 | 9.0 | 72.6 | 65.8 |
| LaDeDa | 86.9 | 100.0 | 69.3 | 88.7 | 70.9 | 91.2 | 70.0 | 94.6 | 70.4 | 68.1 | 84.4 | 28.1 | 99.6 | 10.0 | 72.5 | 65.1 |
| AIDE | 85.6 | 99.8 | 78.4 | 99.3 | 79.2 | 96.0 | 79.3 | 95.0 | 77.4 | 89.3 | 92.1 | 21.4 | 91.6 | 11.1 | 80.6 | **69.0** |
| SAFE | 98.6 | 100.0 | 89.9 | 99.3 | 90.0 | 100.0 | 89.9 | 97.1 | 89.1 | 99.8 | 98.6 | 17.9 | 100.0 | 9.1 | 91.1 | 67.2 |

out-of-distribution (OOD) models. For instance, CLIPDetection attains 91.2% A.P. on R3GAN and 95.2% A.P. on StyleSwim, clearly outperforming conventional CNN-based baselines.

*iii) Advanced methods achieve state-of-the-art generalization.* SAFE demonstrates the strongest generalization capabilities, achieving 93.9% Acc. and 98.2% A.P. on R3GAN, and consistently high scores across various GAN- and diffusion-based models (*e.g.*, 97.8% Acc. on StyleSwim and 97.6% A.P. on StyleGAN-XL). AIDE and LaDeDa also maintain robust and balanced performance, indicating enhanced cross-model generalizability.

*iv) Generalization to deepfake models remains a major challenge.* Even advanced detectors such as SAFE and AIDE perform poorly on datasets like DALLE-3, BlendFace, and SocialRF, underscoring the limitations of current methods and the need for greater training diversity and more robust feature representations in the detection of AI-generated content.

**Generalization results (F.Acc. and R.Acc.) on Training Setting-I.** Table 8 presents the generalization performance of various AI-generated image detectors under Setting-I, where all models are trained on 72K images synthesized by ProGAN and evaluated across a diverse set of generative models and datasets. Performance is reported using R.Acc. and F.Acc. metrics. Consistent with earlier observations, detectors trained exclusively on ProGAN data exhibit significant overfitting and poor generalization, particularly when evaluated on deepfake-generated images. Only a few methods, most notably SAFE and AIDE, demonstrate meaningful cross-model generalization, underscoring the importance of diverse training data or more robust feature learning strategies for real-world applicability.

**Generalization results (Acc. and A.P.) on Training Setting-I.** Table 9 reports the generalization performance of various AI-generated image detectors under Setting-I, where all models are trained on 72K images synthesized by ProGAN and evaluated across a diverse range of generative models and datasets. Performance is measured using accuracy (Acc.) and average precision (A.P.) metrics.

Consistent with previous findings, training solely on images from a single GAN (ProGAN) results in overfitting, highlighting generalization as a central challenge. Modern detectors that exploit frequency artifacts (*e.g.*, SAFE), large-scale pre-training (*e.g.*, AIDE), or robust architectural designs (*e.g.*, AIDE) demonstrate better cross-model generalization, performing well on both GAN- and diffusion-based samples. These results underscore the necessity of more generalized training paradigms or domain-adaptive strategies to enhance robustness in real-world detection scenarios.

Table 9: The generalization results (Acc. and A.P.) for AI-generated image detectors in real-world scenarios, where the training dataset settings is **Setting-I:** Training on 72K images generated by ProGAN.

| Method | ProGAN | | R3GAN | | StyleGAN3 | | StyleGAN-XL | | StyleSwim | | WFIR | | BlendFace | | E4S | | FaceSwap | |
|---|---|---|---|---|---|---|---|---|---|---|---|---|---|---|---|---|---|---|
| | Acc. | A.P. | Acc. | A.P. | Acc. | A.P. | Acc. | A.P. | Acc. | A.P. | Acc. | A.P. | Acc. | A.P. | Acc. | A.P. | Acc. | A.P. |
| Resnet-50 | 99.3 | 100.0 | 73.6 | 72.1 | 73.6 | 80.6 | 77.7 | 82.0 | 77.3 | 82.5 | 62.2 | 66.3 | 24.8 | 33.0 | 25.8 | 32.6 | 35.5 | 35.9 |
| CNNDetection | 97.8 | 100.0 | 49.9 | 48.4 | 64.0 | 91.6 | 49.7 | 60.0 | 58.0 | 88.9 | 62.1 | 89.4 | 49.3 | 34.2 | 49.6 | 35.2 | 50.0 | 44.3 |
| Gram-net | 95.6 | 99.5 | 71.4 | 77.3 | 73.8 | 81.6 | 70.0 | 74.3 | 72.8 | 79.1 | 61.3 | 62.6 | 38.1 | 36.6 | 38.0 | 35.2 | 43.4 | 39.1 |
| LGrad | 98.0 | 99.9 | 65.6 | 69.1 | 59.9 | 64.3 | 61.2 | 62.8 | 68.1 | 75.0 | 55.6 | 53.1 | 38.6 | 39.4 | 39.2 | 39.4 | 28.2 | 33.3 |
| CLIPDetection | 99.5 | 100.0 | 93.0 | 98.6 | 68.4 | 79.0 | 87.1 | 90.0 | 94.5 | 99.3 | 91.0 | 97.8 | 47.2 | 42.7 | 72.4 | 82.5 | 69.3 | 79.4 |
| FreqNet | 99.3 | 100.0 | 62.3 | 56.8 | 83.0 | 92.4 | 79.8 | 84.1 | 80.8 | 91.8 | 58.5 | 48.9 | 23.3 | 34.1 | 25.8 | 34.7 | 40.4 | 43.4 |
| NPR | 99.8 | 100.0 | 80.8 | 75.6 | 84.1 | 89.4 | 74.5 | 75.9 | 84.7 | 91.7 | 77.8 | 77.0 | 33.9 | 33.3 | 34.0 | 34.4 | 41.1 | 39.5 |
| DFFreq | 99.7 | 100.0 | 83.3 | 77.1 | 83.8 | 94.6 | 81.5 | 81.4 | 79.9 | 90.3 | 76.6 | 87.4 | 32.7 | 36.6 | 35.1 | 37.9 | 45.4 | 49.1 |
| LaDeDa | 100.0 | 100.0 | 72.9 | 69.2 | 83.7 | 86.3 | 82.1 | 84.2 | 82.6 | 86.4 | 84.5 | 91.3 | 28.2 | 38.0 | 29.0 | 42.4 | 41.2 | 40.2 |
| AIDE | 98.0 | 99.8 | 89.0 | 94.8 | 79.2 | 77.8 | 84.4 | 87.7 | 85.2 | 85.1 | 90.3 | 97.0 | 43.4 | 43.8 | 39.4 | 38.3 | 46.7 | 43.7 |
| SAFE | 100.0 | 100.0 | 94.0 | 97.3 | 91.4 | 95.9 | 95.0 | 99.0 | 94.7 | 100.0 | 67.6 | 83.6 | 46.7 | 45.2 | 45.5 | 42.9 | 53.9 | 57.6 |

| Method | InSwap | | SimSwap | | FLUX1-dev | | Midjourney V6 | | GLIDE | | DALLE-3 | | Imagen3 | | SD3 | | SDXL | |
|---|---|---|---|---|---|---|---|---|---|---|---|---|---|---|---|---|---|---|
| | Acc. | A.P. | Acc. | A.P. | Acc. | A.P. | Acc. | A.P. | Acc. | A.P. | Acc. | A.P. | Acc. | A.P. | Acc. | A.P. | Acc. | A.P. |
| Resnet-50 | 36.5 | 37.4 | 35.1 | 37.1 | 64.7 | 67.3 | 47.5 | 50.2 | 58.7 | 60.3 | 61.8 | 63.9 | 58.7 | 61.7 | 58.3 | 60.6 | 76.1 | 81.9 |
| CNNDetection | 50.1 | 44.3 | 49.7 | 45.3 | 50.3 | 50.3 | 49.9 | 39.3 | 51.5 | 67.3 | 50.4 | 53.8 | 51.0 | 50.3 | 52.8 | 66.3 | 52.6 | 65.5 |
| Gram-net | 46.0 | 43.8 | 44.8 | 44.1 | 49.7 | 50.5 | 43.9 | 44.0 | 51.2 | 51.4 | 47.7 | 46.6 | 55.1 | 57.2 | 48.5 | 47.8 | 59.8 | 65.9 |
| LGrad | 37.7 | 38.5 | 36.1 | 38.1 | 64.6 | 70.0 | 59.2 | 62.1 | 56.7 | 66.2 | 49.5 | 48.6 | 61.4 | 66.9 | 66.2 | 71.4 | 64.0 | 66.3 |
| CLIPDetection | 48.9 | 48.2 | 50.1 | 51.5 | 46.3 | 39.6 | 44.8 | 35.6 | 69.4 | 81.7 | 46.8 | 38.4 | 48.1 | 44.1 | 49.0 | 48.6 | 50.7 | 53.6 |
| FreqNet | 37.5 | 42.1 | 36.5 | 41.9 | 78.5 | 87.3 | 57.6 | 57.7 | 78.2 | 66.2 | 61.0 | 73.6 | 80.7 | 77.6 | 84.0 | 86.4 | 93.6 | |
| NPR | 41.0 | 36.9 | 40.5 | 38.5 | 87.2 | 93.6 | 69.7 | 71.1 | 78.2 | 85.2 | 57.6 | 61.9 | 84.6 | 91.0 | 86.4 | 93.5 | 87.0 | 93.7 |
| DFFreq | 46.8 | 48.8 | 41.8 | 45.6 | 77.9 | 85.4 | 61.6 | 68.5 | 85.8 | 94.4 | 49.9 | 53.0 | 79.1 | 87.7 | 80.9 | 89.3 | 81.9 | 90.1 |
| LaDeDa | 38.3 | 41.8 | 37.3 | 39.6 | 84.2 | 88.0 | 67.0 | 68.0 | 86.4 | 89.4 | 39.2 | 46.3 | 81.1 | 84.6 | 83.6 | 87.6 | 83.1 | 88.0 |
| AIDE | 46.1 | 43.0 | 47.2 | 46.8 | 84.9 | 86.6 | 77.9 | 78.1 | 90.0 | 89.3 | 56.9 | 57.0 | 86.4 | 90.8 | 89.7 | 94.1 | 88.9 | 90.0 |
| SAFE | 50.7 | 49.8 | 50.5 | 51.8 | 94.6 | 99.5 | 85.1 | 94.7 | 93.1 | 97.2 | 45.4 | 44.5 | 91.8 | 96.8 | 93.9 | 98.4 | 95.6 | 99.4 |

| Method | BLIP | | Infinite-ID | | InstantID | | IP-Adapter | | PhotoMaker | | SocialRF | | CommunityAI | | Mean | |
|---|---|---|---|---|---|---|---|---|---|---|---|---|---|---|---|---|
| | Acc. | A.P. | Acc. | A.P. | Acc. | A.P. | Acc. | A.P. | Acc. | A.P. | Acc. | A.P. | Acc. | A.P. | Acc. | A.P. |
| Resnet-50 | 91.0 | 94.7 | 59.7 | 62.8 | 72.5 | 77.9 | 61.7 | 59.3 | 43.7 | 44.8 | 48.5 | 49.9 | 54.9 | 68.4 | 59.2 | 62.5 |
| CNNDetection | 49.8 | 44.7 | 50.8 | 55.8 | 50.1 | 67.1 | 50.0 | 50.1 | 49.6 | 51.5 | 50.3 | 46.3 | 49.9 | 50.5 | 53.6 | 57.6 |
| Gram-net | 55.9 | 61.5 | 47.3 | 49.6 | 50.5 | 51.0 | 44.0 | 41.4 | 46.8 | 45.5 | 51.9 | 52.6 | 51.4 | 54.0 | 54.4 | 55.7 |
| LGrad | 56.4 | 57.8 | 31.3 | 34.4 | 53.9 | 54.8 | 45.5 | 46.3 | 32.0 | 35.8 | 49.6 | 49.3 | 56.4 | 63.0 | 53.4 | 56.2 |
| CLIPDetection | 57.8 | 68.9 | 72.4 | 84.2 | 67.4 | 79.5 | 62.3 | 73.9 | 47.0 | 41.4 | 49.2 | 46.2 | 49.9 | 50.0 | 50.0 | 66.4 |
| FreqNet | 93.8 | 99.9 | 79.0 | 74.5 | 79.8 | 86.3 | 78.8 | 79.9 | 77.0 | 74.9 | 54.2 | 58.1 | 55.9 | 69.7 | 60.2 | 61.7 |
| NPR | 94.0 | 97.0 | 62.0 | 62.8 | 71.3 | 75.9 | 79.0 | 81.1 | 46.4 | 52.0 | 54.5 | 58.4 | 54.4 | 60.0 | 68.2 | 70.8 |
| DFFreq | 93.2 | 95.5 | 68.8 | 72.6 | 80.7 | 85.1 | 73.6 | 75.8 | 79.7 | 76.6 | 55.8 | 58.9 | 54.6 | 51.3 | 69.2 | 73.3 |
| LaDeDa | 93.4 | 96.7 | 79.0 | 67.5 | 81.0 | 81.8 | 82.3 | 82.2 | 69.3 | 66.6 | 56.2 | 61.7 | 54.8 | 55.9 | 68.8 | 71.3 |
| AIDE | 92.7 | 96.7 | 88.9 | 87.9 | 87.6 | 89.0 | 87.1 | 91.0 | 83.4 | 80.9 | 56.8 | 62.1 | 51.3 | 52.8 | 74.9 | 76.2 |
| SAFE | 99.3 | 99.8 | 94.6 | 98.9 | 95.0 | 99.2 | 93.5 | 97.6 | 94.4 | 99.2 | 58.2 | 66.6 | 54.5 | 68.2 | **79.2** | **83.3** |

## A.2  More Results on Task 2: Robustness Assessment

In this section, we evaluate the robustness of various AI-generated image detectors under three common real-world degradations: JPEG compression, Gaussian noise, and up-down sampling, using Training Setting-I. Specifically, JPEG compression with a quality factor of 50 is applied to simulate compression artifacts. Gaussian noise with a standard deviation of $\sigma = 4$ is introduced to mimic sensor noise. For sampling artifacts, we apply down-sampling using nearest neighbor interpolation to reduce the image resolution by half, followed by up-sampling back to the original size.

As shown in Table 10, our experimental results demonstrate that under the Origin setting (*i.e.*, clean test images), most detectors achieve high real image accuracy (R.Acc.) and reasonable fake image accuracy (F.Acc.). However, significant performance degradation is observed when perturbations are introduced. Key observations include: **i) Traditional CNN-based Detectors Perform Poorly Under Perturbations:** Methods such as CNNDetection, Gram-net, and ResNet-

50 suffer substantial robustness drops, particularly under JPEG compression and Gaussian noise. For example, CNNDetection achieves a high R.Acc. of 99.0% on clean images, but its F.Acc. drops sharply to 0.8% under JPEG compression. **2) Some Advanced Methods Offer Moderate Robustness:** Some Advanced methods SAFE and LaDeDa show slightly better average performance than traditional CNNs under distribution shifts. However, they still suffer from F.Acc. dropping significantly under JPEG compression and Gaussian noise. **3) CLIP-Based and Frequency-Based Methods Show Better Trade-offs Between Robustness and Accuracy:** CLIP-based method AIDE and Frequency-based method DFFreq generally maintain more stable performance across perturbations. Specifically, both DFFreq and AIDE have the best robust for detecting fake images to different perturbations (Mean F.Acc.= 51.5). **4) JPEG Compression is the Most Challenging Perturbation:** Nearly all detectors experience a dramatic drop in F.Acc., often near 0%, under JPEG compression. This indicates a critical vulnerability in current detection methods.

In summary, while traditional CNN-based detectors perform well in clean settings, they fail to generalize under realistic image distortions. In contrast, more recent approaches, particularly AIDE, DFFreq, and SAFE, exhibit improved robustness, making them better suited for deployment in practical scenarios involving diverse and degraded inputs.

Table 10: The overall robust performance of AI-generated image detectors is demonstrated in real-world scenarios, where the training dataset follows Setting-I: training on 72K images generated by ProGAN. Notably, all reported results represent average values computed across 25 diverse test datasets.

| Detectors → | Resnet-50 | | | CNNDetection | | | Gram-net | | | LGrad | | | CLIPDetection | | | FreqNet | | |
|---|---|---|---|---|---|---|---|---|---|---|---|---|---|---|---|---|---|---|
| Robust Settings ↓ | R.Acc. | F.Acc. | A.P. | R.Acc. | F.Acc. | A.P. | R.Acc. | F.Acc. | A.P. | R.Acc. | F.Acc. | A.P. | R.Acc. | F.Acc. | A.P. | R.Acc. | F.Acc. | A.P. |
| Origin | 65.6 | 52.7 | 62.5 | 99.0 | 8.1 | 57.6 | 80.6 | 28.1 | 55.7 | 55.5 | 51.1 | 56.2 | 91.1 | 35.4 | 66.4 | 81.3 | 38.7 | 61.7 |
| JPEG Compression | 99.5 | 0.8 | 56.2 | 98.2 | 6.9 | 54.4 | 97.9 | 3.5 | 55.5 | 61.6 | 46.2 | 53.9 | 85.4 | 30.5 | 61.1 | 88.7 | 7.3 | 44.6 |
| Gaussian Noise | 78.2 | 35.7 | 65.7 | 99.3 | 5.1 | 52.8 | 87.9 | 18.4 | 58.0 | 62.7 | 34.6 | 49.7 | 85.2 | 29.0 | 58.6 | 85.0 | 16.5 | 52.0 |
| Up-down Sampling | 77.6 | 49.4 | 67.6 | 99.4 | 2.9 | 52.6 | 86.3 | 26.4 | 61.4 | 53.8 | 63.0 | 56.3 | 77.8 | 35.3 | 58.6 | 82.2 | 55.4 | 72.7 |
| Mean | 80.2 | 34.7 | 63.0 | 99.0 | 5.8 | 54.4 | 88.2 | 19.1 | 57.7 | 58.4 | 48.7 | 54.0 | 84.9 | 32.6 | 61.2 | 84.3 | 29.5 | 57.8 |

| Detectors → | NPR | | | DFFreq | | | LaDeDa | | | AIDE | | | SAFE | | |
|---|---|---|---|---|---|---|---|---|---|---|---|---|---|---|---|
| Robust Settings ↓ | R.Acc. | F.Acc. | A.P. | R.Acc. | F.Acc. | A.P. | R.Acc. | F.Acc. | A.P. | R.Acc. | F.Acc. | A.P. | R.Acc. | F.Acc. | A.P. |
| Origin | 78.3 | 58.0 | 70.8 | 72.6 | 65.8 | 73.3 | 72.5 | 65.1 | 71.3 | 80.6 | 69.0 | 76.2 | 91.1 | 67.2 | 83.3 |
| JPEG Compression | 99.9 | 0.1 | 58.7 | 99.8 | 0.6 | 59.8 | 99.9 | 0.1 | 60.9 | 94.9 | 6.9 | 54.0 | 99.7 | 0.5 | 55.9 |
| Gaussian Noise | 90.1 | 20.7 | 68.1 | 62.3 | 90.3 | 74.9 | 88.6 | 26.6 | 73.8 | 71.6 | 44.7 | 60.0 | 83.2 | 51.6 | 78.0 |
| Up-down Sampling | 86.3 | 43.9 | 78.7 | 81.3 | 49.4 | 70.6 | 81.1 | 50.4 | 78.8 | 25.6 | 85.5 | 69.7 | 100.0 | 16.2 | 73.5 |
| Mean | 88.6 | 30.7 | 69.1 | 79.0 | 51.5 | 69.7 | 85.5 | 35.6 | 71.2 | 68.2 | 51.5 | 65.0 | 93.5 | 33.9 | 72.7 |

### A.3 More Results on Task 3: Data Augmentation Variation Assessment

This section presents detailed results of various data augmentation strategies for AI-generated image (AIGI) detectors under Setting-II, including Rotation (Table 11), Color-Jitter (Table 12), Masking (Table 13), Rotation & Color-Jitter (Table 14), and Rotation & Color-Jitter & Masking (Table 15). The results are consistent with the observations in Table 5, demonstrating that standard data augmentation techniques provide limited improvement for AIGI detection performance and, in some cases, may even lead to performance trade-offs.

### A.4 More Results on Task 4: Test Data Pre-processing Assessment

Table 16 further validates the asymmetrical impact of data pre-processing under Training Setting-I, where detectors are trained solely on ProGAN-generated images. Across all models, Crop preprocessing notably boosts real image accuracy (R.Acc.), while its effect on fake image accuracy (F.Acc.) is minimal or even negative. For example, LaDeDa and SAFE show large R.Acc. improvements (from 72.5% to 98.9%, and 58.4% to 91.1%, respectively), but their F.Acc. increases only modestly or slightly decreases. This suggests that Crop enhances detectors' ability to identify authentic content by preserving high-frequency textures and local structures lost in Resize. However, the diverse generative artifacts in fake images make F.Acc. harder to consistently improve—Crop may expose certain artifacts while masking others. Thus, the results highlight a modality asymmetry: real images benefit more from localized detail preservation, whereas the heterogeneous nature of synthetic content leads to mixed outcomes.

Table 11: A detailed evaluation of the impact of Rotation data augmentation strategies on AI-generated image detectors under Setting-II, where the training dataset consists of 144K images generated by both SD-v1.4 and ProGAN.

| Test Dataset → | ProGAN | | R3GAN | | StyleGAN3 | | StyleGAN-XL | | StyleSwim | | WFIR | | BlendFace | | E4S | | FaceSwap | |
|---|---|---|---|---|---|---|---|---|---|---|---|---|---|---|---|---|---|---|
| Detectors ↓ | Acc. | A.P. | Acc. | A.P. | Acc. | A.P. | Acc. | A.P. | Acc. | A.P. | Acc. | A.P. | Acc. | A.P. | Acc. | A.P. | Acc. | A.P. |
| CLIPDetection | 96.3 | 99.5 | 75.5 | 85.4 | 79.3 | 88.7 | 72.2 | 82.4 | 87.6 | 94.2 | 66.6 | 74.6 | 42.3 | 35.1 | 51.3 | 51.7 | 50.1 | 49.0 |
| FreqNet | 99.3 | 100.0 | 68.0 | 70.0 | 84.9 | 92.1 | 76.0 | 81.0 | 82.5 | 91.5 | 58.5 | 53.2 | 32.6 | 34.3 | 33.6 | 34.3 | 41.3 | 42.0 |
| NPR | 99.8 | 100.0 | 56.7 | 72.7 | 87.6 | 94.5 | 67.4 | 81.9 | 87.5 | 95.3 | 53.6 | 73.0 | 44.5 | 34.5 | 44.9 | 36.2 | 47.8 | 42.6 |
| DFFreq | 99.7 | 100.0 | 63.9 | 78.5 | 90.2 | 96.2 | 75.9 | 87.1 | 87.0 | 95.3 | 67.9 | 76.4 | 43.1 | 35.3 | 43.6 | 35.3 | 48.4 | 46.6 |
| SAFE | 100.0 | 100.0 | 90.1 | 96.1 | 89.6 | 94.3 | 89.7 | 94.1 | 89.3 | 96.3 | 61.5 | 83.2 | 40.8 | 42.9 | 39.1 | 41.5 | 58.3 | 62.5 |

| Test Dataset → | InSwap | | SimSwap | | FLUX1-dev | | Midjourney-V6 | | GLIDE | | DALLE-3 | | Imagen3 | | SD3 | | SDXL | |
|---|---|---|---|---|---|---|---|---|---|---|---|---|---|---|---|---|---|---|
| Detectors ↓ | Acc. | A.P. | Acc. | A.P. | Acc. | A.P. | Acc. | A.P. | Acc. | A.P. | Acc. | A.P. | Acc. | A.P. | Acc. | A.P. | Acc. | A.P. |
| CLIPDetection | 46.6 | 38.7 | 46.4 | 39.1 | 82.7 | 90.1 | 77.2 | 82.7 | 64.0 | 74.2 | 79.2 | 87.5 | 82.0 | 88.8 | 82.7 | 90.7 | 85.7 | 93.0 |
| FreqNet | 40.6 | 41.2 | 40.2 | 42.2 | 85.2 | 92.2 | 64.0 | 64.9 | 59.3 | 65.2 | 64.5 | 66.6 | 71.1 | 83.3 | 85.2 | 90.5 | 85.5 | 91.9 |
| NPR | 47.6 | 39.6 | 47.0 | 41.7 | 95.6 | 99.0 | 70.6 | 76.6 | 86.7 | 95.2 | 52.9 | 63.8 | 87.1 | 94.8 | 92.1 | 96.7 | 90.9 | 94.8 |
| DFFreq | 48.3 | 43.1 | 46.7 | 43.4 | 87.4 | 93.8 | 69.5 | 73.7 | 86.4 | 94.8 | 57.5 | 67.4 | 79.6 | 90.9 | 89.0 | 94.9 | 93.4 | 97.6 |
| SAFE | 52.2 | 54.7 | 51.4 | 56.5 | 89.5 | 96.7 | 72.1 | 89.5 | 91.6 | 95.5 | 39.8 | 43.2 | 89.9 | 97.2 | 91.9 | 97.3 | 92.0 | 96.2 |

| Test Dataset → | BLIP | | Infinite-ID | | InstantID | | IP-Adapter | | PhotoMaker | | SocialRF | | CommunityAI | | Mean | |
|---|---|---|---|---|---|---|---|---|---|---|---|---|---|---|---|---|
| Detectors ↓ | Acc. | A.P. | Acc. | A.P. | Acc. | A.P. | Acc. | A.P. | Acc. | A.P. | Acc. | A.P. | Acc. | A.P. | Acc. | A.P. |
| CLIPDetection | 78.0 | 89.3 | 77.7 | 87.0 | 85.1 | 92.1 | 76.7 | 85.4 | 57.7 | 65.3 | 58.6 | 63.3 | 61.7 | 65.8 | 70.5 | 75.7 |
| FreqNet | 94.9 | 99.4 | 78.4 | 75.6 | 79.9 | 87.8 | 82.9 | 83.5 | 75.8 | 80.8 | 54.6 | 58.1 | 55.0 | 66.2 | 68.0 | 71.5 |
| NPR | 99.2 | 99.9 | 52.7 | 71.6 | 59.0 | 73.3 | 86.0 | 92.2 | 48.2 | 58.1 | 58.0 | 66.1 | 54.1 | 58.0 | 68.1 | 74.1 |
| DFFreq | 98.9 | 99.9 | 71.3 | 82.4 | 87.7 | 92.5 | 81.1 | 89.7 | 88.1 | 94.3 | 59.2 | 67.7 | 54.7 | 58.4 | _72.7_ | _77.4_ |
| SAFE | 99.3 | 100.0 | 90.8 | 95.1 | 89.8 | 96.3 | 89.9 | 95.5 | 89.5 | 95.0 | 59.5 | 67.4 | 54.9 | 61.9 | **76.1** | **82.0** |

Table 12: A detailed evaluation of the impact of Color-jitter data augmentation strategies on AI-generated image detectors under Setting-II, where the training dataset consists of 144K images generated by both SD-v1.4 and ProGAN.

| Test Dataset → | ProGAN | | R3GAN | | StyleGAN3 | | StyleGAN-XL | | StyleSwim | | WFIR | | BlendFace | | E4S | | FaceSwap | |
|---|---|---|---|---|---|---|---|---|---|---|---|---|---|---|---|---|---|---|
| Detectors ↓ | Acc. | A.P. | Acc. | A.P. | Acc. | A.P. | Acc. | A.P. | Acc. | A.P. | Acc. | A.P. | Acc. | A.P. | Acc. | A.P. | Acc. | A.P. |
| CLIPDetection | 98.1 | 99.9 | 84.6 | 92.8 | 78.4 | 86.6 | 86.7 | 93.3 | 88.5 | 96.8 | 75.6 | 86.3 | 39.8 | 37.7 | 60.6 | 64.6 | 55.3 | 58.0 |
| FreqNet | 99.6 | 100.0 | 57.7 | 69.4 | 67.5 | 80.8 | 68.7 | 80.4 | 82.6 | 92.1 | 53.0 | 51.6 | 45.3 | 41.2 | 47.9 | 44.3 | 47.6 | 43.5 |
| NPR | 99.8 | 100.0 | 51.3 | 62.5 | 72.6 | 87.2 | 57.4 | 71.5 | 83.7 | 93.6 | 49.9 | 58.5 | 44.4 | 34.0 | 44.1 | 32.3 | 47.7 | 39.6 |
| DFFreq | 98.5 | 100.0 | 53.1 | 62.3 | 70.7 | 83.4 | 65.8 | 77.4 | 80.7 | 90.9 | 52.6 | 43.9 | 42.5 | 34.5 | 42.6 | 34.2 | 47.6 | 42.8 |
| SAFE | 100.0 | 100.0 | 70.0 | 96.5 | 69.2 | 91.5 | 88.2 | 99.3 | 89.4 | 99.2 | 59.9 | 86.1 | 50.5 | 56.0 | 50.5 | 57.8 | 51.0 | 56.1 |

| Test Dataset → | InSwap | | SimSwap | | FLUX1-dev | | Midjourney-V6 | | GLIDE | | DALLE-3 | | Imagen3 | | SD3 | | SDXL | |
|---|---|---|---|---|---|---|---|---|---|---|---|---|---|---|---|---|---|---|
| Detectors ↓ | Acc. | A.P. | Acc. | A.P. | Acc. | A.P. | Acc. | A.P. | Acc. | A.P. | Acc. | A.P. | Acc. | A.P. | Acc. | A.P. | Acc. | A.P. |
| CLIPDetection | 45.8 | 43.0 | 46.8 | 44.9 | 74.1 | 76.3 | 65.8 | 63.6 | 66.1 | 71.7 | 75.0 | 78.2 | 71.9 | 75.2 | 80.8 | 85.4 | 76.1 | 81.4 |
| FreqNet | 50.7 | 51.2 | 50.5 | 52.9 | 66.8 | 78.7 | 50.0 | 49.2 | 75.4 | 87.3 | 60.4 | 69.8 | 54.6 | 65.0 | 60.8 | 73.3 | 80.7 | 89.9 |
| NPR | 47.5 | 37.0 | 46.8 | 39.3 | 90.7 | 96.1 | 57.8 | 63.1 | 84.5 | 94.4 | 57.4 | 69.0 | 75.6 | 87.4 | 83.5 | 92.5 | 80.0 | 89.9 |
| DFFreq | 47.6 | 40.9 | 45.9 | 41.1 | 84.9 | 92.5 | 55.6 | 58.0 | 77.6 | 89.6 | 67.7 | 78.0 | 66.7 | 79.0 | 75.0 | 85.7 | 80.4 | 89.3 |
| SAFE | 50.7 | 57.4 | 50.2 | 59.4 | 93.5 | 99.3 | 76.3 | 92.1 | 86.6 | 97.1 | 50.7 | 67.9 | 68.7 | 93.5 | 70.2 | 90.8 | 97.8 | 99.8 |

| Test Dataset → | BLIP | | Infinite-ID | | InstantID | | IP-Adapter | | PhotoMaker | | SocialRF | | CommunityAI | | Mean | |
|---|---|---|---|---|---|---|---|---|---|---|---|---|---|---|---|---|
| Detectors ↓ | Acc. | A.P. | Acc. | A.P. | Acc. | A.P. | Acc. | A.P. | Acc. | A.P. | Acc. | A.P. | Acc. | A.P. | Acc. | A.P. |
| CLIPDetection | 82.9 | 91.3 | 82.6 | 88.9 | 86.5 | 93.7 | 79.0 | 83.1 | 65.5 | 70.8 | 56.3 | 56.4 | 63.3 | 70.6 | _71.4_ | _75.6_ |
| FreqNet | 98.6 | 100.0 | 63.8 | 75.9 | 73.4 | 85.4 | 74.5 | 85.3 | 51.9 | 61.0 | 52.6 | 53.0 | 52.6 | 72.9 | 63.5 | 70.2 |
| NPR | 99.1 | 100.0 | 69.3 | 82.1 | 54.4 | 66.3 | 85.9 | 92.2 | 46.6 | 47.7 | 57.9 | 66.3 | 52.7 | 68.4 | 65.6 | 70.8 |
| DFFreq | 98.1 | 99.8 | 59.5 | 70.2 | 75.1 | 85.1 | 79.0 | 87.7 | 58.3 | 69.6 | 58.1 | 63.5 | 54.5 | 63.4 | 65.5 | 70.5 |
| SAFE | 99.9 | 100.0 | 97.9 | 99.4 | 99.0 | 99.9 | 92.7 | 99.3 | 97.7 | 99.8 | 56.4 | 62.8 | 52.4 | 59.5 | **74.8** | **84.8** |

## A.5 More Confirmatory Analyses

In our data curation process, we employed CLIP image embeddings with a cosine similarity threshold of 0.98 to identify and remove near-duplicate instances. Additionally, we used an off-the-shelf aesthetic quality predictor and retained only high-resolution (4K) images with the highest aesthetic scores to increase the difficulty and discriminative power of the detection benchmark. Low-quality or visually homogeneous images often reduce the evaluation challenge and may artificially inflate detector performance. By removing redundant and low-aesthetic samples, we ensure that the curated dataset better reflects real-world, high-fidelity scenarios and enables a more rigorous and meaningful evaluation of detection capabilities.

Table 13: A detailed evaluation of the impact of Mask data augmentation strategies on AI-generated image detectors under Setting-II, where the training dataset consists of 144K images generated by both SD-v1.4 and ProGAN.

| Test Dataset → | ProGAN | | R3GAN | | StyleGAN3 | | StyleGAN-XL | | StyleSwim | | WFIR | | BlendFace | | E4S | | FaceSwap | |
|---|---|---|---|---|---|---|---|---|---|---|---|---|---|---|---|---|---|---|
| Detectors ↓ | Acc. | A.P. | Acc. | A.P. | Acc. | A.P. | Acc. | A.P. | Acc. | A.P. | Acc. | A.P. | Acc. | A.P. | Acc. | A.P. | Acc. | A.P. |
| CLIPDetection | 98.3 | 99.9 | 76.5 | 87.2 | 74.2 | 84.4 | 73.1 | 81.8 | 84.3 | 96.6 | 71.0 | 83.6 | 54.7 | 54.3 | 70.8 | 73.9 | 69.1 | 73.6 |
| FreqNet | 99.5 | 100.0 | 50.4 | 53.6 | 85.9 | 91.5 | 81.9 | 84.3 | 82.3 | 89.4 | 57.4 | 49.5 | 29.3 | 34.3 | 31.1 | 34.7 | 42.6 | 44.1 |
| NPR | 99.8 | 100.0 | 50.4 | 62.0 | 80.4 | 94.5 | 58.1 | 75.2 | 86.8 | 95.6 | 50.0 | 67.4 | 46.7 | 34.0 | 47.2 | 34.3 | 49.2 | 41.6 |
| DFFreq | 99.6 | 100.0 | 57.4 | 69.8 | 86.8 | 94.5 | 71.2 | 82.2 | 85.7 | 94.6 | 65.1 | 69.1 | 41.8 | 36.1 | 42.2 | 36.6 | 49.3 | 52.0 |
| SAFE | 100.0 | 100.0 | 90.9 | 97.0 | 92.8 | 98.6 | 79.4 | 93.5 | 97.8 | 99.9 | 51.8 | 82.8 | 49.4 | 49.5 | 48.0 | 46.6 | 54.6 | 66.7 |

| Test Dataset → | InSwap | | SimSwap | | FLUX1-dev | | Midjourney-V6 | | GLIDE | | DALLE-3 | | Imagen3 | | SD3 | | SDXL | |
|---|---|---|---|---|---|---|---|---|---|---|---|---|---|---|---|---|---|---|
| Detectors ↓ | Acc. | A.P. | Acc. | A.P. | Acc. | A.P. | Acc. | A.P. | Acc. | A.P. | Acc. | A.P. | Acc. | A.P. | Acc. | A.P. | Acc. | A.P. |
| CLIPDetection | 63.8 | 65.7 | 61.3 | 64.3 | 55.3 | 56.1 | 53.5 | 52.8 | 72.0 | 77.8 | 70.7 | 75.2 | 59.8 | 63.5 | 76.2 | 82.4 | 76.2 | 83.2 |
| FreqNet | 41.0 | 42.7 | 40.3 | 42.0 | 81.6 | 88.4 | 57.0 | 58.2 | 71.8 | 74.0 | 61.4 | 60.1 | 74.3 | 82.4 | 79.9 | 85.6 | 86.6 | 95.5 |
| NPR | 48.9 | 38.6 | 48.5 | 40.1 | 96.4 | 99.1 | 66.8 | 78.9 | 74.5 | 92.0 | 54.7 | 71.2 | 89.3 | 96.9 | 93.6 | 98.1 | 86.2 | 95.5 |
| DFFreq | 47.3 | 44.9 | 45.8 | 45.0 | 88.3 | 95.1 | 63.4 | 67.8 | 90.0 | 96.5 | 56.5 | 64.7 | 80.0 | 90.1 | 85.6 | 93.5 | 90.3 | 95.9 |
| SAFE | 51.8 | 60.4 | 51.5 | 62.0 | 97.3 | 99.5 | 89.7 | 96.1 | 96.7 | 99.3 | 47.9 | 44.5 | 78.2 | 93.2 | 86.8 | 96.8 | 98.4 | 99.9 |

| Test Dataset → | BLIP | | Infinite-ID | | InstantID | | IP-Adapter | | PhotoMaker | | SocialRF | | CommunityAI | | Mean | |
|---|---|---|---|---|---|---|---|---|---|---|---|---|---|---|---|---|
| Detectors ↓ | Acc. | A.P. | Acc. | A.P. | Acc. | A.P. | Acc. | A.P. | Acc. | A.P. | Acc. | A.P. | Acc. | A.P. | Acc. | A.P. |
| CLIPDetection | 63.5 | 68.4 | 73.1 | 79.8 | 78.4 | 86.7 | 74.9 | 80.7 | 50.2 | 50.5 | 56.7 | 57.1 | 53.4 | 58.1 | 68.4 | 73.5 |
| FreqNet | 96.1 | 99.9 | 78.1 | 74.5 | 82.7 | 88.1 | 81.5 | 84.0 | 71.2 | 69.7 | 54.9 | 58.9 | 55.0 | 69.2 | 66.9 | 70.2 |
| NPR | 99.5 | 100.0 | 53.7 | 76.5 | 55.0 | 72.8 | 72.5 | 89.9 | 48.8 | 53.2 | 58.1 | 66.9 | 53.4 | 55.4 | 66.7 | 73.2 |
| DFFreq | 98.5 | 99.9 | 62.8 | 74.2 | 91.9 | 95.5 | 84.8 | 91.1 | 87.6 | 93.2 | 59.2 | 65.4 | 54.9 | 58.1 | _71.4_ | _76.2_ |
| SAFE | 99.9 | 100.0 | 97.9 | 99.8 | 98.0 | 99.8 | 97.4 | 99.4 | 97.7 | 99.7 | 59.1 | 69.6 | 54.3 | 58.0 | **78.7** | **84.5** |

Table 14: A detailed evaluation of the impact of Rotation & Color-Jitter data augmentation strategies on AI-generated image detectors under Setting-II, where the training dataset consists of 144K images generated by both SD-v1.4 and ProGAN.

| Test Dataset → | ProGAN | | R3GAN | | StyleGAN3 | | StyleGAN-XL | | StyleSwim | | WFIR | | BlendFace | | E4S | | FaceSwap | |
|---|---|---|---|---|---|---|---|---|---|---|---|---|---|---|---|---|---|---|
| Detectors ↓ | Acc. | A.P. | Acc. | A.P. | Acc. | A.P. | Acc. | A.P. | Acc. | A.P. | Acc. | A.P. | Acc. | A.P. | Acc. | A.P. | Acc. | A.P. |
| CLIPDetection | 95.1 | 99.2 | 76.0 | 84.1 | 79.8 | 87.7 | 73.9 | 81.7 | 86.9 | 93.4 | 68.6 | 75.0 | 42.3 | 38.9 | 56.6 | 60.7 | 53.8 | 55.7 |
| FreqNet | 99.3 | 100.0 | 70.3 | 74.4 | 73.0 | 78.1 | 80.0 | 82.3 | 82.4 | 89.3 | 52.8 | 52.0 | 33.2 | 36.4 | 33.6 | 35.1 | 43.5 | 45.0 |
| NPR | 99.8 | 100.0 | 49.6 | 57.1 | 72.7 | 88.4 | 56.5 | 71.5 | 81.6 | 91.6 | 49.9 | 56.1 | 45.7 | 32.5 | 45.9 | 31.4 | 47.9 | 37.3 |
| DFFreq | 99.1 | 100.0 | 54.4 | 62.0 | 76.0 | 87.1 | 68.2 | 76.5 | 81.4 | 89.4 | 51.4 | 43.4 | 40.1 | 36.6 | 40.0 | 37.1 | 46.0 | 46.3 |
| SAFE | 99.9 | 100.0 | 90.8 | 95.7 | 91.4 | 97.3 | 96.2 | 99.7 | 96.0 | 99.6 | 70.5 | 86.3 | 45.6 | 41.5 | 44.7 | 41.1 | 49.1 | 38.0 |

| Test Dataset → | InSwap | | SimSwap | | FLUX1-dev | | Midjourney-V6 | | GLIDE | | DALLE-3 | | Imagen3 | | SD3 | | SDXL | |
|---|---|---|---|---|---|---|---|---|---|---|---|---|---|---|---|---|---|---|
| Detectors ↓ | Acc. | A.P. | Acc. | A.P. | Acc. | A.P. | Acc. | A.P. | Acc. | A.P. | Acc. | A.P. | Acc. | A.P. | Acc. | A.P. | Acc. | A.P. |
| CLIPDetection | 47.1 | 44.4 | 47.2 | 44.9 | 82.4 | 88.9 | 78.5 | 85.1 | 64.6 | 71.0 | 82.0 | 88.9 | 81.7 | 88.5 | 83.3 | 90.7 | 84.5 | 91.9 |
| FreqNet | 42.2 | 45.0 | 42.8 | 45.9 | 73.5 | 74.6 | 51.6 | 51.0 | 59.9 | 64.3 | 64.6 | 67.6 | 62.2 | 65.7 | 76.2 | 78.5 | 79.0 | 80.2 |
| NPR | 47.9 | 34.0 | 47.5 | 35.3 | 91.9 | 96.5 | 62.3 | 71.8 | 81.5 | 92.7 | 60.3 | 73.4 | 72.6 | 87.6 | 84.5 | 94.1 | 82.2 | 93.0 |
| DFFreq | 45.9 | 46.6 | 44.2 | 45.4 | 80.4 | 88.5 | 62.0 | 62.6 | 72.8 | 84.3 | 65.4 | 72.5 | 67.1 | 77.0 | 75.9 | 83.9 | 85.3 | 91.7 |
| SAFE | 49.4 | 45.4 | 48.7 | 45.0 | 95.9 | 99.5 | 88.9 | 96.9 | 90.5 | 96.9 | 46.2 | 42.4 | 94.2 | 98.1 | 93.3 | 97.3 | 96.9 | 98.9 |

| Test Dataset → | BLIP | | Infinite-ID | | InstantID | | IP-Adapter | | PhotoMaker | | SocialRF | | CommunityAI | | Mean | |
|---|---|---|---|---|---|---|---|---|---|---|---|---|---|---|---|---|
| Detectors ↓ | Acc. | A.P. | Acc. | A.P. | Acc. | A.P. | Acc. | A.P. | Acc. | A.P. | Acc. | A.P. | Acc. | A.P. | Acc. | A.P. |
| CLIPDetection | 78.7 | 87.7 | 78.4 | 86.7 | 84.3 | 91.0 | 76.0 | 82.8 | 62.4 | 69.4 | 63.4 | 68.9 | 57.8 | 58.2 | _71.4_ | _76.6_ |
| FreqNet | 95.4 | 99.7 | 66.2 | 70.0 | 69.2 | 75.7 | 76.0 | 79.2 | 72.2 | 79.1 | 52.9 | 53.8 | 53.7 | 69.7 | 64.2 | 67.7 |
| NPR | 98.7 | 100.0 | 58.7 | 75.9 | 62.1 | 78.0 | 81.6 | 92.2 | 49.4 | 52.5 | 57.4 | 65.4 | 52.9 | 56.0 | 65.6 | 70.6 |
| DFFreq | 96.2 | 99.3 | 56.2 | 63.8 | 85.7 | 91.3 | 79.6 | 88.3 | 70.4 | 78.5 | 57.5 | 62.0 | 54.8 | 65.0 | 66.2 | 71.2 |
| SAFE | 99.6 | 100.0 | 95.6 | 99.0 | 96.3 | 99.4 | 94.3 | 97.9 | 96.0 | 99.5 | 58.0 | 62.7 | 54.6 | 58.8 | **79.3** | **81.5** |

To further verify its effectiveness, Table 17 presents the F.Acc (Fake Accuracy) results of six representative detectors on both the curated dataset and the original unfiltered dataset. The main value in each cell corresponds to the detector's performance on the original unfiltered dataset, which indicates the absolute performance difference between the unfiltered and curated datasets. A positive subscript (e.g. +2.9) indicates that the detector performed worse on the curated dataset, implying that the curated dataset is more challenging. Conversely, a negative subscript (e.g. -1.0) suggests improved performance. Across most models and subsets, we observe a consistent decline in F.Acc following curation, particularly for state-of-the-art detectors such as AIDE and SAFE. These results underscore the increased challenge presented by the curated dataset and validate our filtering approach as a means of constructing a more robust benchmark.

Table 15: A detailed evaluation of the impact of Rotation & Color-Jitter & Mask data augmentation strategies on AI-generated image detectors under Setting-II, where the training dataset consists of 144K images generated by both SD-v1.4 and ProGAN.

| Test Dataset → | ProGAN | | R3GAN | | StyleGAN3 | | StyleGAN-XL | | StyleSwim | | WFIR | | BlendFace | | E4S | | FaceSwap | |
|---|---|---|---|---|---|---|---|---|---|---|---|---|---|---|---|---|---|---|
| Detectors ↓ | Acc. | A.P. | Acc. | A.P. | Acc. | A.P. | Acc. | A.P. | Acc. | A.P. | Acc. | A.P. | Acc. | A.P. | Acc. | A.P. | Acc. | A.P. |
| CLIPDetection | 93.5 | 98.9 | 72.9 | 80.2 | 75.6 | 83.7 | 69.5 | 76.7 | 85.2 | 91.4 | 67.0 | 73.0 | 42.2 | 39.6 | 55.2 | 58.7 | 52.7 | 54.7 |
| FreqNet | 98.9 | 100.0 | 72.6 | 73.7 | 71.7 | 72.3 | 78.1 | 79.8 | 78.3 | 86.0 | 52.4 | 50.1 | 26.1 | 34.7 | 25.6 | 33.8 | 39.7 | 41.0 |
| NPR | 99.9 | 100.0 | 55.4 | 86.6 | 81.3 | 96.6 | 70.8 | 94.9 | 85.7 | 97.7 | 51.4 | 88.9 | 48.4 | 40.7 | 48.6 | 36.0 | 49.8 | 51.8 |
| DFFreq | 99.2 | 100.0 | 55.9 | 63.9 | 72.1 | 83.8 | 68.8 | 76.7 | 85.2 | 92.3 | 53.5 | 46.7 | 39.3 | 35.6 | 39.6 | 35.9 | 46.1 | 44.4 |
| SAFE | 100.0 | 100.0 | 93.9 | 98.2 | 89.7 | 97.6 | 93.1 | 97.6 | 97.8 | 99.6 | 60.4 | 81.8 | 47.3 | 45.6 | 47.6 | 46.0 | 50.7 | 45.7 |

| Test Dataset → | InSwap | | SimSwap | | FLUX1-dev | | Midjourney-V6 | | GLIDE | | DALLE-3 | | Imagen3 | | SD3 | | SDXL | |
|---|---|---|---|---|---|---|---|---|---|---|---|---|---|---|---|---|---|---|
| Detectors ↓ | Acc. | A.P. | Acc. | A.P. | Acc. | A.P. | Acc. | A.P. | Acc. | A.P. | Acc. | A.P. | Acc. | A.P. | Acc. | A.P. | Acc. | A.P. |
| CLIPDetection | 47.2 | 45.7 | 47.4 | 46.1 | 82.9 | 90.4 | 80.9 | 88.3 | 64.3 | 70.3 | 83.0 | 90.4 | 82.4 | 89.9 | 74.5 | 72.5 | 83.9 | 91.9 |
| FreqNet | 36.5 | 40.2 | 38.2 | 42.4 | 71.1 | 65.7 | 50.5 | 48.6 | 51.9 | 52.0 | 65.1 | 65.1 | 65.6 | 62.3 | 74.5 | 72.5 | 77.6 | 76.3 |
| NPR | 50.0 | 46.1 | 49.3 | 47.0 | 90.9 | 98.6 | 73.6 | 89.3 | 83.4 | 97.2 | 49.2 | 48.5 | 90.5 | 98.7 | 82.2 | 97.1 | 58.5 | 84.1 |
| DFFreq | 45.6 | 44.6 | 44.0 | 44.4 | 83.1 | 90.4 | 64.2 | 67.8 | 72.2 | 82.9 | 71.7 | 80.8 | 67.9 | 77.5 | 78.4 | 87.3 | 89.9 | 95.6 |
| SAFE | 49.7 | 49.9 | 49.0 | 49.5 | 98.1 | 99.7 | 94.1 | 98.4 | 92.5 | 97.9 | 49.0 | 45.8 | 96.7 | 98.8 | 94.1 | 98.8 | 98.3 | 99.7 |

| Test Dataset → | BLIP | | Infinite-ID | | InstantID | | IP-Adapter | | PhotoMaker | | SocialRF | | CommunityAI | | Mean | |
|---|---|---|---|---|---|---|---|---|---|---|---|---|---|---|---|---|
| Detectors ↓ | Acc. | A.P. | Acc. | A.P. | Acc. | A.P. | Acc. | A.P. | Acc. | A.P. | Acc. | A.P. | Acc. | A.P. | Acc. | A.P. |
| CLIPDetection | 77.4 | 86.1 | 77.1 | 85.5 | 81.9 | 89.0 | 76.7 | 83.8 | 60.2 | 66.4 | 63.3 | 68.9 | 55.8 | 54.7 | 70.5 | 75.8 |
| FreqNet | 90.3 | 99.1 | 70.5 | 73.3 | 70.5 | 73.2 | 75.8 | 76.8 | 73.0 | 77.0 | 50.9 | 50.6 | 56.1 | 73.0 | 62.4 | 64.8 |
| NPR | 98.8 | 100.0 | 50.1 | 61.2 | 54.5 | 76.5 | 50.5 | 68.8 | 53.1 | 69.6 | 55.8 | 57.4 | 51.8 | 56.0 | 65.3 | 75.6 |
| DFFreq | 97.0 | 99.6 | 61.6 | 69.6 | 78.6 | 87.3 | 79.2 | 87.1 | 74.5 | 83.6 | 58.7 | 63.9 | 55.2 | 68.5 | 67.3 | 72.4 |
| SAFE | 99.7 | 100.0 | 96.9 | 99.2 | 98.2 | 99.6 | 92.8 | 98.1 | 97.0 | 99.3 | 58.0 | 64.2 | 54.2 | 55.2 | **79.9** | **82.6** |

Table 16: Evaluating the impact of different data pre-processing strategies on AI-generated image detectors under Training Setting-I, where the training dataset consists of 72k images generated by ProGAN. Note that Crop prep-rocessing primarily improves R.Acc. with limited or even negative impact on F.Acc. across several detectors.

| Detectors → | CLIPDetection | | FreqNet | | NPR | | DFFreq | | LaDeDa | | SAFE | |
|---|---|---|---|---|---|---|---|---|---|---|---|---|
| Process ↓ | R.Acc./F.Acc | Acc./A.P. | R.Acc./F.Acc | Acc./A.P. | R.Acc./F.Acc | Acc./A.P. | R.Acc./F.Acc | Acc./A.P. | R.Acc./F.Acc | Acc./A.P. | R.Acc./F.Acc | Acc./A.P. |
| Resize | 91.1/35.4 | 63.3/66.4 | 81.3/38.7 | 60.2/61.7 | 78.3/58.0 | 68.2/70.8 | 72.6/65.8 | 69.2/73.3 | 72.5/65.1 | 68.8/71.3 | 58.4/69.3 | 63.9/65.0 |
| Crop | 83.3/35.0 | 59.2/62.2 | 87.8/44.6 | 66.5/73.0 | 97.5/41.2 | 69.4/77.8 | 88.6/66.5 | 77.5/82.1 | 98.9/56.1 | 77.5/82.5 | 91.1/67.2 | 79.2/83.3 |

## A.6 Details of Generation Methods Used in Our Evaluation Datasets

We constructed this dataset from a comprehensive perspective, ensuring that each sub-dataset contains an equal number of real and fake images. The real images are sourced from FFHQ, CelebA-HQ, and Open Images V7. The first two datasets provide high-resolution facial images, while Open Images V7 offers a diverse set of categories—including Car, Taxi, Ambulance, and others—to enhance the variety of real-world content. We randomly selected and merged an equal number of images from

Table 17: **F.Acc results on the original unfiltered dataset (main score) and the curated dataset (subscript indicates the difference).** A positive subscript (e.g., +2.9) indicates that the detector performed worse on the curated dataset, implying that the curated dataset is more challenging.

| Detector | FaceSwap | IP-Adapter | Midjourney | SD3 | StyleSwim |
|---|---|---|---|---|---|
| CNNDetection | $1.8_{+0.4}$ | $5.0_{-1.0}$ | $6.1_{+0.3}$ | $16.2_{+2.9}$ | $8.7_{+1.8}$ |
| CLIPDetection | $29.8_{+2.5}$ | $94.3_{+2.3}$ | $78.5_{-2.1}$ | $92.1_{+1.5}$ | $97.9_{-0.2}$ |
| DFFreq | $0.9_{+0.6}$ | $76.8_{-1.3}$ | $55.2_{+1.2}$ | $75.7_{+2.3}$ | $87.9_{+7.1}$ |
| Ladeda | $0.7_{+0.7}$ | $92.6_{+2.0}$ | $84.6_{+1.2}$ | $98.5_{-0.5}$ | $96.3_{-1.0}$ |
| AIDE | $12.0_{-2.3}$ | $94.1_{+0.6}$ | $80.1_{+0.3}$ | $99.3_{+0.0}$ | $91.4_{+9.4}$ |
| SAFE | $22.9_{+19.6}$ | $99.3_{+9.5}$ | $98.5_{+1.3}$ | $86.3_{-5.4}$ | $99.9_{+0.6}$ |

these sources to form the real image set, matching the quantity of fake images. The fake images are generated using 25 forgery methods spanning various paradigms, including GANs, diffusion models, deepfakes, and customized generation techniques. Additionally, we collected real and manipulated images from real-world scenarios by crawling social media platforms and AI communities. All images were curated to maintain high clarity and quality. A detailed description of each sub-dataset and generation method is provided in the following sections.

### A.6.1 GANs for Noise-to-image Generation

Generative Adversarial Networks (GANs) are a class of deep learning models that synthesize realistic images from random noise through adversarial training between a generator and a discriminator. To construct the GAN-based subset of our dataset, we selected six state-of-the-art methods: ProGAN, StyleGAN3, StyleGAN-XL, StyleSwim, R3GAN, and WFIR. After generating synthetic images using these models, we applied a rigorous refinement pipeline to enhance both the diversity and fairness of the dataset for downstream forgery detection tasks. Specifically, we utilized the Contrastive Language–Image Pretraining (CLIP) model [61] to detect and filter out images with excessive similarity, thereby reducing redundancy. Additionally, we incorporated an aesthetic evaluation protocol [62] to assess and rank the visual quality of the generated images using quantitative metrics. This process enabled the exclusion of low-quality samples, ensuring a balanced and high-quality dataset.

**1) ProGAN** [36]: ProGAN introduces a progressive growing strategy, where training begins with low-resolution images and incrementally adds layers to both the generator and discriminator to gradually increase image resolution. This approach enhances training stability and results in high-quality image synthesis. Since ProGAN is also used in the training of certain detection models, we included it in our evaluation to ensure domain consistency. The corresponding data is sourced from the ForenSynths dataset [64], which contains 4,000 generated images and 4,000 real images.

**2) StyleGAN3** [37]: StyleGAN3 builds upon the GAN framework by enhancing the rendering of high-frequency details through the use of Fourier feature mapping, which transforms input coordinates into the frequency domain to mitigate aliasing artifacts. Furthermore, StyleGAN3 introduces rotation-invariant design principles, ensuring geometric consistency under spatial transformations and resulting in highly realistic and coherent image synthesis. For our dataset, we employed the official pre-trained model stylegan3-t-ffhq-1024x1024.pkl for image generation. After post-processing, the resulting subset consists of 4,500 generated images and 4,500 real images.

**3) StyleGAN-XL** [38]: StyleGAN-XL, an extension of the StyleGAN architecture, leverages a GANs framework for image synthesis. It utilizes a style-based synthesis network with Adaptive Instance Normalization (AdaIN), which maps latent codes through a multi-layer perceptron to control image features at multiple scales, from coarse structures to fine details. The model incorporates progressive growing and stochastic noise injection, enabling the generation of high-resolution images (up to 1024×1024) with diverse textures and intricate details, such as hair and skin patterns. For our dataset, we utilized the official pre-trained model ffhq1024.pkl for generation. After processing, the dataset includes 4,500 generated images and 4,500 real images.

**4) StyleSwim [39]**: StyleSwin utilizes a GANs framework with a Swin Transformer-based generator to synthesize high-resolution images, optimizing computational efficiency through local attention mechanisms. The model incorporates double attention by combining local and shifted window contexts, expanding the receptive field and enhancing image coherence. Additionally, a wavelet discriminator is employed to minimize blocking artifacts, ensuring high-fidelity details in images up to a 1024×1024 resolution. For our dataset, we used the official pre-trained model FFHQ_1024.pt and CelebAHQ_1024.pt. After processing, the dataset includes 4,500 generated images and 4,500 real images.

**5) R3GAN** [40]: R3GAN utilized a lightweight generator that synthesizes images from latent codes under the guidance of a regularized relativistic loss. This loss formulation promotes stable training and effectively mitigates mode collapse. The model incorporates modern convolutional architectures inspired by ConvNeXt, enabling efficient extraction of fine-grained image features and producing high-quality results without relying on ad hoc design tricks. Image generation is performed through a progressive architecture with stacked resolution stages, achieving strong fidelity and diversity on benchmarks such as FFHQ and ImageNet. For our dataset, we used the official pre-trained model

ffhq-256x256.pkl for generation. After post-processing, the dataset consists of 4,500 generated images and 4,500 real images.

**6) WFIR** [40]: The WhichFaceIsReal (WFIR) dataset is designed to evaluate the ability to distinguish real faces from AI-generated ones. It employs StyleGAN to synthesize high-resolution facial images from latent codes via a mapping network and synthesis layers incorporating Adaptive Instance Normalization (AdaIN). The model uses a progressive growing strategy to stabilize training, enabling the generation of highly realistic faces with diverse attributes such as facial expressions and lighting conditions. These synthetic images are paired with real faces sourced from public datasets, forming a binary classification task that challenges detection models to identify subtle artifacts indicative of forgery. The dataset used in this work is sourced from ForenSynths [64], comprising 1,000 generated images and 1,000 real images.

### A.6.2 Diffusion for Text-to-image Generation

Text-to-image diffusion models synthesize images by iteratively denoising random noise, guided by textual input. As such, generating high-quality textual descriptions is a prerequisite for constructing a high-quality dataset. To this end, we employed the official Gemini API to produce diverse text prompts. Specifically, we guided the model using carefully crafted instructions, such as: "To perfect the description of a person in photo-realistic style, I need one thousand different descriptions, each 20–25 words long. Each should describe a completely different scenario. For example: 'A woman wearing a red dress in the park, Disney cartoon style.'"

To ensure content diversity, we generated text prompts across four categories—people, animals, objects, and landscapes—each containing 1,500 unique sentences. Based on these prompts, we constructed a diffusion-generated dataset using seven state-of-the-art models: GLIDE, DALLE-3, Imagen3, FLUX1-dev, Midjourney V6, Stable Diffusion 3 (SD3), and Stable Diffusion XL (SDXL).

Following image generation, we applied a rigorous post-processing pipeline to refine the dataset, enhancing both its diversity and suitability for forgery detection tasks. The Contrastive Language–Image Pretraining (CLIP) model [61] was used to systematically detect and exclude visually redundant images. Additionally, an aesthetic evaluation protocol [62] was implemented to assess and rank image quality using quantitative metrics, ensuring the removal of samples with subpar visual attributes.

**1) GLIDE** [48]: GLIDE utilizes a diffusion-based generative framework that transforms random noise into coherent images through iterative denoising, guided by text prompts encoded via a transformer. It adopts classifier-free guidance to balance fidelity and diversity, enabling the generation of photorealistic images conditioned on joint image–text training data. Furthermore, GLIDE supports text-driven image inpainting through its fine-tuned diffusion decoder, allowing precise editing by filling in masked regions while preserving contextual integrity. For our dataset, we use the official pre-trained model. After data processing, the dataset comprises 4,500 generated images and 4,500 real images.

**2) DALLE-3** [69]: DALLE-3 adopts a text-conditioned diffusion model in which transformer-encoded text captions guide the iterative denoising of Gaussian noise into coherent, high-resolution images. The model benefits from enhanced caption quality and extensive synthetic training data, significantly improving the alignment between textual prompts and generated visuals. By employing classifier-free guidance, DALLE-3 effectively balances prompt fidelity with output diversity, enabling the generation of photorealistic and semantically rich images even for complex descriptions. For our dataset, We utilize the official OpenAI API for image generation. After data processing, the dataset includes 4,000 generated images and 4,000 real images.

**3) Imagen3** [70]: Imagen 3, developed by Google, utilizes a latent diffusion model framework that progressively denoises latent representations into high-resolution images, guided by text prompts encoded through a large language model. It achieves enhanced photorealism via multi-stage upsampling, enabling the synthesis of fine textures and accurate spatial structures at resolutions up to 1024×1024. Through advanced prompt comprehension and classifier-free guidance, Imagen 3 demonstrates strong alignment with complex textual inputs, producing visually rich and artifact-free outputs. For our dataset, We employ the official Google API for image generation. After data processing, the dataset comprises 4,500 generated images and 4,500 real images.

**4) FLUX1-dev** [71]: FLUX.1 adopts a hybrid diffusion-transformer architecture, combining iterative denoising with transformer layers to synthesize high-resolution images conditioned on text prompts.

The model integrates flow matching to stabilize the training process, significantly improving its ability to render fine-grained details such as textures and anatomically accurate human hands. To effectively handle complex prompts, FLUX.1 incorporates rotary positional embeddings and parallel attention layers, enabling coherent and diverse outputs at resolutions up to 1024×1024. For our dataset, We utilize the official pre-trained model available at black-forest-labs/FLUX.1-dev for image generation. After data processing, the resulting dataset consists of 4,500 generated images and 4,500 real images.

**5) Midjourney V6** [45]: MidJourney V6 is built upon an advanced diffusion model that iteratively refines random noise into high-resolution images, guided by text prompts processed through an enhanced language encoder for precise semantic alignment. The model integrates improved up-sampling techniques and sophisticated attention mechanisms to produce detailed textures, realistic lighting, and coherent scene composition at resolutions up to 2048×2048. Trained on a diverse corpus of image-text pairs, MidJourney V6 demonstrates strong prompt comprehension and generates photorealistic images with minimal visual artifacts. As MidJourney does not offer a public API, we manually submitted consistent prompt texts via the official website to ensure alignment with other generation methods. After data processing, the resulting dataset consists of 3,000 generated images and 3,000 real images.

**6) SD-3** [43]: a popular text-to-image model built upon the Multimodal Diffusion Transformer (MMDiT) architecture, integrating latent diffusion to substantially enhance image fidelity, prompt comprehension, typography rendering, and computational efficiency. It leverages multiple text encoders—OpenCLIP-ViT/G, CLIP-ViT/L (with a context length of 77 tokens), and T5-XXL (with context lengths of 77 or 256 tokens depending on the training phase)—to robustly capture semantic information. Additionally, SD-3 incorporates Query-Key (QK) normalization to improve training stability and image generation quality. For our dataset, we use the official pre-trained model available at stabilityai/stable-diffusion-3.5-large for generation. After data processing, the final dataset consists of 4,500 generated images and 4,500 real images.

**7) SD-XL** [42]: SD-XL [42], developed by Stability AI, is an advanced text-to-image model built upon the Stable Diffusion framework. Compared to its predecessors, SDXL incorporates a significantly larger UNet backbone and leverages two fixed pre-trained text encoders—OpenCLIP-ViT/G and CLIP-ViT/L—to enhance its capacity for generating high-fidelity images with fine-grained detail. The model adopts an expert ensemble latent diffusion pipeline: an initial base model generates noisy latent representations, which are subsequently refined using a dedicated denoising module (Refiner 1.0) to improve image quality. Constructing our dataset, we use the official pre-trained model stabilityai/stable-diffusion-xl-base-1.0 for image generation. After processing, the final dataset comprises 4,500 generated images and 4,500 real images.

### A.6.3   GANs for Deepfake

The DeepFake Detection Challenge (DFDC) dataset [72], developed by Facebook AI, contains over 100,000 manipulated video clips generated from 3,426 consenting actors using a variety of techniques, including GAN-based face-swapping and non-learning-based manipulation algorithms. These approaches incorporate autoencoder-driven facial replacement and lip-syncing modifications, blending source and target identities to produce highly realistic deepfakes with diverse visual and contextual characteristics. The dataset generation process ensures variation in gender, skin tone, and background, closely simulating real-world scenarios and providing a significant challenge for detection models. For our dataset construction, we randomly sampled a diverse subset of frames from the DFDC dataset to ensure broad visual representation. Additionally, we also applied a rigorous refinement pipeline to enhance both the diversity and fairness of the dataset for downstream forgery detection tasks. Specifically, we utilized the Contrastive Language–Image Pretraining (CLIP) model [61] to detect and filter out images with excessive similarity, thereby reducing redundancy. Additionally, we incorporated an aesthetic evaluation protocol [62] to assess and rank the visual quality of the generated images using quantitative metrics. This process enabled the exclusion of low-quality samples, ensuring a balanced and high-quality dataset.

**1) BlendFace** [49]: BlendFace employs a GANs framework for face-swapping, leveraging a novel identity encoder that extracts disentangled identity features from source images. This encoder processes blended inputs to reduce attribute bias—such as hairstyle—ensuring that only identity-relevant features are transferred while preserving target-specific attributes. These identity features are integrated with target image attributes via a feature fusion module within the generator, enabling

seamless and realistic face swaps. A discriminator, guided by an identity-preserving loss, evaluates the visual fidelity and identity consistency of the generated outputs. After data processing, the resulting dataset comprises 4,500 generated images and 4,500 real images.

**2) E4S** [50]: E4S adopts a GANs framework that operates in the latent space of a pre-trained StyleGAN to achieve fine-grained face swapping by explicitly disentangling facial shape and texture. A multi-scale, mask-guided encoder projects the texture of each facial region into regional style codes, which are injected into feature maps via a mask-guided injection module. This design reduces the face-swapping process to style and mask swapping, enabling accurate identity transfer from the source while preserving target attributes such as pose and expression. To further enhance realism, a face re-coloring network adjusts lighting conditions, and an inpainting network resolves shape mismatches, ensuring seamless and high-fidelity face swaps. After data processing, the resulting dataset consists of 4,500 generated images and 4,500 real images.

**3) FaceSwap** [51]: FaceSwap adopts a traditional computer vision pipeline that combines 3D facial modeling with image blending techniques to perform realistic face replacement without the use of deep learning. The process begins by detecting 68 facial landmarks in both the source and target images using Dlib. These landmarks are then used to fit a standard 3D face model (Candide) by optimizing parameters such as pose, facial shape, and expression. After alignment, the source face texture is projected onto the 3D model and rendered into the target image. The final result is produced by blending the rendered face with the target using feathering and color correction, ensuring smooth transitions and a natural appearance. After data processing, the resulting dataset includes 4,500 generated images and 4,500 real images.

**4) InSwap** [52]: InSwap utilizes a deep learning-based face swapping pipeline that combines identity embedding extraction with generative synthesis to produce realistic and identity-preserving face replacements. The process begins by detecting and aligning faces in both the source and target images. A 512-dimensional identity embedding is then extracted from the source face using the ArcFace model. This embedding, along with the aligned target face, is passed to an ONNX-based generative network, which synthesizes a new face that retains the target's pose and expression while adopting the source's identity. The resulting face is seamlessly integrated into the original target image using a facial mask for smooth blending. After data processing, the final dataset consists of 4,500 generated images and 4,500 real images.

**5) SimSwap** [53]: SimSwap adopts a GANs framework to achieve high-fidelity face swapping by transferring the identity of a source face onto a target face while preserving critical attributes such as expression and gaze. The core component, the Identity Injection Module (IIM), embeds the source identity into the target's feature representations, enabling arbitrary identity swapping without imposing decoder-specific constraints. To ensure the preservation of target facial attributes, a Weak Feature Matching Loss is employed, guiding the generator by comparing intermediate features from the discriminator. The overall encoder–decoder architecture processes both source and target images to synthesize seamless and realistic face swaps with enhanced attribute consistency. After data processing, the dataset comprises 4,500 generated images and 4,500 real images.

### A.6.4 Diffusion for Personalized Generation

Personalized generation represents a cutting-edge image synthesis approach that leverages diffusion models to generate highly customized images based on user-specific inputs. Since these personalized generations require only face-related prompt words, we utilized the official API of Gemini to generate 6,000 descriptive sentences, following the same prompt formulation method as previously described. We then selected five state-of-the-art diffusion-based personalization methods—BLIP, IP-Adapter, Infinite-ID, InstantID, and PhotoMaker—to construct a synthetic dataset for the Diffusion category.

To ensure the quality, diversity, and fairness of the resulting dataset for subsequent forgery detection tasks, we applied a multi-step refinement process. First, the CLIP model [61] was employed to measure semantic similarity across image–text pairs, allowing us to systematically identify and remove images with excessive redundancy. In parallel, we implemented an aesthetic evaluation protocol [62] to quantitatively assess and rank the visual quality of the generated images. Images deemed to have low aesthetic appeal were filtered out based on objective scoring metrics, resulting in a curated dataset with high visual fidelity and meaningful diversity.

**1) BLIP** [55]: BLIP utilizes a vision-language pre-trained transformer to synthesize images by mapping text prompts to visual features through a generative decoder fine-tuned on large-scale image–text datasets. It incorporates both contrastive learning and image captioning tasks to effectively align textual and visual modalities, enabling the generation of semantically rich and visually coherent images. The model employs an iterative refinement strategy inspired by diffusion processes to progressively enhance image quality and semantic fidelity. For our dataset, we use the official pre-trained model from salesforce LAVIS repository to generate various imgaes. After data processing, the resulting dataset consists of 4,500 generated images and 4,500 real images.

**2) IP-Adapter** [56]: IP-Adapter introduces a decoupled cross-attention mechanism into a pre-trained diffusion model (*e.g.*, Stable Diffusion) to enable image generation conditioned on both visual and textual inputs. It incorporates a lightweight adapter module to encode image features from reference images, effectively capturing fine-grained visual details while maintaining strong alignment with text prompts. This tuning-free framework supports flexible and high-quality image synthesis that preserves the identity or style of the reference image alongside the semantic intent of the text. For our dataset, we use the official pre-trained model from Tencent AI Lab's IP-Adapter repository for image generation. After data processing, the final dataset comprises 4,500 generated images and 4,500 real images.

**3) Infinite-ID** [8]: Infinite-ID utilizes a diffusion model framework augmented with an identity-enhanced training strategy. It incorporates an additional image cross-attention module to extract detailed identity features from a single reference image, while deactivating the text cross-attention to minimize potential interference. The model introduces a mixed attention mechanism and an AdaIN-mean operation to effectively blend identity and semantic information, ensuring high-fidelity image generation. This approach facilitates identity-preserved personalization, allowing for the generation of images that retain facial characteristics while aligning with text prompts across various styles. For our dataset, we use the official pre-trained model from Infinite-ID for image generation. After data processing, the dataset contains 4,500 generated images and 4,500 real images.

**4) InstantID** [54]: InstantID adopts a diffusion model framework to generate identity-preserving images from a single reference facial image, leveraging a lightweight adapter for compatibility with pre-trained models such as Stable Diffusion XL. It incorporates IdentityNet, which encodes detailed facial features using strong semantic and weak spatial conditions, and employs a decoupled cross-attention module to guide generation based on facial landmarks and textual prompts. This tuning-free approach enables rapid personalization and high-fidelity image synthesis across diverse styles without requiring extensive fine-tuning. For our dataset, we use the official pre-trained model from instantX-research/InstantID for generation. After data processing, the dataset comprises 4,500 generated images and 4,500 real images.

**5) PhotoMaker** [16]: PhotoMaker [16] adopts a latent diffusion model framework that encodes reference images into stacked identity embeddings using a CLIP-based image encoder to preserve facial characteristics for personalized image synthesis. These embeddings are injected into the Stable Diffusion XL (SDXL) pipeline through cross-attention layers, enabling high-fidelity image generation without the need for additional training. The model supports fast customization by integrating identity features with text prompts, producing photorealistic images at resolutions up to 1024×1024. We utilize the official pre-trained model from TencentARC/PhotoMaker for generation. After data processing, the dataset comprises 4,500 generated images and 4,500 real images.

### A.6.5   Open-source Platforms

Advanced generative techniques present significant challenges for forgery detection, particularly in real-world scenarios where variability and complexity often exceed the constraints of controlled laboratory settings. To improve the robustness and practical relevance of our detection methods, we construct a dataset by crawling images from online platforms, aiming to better reflect the diversity encountered in real social contexts. A multi-step refinement process is applied to ensure both complexity and fairness in the subsequent detection task. Specifically, the Contrastive Language-Image Pretraining (CLIP) model [61] is employed to systematically identify and filter out images exhibiting excessive similarity, thereby promoting diversity within the dataset. In parallel, we implement an aesthetic evaluation protocol [62] to quantitatively assess image quality and discard samples with suboptimal visual attributes.

To build a robust social media dataset (SocialRF), we utilize targeted web crawling with specific hashtags. Authentic images are collected using tags such as #nature, #photography, and #realphoto, while AI-generated images are sourced using tags like #aiart, #aigenerated, and #fakephoto. Recognizing the high quality of synthetic content in AI art communities, we further augment the dataset with visually compelling AI-generated images from prominent platforms including ArtStation, Civitai, and Liblib, forming what we refer to as the CommunityAI dataset. After processing, each dataset contains 4,500 generated images and 4,500 real images.

### A.7    Details of Detection Models Evaluated in Our AIGIBench

To comprehensively evaluate our dataset and the proposed evaluation criteria, we selected several state-of-the-art detection methods from recent years. The following sections provide detailed descriptions of these detection approaches[3]:

**1) Resnet-50** [65]: ResNet utilizes a deep convolutional neural network (CNN) architecture with residual connections to detect AI-generated images by learning subtle spatial artifacts, such as unnatural textures and blending inconsistencies, from paired real and synthetic datasets. Its design incorporates stacked residual blocks with shortcut connections, which alleviate vanishing gradient issues and facilitate the extraction of fine-grained features, including generative noise patterns. Trained end-to-end as a binary classifier, ResNet distinguishes real from fake images by leveraging multi-scale feature representations. Robustness across diverse AIGC methods, including GANs and diffusion models, is further enhanced through extensive data augmentation.

**2) CNNDetection** [64]: CNNDetection leverages a convolutional neural networks (CNNs) to identify AI-generated images by analyzing spatial artifacts commonly found in synthetic outputs, such as high-frequency inconsistencies and unnatural pixel correlations. The network extracts hierarchical features directly from raw pixel data using stacked convolutional layers, effectively capturing generative anomalies like aliasing patterns and irregular textures. A feature aggregation mechanism further enhances detection performance by combining multi-scale representations, amplifying subtle artifacts that are characteristic of AI-generated content.

**3) Gram-Net** [66]: Gram-Net is designed to detect GAN-generated images by focusing on texture inconsistencies that are often overlooked by conventional detectors. Instead of relying on local pixel-level cues, Gram-Net incorporates Gram matrix blocks into a CNNs to learn global texture statistics. These Gram-based features are more robust to common post-processing operations such as downsampling, JPEG compression, blurring, and noise. Moreover, Gram-Net demonstrates strong generalization across unseen GAN architectures and datasets, making it highly effective for in-the-wild fake face detection.

**4) LGrad** [67]: LGrad (Learning on Gradients) introduces a novel approach to fake image forensics by leveraging gradient-based representations. Each input image is first passed through a fixed, pretrained CNN (**e.g.**, a GAN discriminator), and the sum of its activations is back-propagated to produce an image-sized gradient map. These sparse, content-agnostic gradient maps effectively highlight generation artifacts. By using these gradients as a universal representation, a simple binary classifier trained on one GAN's outputs can generalize well to different categories and previously unseen generators. This method shifts the detection challenge from data dependency to transformation-model dependency, enabling improved generalization in open-world scenarios.

**5) CLIPDetection** [19]: Traditional real/fake classifiers trained on a single generative model often overfit to low-level, model-specific artifacts, leading them to misclassify any image lacking these fingerprints—including those generated by unseen models—as real. To address this limitation, CLIPDetection proposes conducting real/fake classification in a feature space not explicitly optimized for detection. Specifically, they leverage fixed embeddings from a large pre-trained CLIP-ViT vision-language model. By applying simple techniques such as nearest-neighbor lookup and linear probing on a feature library built from real and fake images of a known model, their method avoids overfitting to model-specific cues and achieves strong generalization across different types of generators, including GANs, diffusion models, and autoregressive frameworks.

**6) FreqNet** [57]: FreqNet encourages deepfake detectors to "think in frequencies" by explicitly modeling and leveraging frequency-domain information. It begins by isolating high-frequency

---

[3]Code and pre-trained checkpoint are publicly available at: https://github.com/HorizonTEL/AIGIBench

components of each image using an FFT-based high-pass filter, which are then input to a lightweight CNN. To enhance frequency-space reasoning, FreqNet introduces a plug-in frequency-domain learning block that transforms intermediate feature maps via FFT, applies learnable magnitude and phase transformations, and then performs an inverse FFT (iFFT), enabling optimization directly in the frequency domain. Additionally, a high-frequency preserving loss concentrates the energy of hidden features within the high-frequency bands, guiding the network to learn source-independent artifacts rather than model-specific fingerprints, thus improving generalization across generative methods.

**7) NPR** [23]: NPR targets the universal structural artifacts introduced by up-sampling layers in generative models. Instead of using raw RGB values, the method transforms each input image into NPR maps—channel-wise grids that capture signed intensity differences between each pixel and its four immediate neighbors. These maps make local pixel-dependency patterns explicit, revealing artifacts characteristic of synthetic up-sampling operations. A compact CNN is then trained on these NPR maps to learn a decision boundary that distinguishes real from generated images. By focusing on structural dependencies rather than texture or color, the approach achieves strong cross-model generalization.

**8) LaDeDa** [18]: LaDeDa is a patch-level deepfake detector that partitions each input image into $9 \times 9$ pixel patches and processes them using a BagNet-style ResNet-50 variant with its receptive field constrained to the same $9 \times 9$ region. The model assigns a deepfake likelihood to each patch, and the final prediction is obtained by globally pooling the patch-level scores. The network is trained end-to-end using binary cross-entropy loss on image-level labels, enabling it to learn localized generative artifacts indicative of fake content.

**9) DFFreq** [68]: DFFreq introduces a dual-branch architecture that captures both local and global frequency-based features. The Local Spatial-Frequency Branch (LoSFB) first applies a single-level DWT to the input image, decomposing it into four sub-bands. These are then tiled into fixed-size sliding windows and processed through a Window-Attention block, where localized self-attention and an in-window MLP extract fine-grained spatial-frequency patterns. In parallel, the Global Frequency Branch (GloFB) performs a FFT, retains only the phase spectrum (discarding amplitude), and feeds the resulting phase maps through the same Window-Attention mechanism to capture global frequency cues. The phase and amplitude components are then recombined via an inverse FFT to enhance representation fidelity.

**10) AIDE** [20]: AIDE formulates detection as a hybrid-feature learning problem, integrating both semantic and low-level artifact cues. It employs two expert branches: i) a Semantic Feature Extractor, which utilizes CLIP-ConvNeXt embeddings to detect high-level content inconsistencies, and ii) a Patchwise Feature Extractor, which ranks image patches by spectral energy, selects the highest- and lowest-frequency regions, and applies a lightweight CNN to capture fine-grained noise and artifact patterns. A gating mechanism dynamically fuses the outputs from both branches, enabling the detector to adaptively prioritize either semantic or low-level signals based on the characteristics of each image.

**11) SAFE** [58]: SAFE addresses two key training-stage limitations in deepfake detection: i) the weakening of forensic artifacts due to aggressive downsampling, and ii) overfitting to superficial color or semantic cues. Rather than modifying the model architecture, SAFE redesigns the input pre-processing pipeline. It replaces conventional resizing with random cropping to better preserve high-frequency details, applies data augmentations such as Color-Jitter and RandomRotation to break correlations tied to color and layout, and introduces patch-level random masking to encourage the model to focus on localized regions where synthetic pixel correlations typically emerge.

