# OpenReview forum: "Is Artificial Intelligence Generated Image Detection a Solved Problem?"
_NeurIPS.cc/2025/Datasets_and_Benchmarks_Track — NeurIPS 2025 Datasets and Benchmarks Track poster_

### Official Review · Reviewer_PjrR · 2025-06-16

**Rating:** 6
**Confidence:** 5

**Summary:**

The paper challenges the assumption that AI-generated image (AIGI) detection is a solved problem, despite existing methods reporting >95% accuracy in controlled settings. To address this, the authors introduce AIGIBench, a comprehensive benchmark designed to evaluate the real-world robustness of AIGI detectors through four core tasks: Generalization (unseen generative sources), Robustness (image degradations like compression/noise), Sensitivity to data augmentation, Impact of test-time preprocessing (e.g., cropping vs. resizing).

AIGIBench includes 23 diverse test subsets spanning GANs, diffusion models, deepfakes, personalized generation, and real-world samples from social media/AI art platforms. Experiments on 11 state-of-the-art detectors reveal severe performance drops in real-world scenarios. Key findings: Detectors fail on deepfakes and in-the-wild content (e.g., near 0% fake accuracy under JPEG compression). Data augmentation offers minimal benefits and may harm fake detection. Cropping improves real-image detection but not fake-image detection due to modality gaps.

The work concludes that AIGI detection remains an open challenge and advocates for more robust, generalizable methods.

**Dataset Code Accessibility:**

Yes

**Ethical Considerations:**

No, there are no or only very minor ethics concerns

**Final Justification:**

This work is intersting and solid, which can contribute to the community.
The rebuttal has solved all my concerns.
To avoid any misunderstanding, I will reiterate that the rating is **6 (Strong Accept)**.

**Limitations Weaknesses:**

1. Degradations tested (JPEG, noise, resampling) are common but not exhaustive (e.g., blur, adversarial attacks).

2. No study on compound corruptions (e.g., social media images with compression + filters).

3. Computational costs: Training/inference times and hardware details are omitted.

4. Detectors are evaluated only on AIGIBench subsets. Testing on existing benchmarks (e.g., GenImage) would strengthen claims.

**Strengths Contributions:**

1. Novel Benchmark Design.
  - Real-world relevance: The benchmark integrates samples from social media and AI platforms (e.g., Twitter, ArtStation), addressing distribution shifts often neglected by prior benchmarks.
  - Holistic evaluation: The proposed four-task framework—covering generalization, robustness, data augmentation, and preprocessing—captures key challenges in practical deployment.
  - Scale and diversity: It encompasses over 25 generative models (e.g., StyleGAN-XL, SD-3, DALL·E-3) and 11 detection methods, including state-of-the-art approaches from 2024–2025.

2. Rigorous Methodology.
  - Balanced dataset curation: The benchmark employs CLIP-based deduplication, aesthetic scoring for quality filtering, and manual screening to ensure content realism.
  - Insightful evaluation metrics: It decomposes accuracy into real-image detection (R.Acc) and fake-image detection (F.Acc), exposing performance asymmetries—such as high R.Acc but low F.Acc under perturbations.
  - Comprehensive training settings: Both single-model (e.g., ProGAN) and multi-model (e.g., ProGAN + SD-v1.4) training scenarios are evaluated, revealing trade-offs (e.g., improved R.Acc at the expense of F.Acc).

3. Valuable Findings.
  - Identification of critical limitations: Even leading detectors (e.g., SAFE) fail on deepfakes (e.g., FaceSwap) and user-generated content from social media.
  - Promising directions: Frequency-based (e.g., FreqNet, DFFreq) and CLIP-based detectors demonstrate superior robustness under distribution shift.
  - Modality gap insights: Simple preprocessing such as cropping benefits real images (structured signals) but not synthetic ones, which display high artifact variability.

4. Reproducibility and Ethics.
  - The authors provide public access to all code, data, and model checkpoints via Github.
  - Broader impacts, including potential misuse risks and benchmark limitations, are discussed with transparency.

---

> ### Author Rebuttal · Authors · 2025-07-27
>
> **Q1&Q2. More degradations testing and compound degradations.**
>
> We appreciate the reviewer’s insightful feedback regarding the evaluation setup. In response, we have expanded our evaluation in the revised manuscript to include additional degradation types commonly encountered in real-world scenarios. Specifically, we consider: (1) JPEG compression with a quality factor of 90, (2) Gaussian noise with a moderate intensity (σ = 1), and (3)  A compound degradation combining JPEG-95 compression and Gaussian noise (σ = 1).
>
> These additions allow for a more practical assessment of robustness under conditions likely to occur in real-world image sharing or editing pipelines. The updated results are summarized in Table I below, where all detectors are trained under Setting-II and evaluated across 25 diverse test datasets.The metrics reported include Accuracy (Acc.), Average Precision (A.P.), Real Accuracy (R.Acc.), and Fake Accuracy (F.Acc.). Key observations include:
> (1) CLIPDetection consistently outperforms other detectors across all perturbation settings, demonstrating strong robustness in both clean and noisy conditions.
> (2)CNNDetection, DFFreq, Ladeda, AIDE, and SAFE suffer significant performance degradation, particularly in terms of Fake Accuracy, indicating a high false negative rate under perturbations.
> (3) The composite perturbation setting (JPEG-95 + noise) further amplifies these challenges, exposing the vulnerability of frequency- and CNN-based detectors.
> (4) Interestingly, AIDE and SAFE maintain near-perfect Real Accuracy, but this comes at the cost of very poor Fake Accuracy, revealing a strong bias toward classifying images as real under noise.
>
> These results reinforce the importance of evaluating detectors under more moderate yet realistic perturbation levels and highlight the varying robustness characteristics of different detection paradigms.
>
> **Table I. The overall robust performance of AI-generated image detectors, where the training dataset follows Setting-II: training on 144K images generated by both SD-v1.4 and ProGAN. Notably, all reported results represent average values computed across 25 diverse test datasets.**
> | Detector      |       JPEG-90 （Acc./A.P./R.Acc./F.Acc.）      |  Noise -$\sigma=1$   | Noise -$\sigma=1$ & JPEG-95 |
> | :------------ | :-----------------: | :------------------: | :-----------------------: |
> | CNNDetection  | 54.6/66.3/97.9/11.4 | 53.9/57.4/96.1/11.7  |    53.8/57.7/96.4/11.1    |
> | CLIPDetection | 69.7/73.2/74.5/64.8 | 71.9/74.4/75.3/68.4  |    72.0/75.6/76.3/67.6    |
> | DFFreq        | 51.1/54.9/99.5/1.8  | 64.9/73.3/80.8/48.6  |    52.3/63.5/99.1/4.7     |
> | Ladeda        | 50.1/57.0/99.9/0.3  | 64.2/74.2/83.8/44.6  |    50.5/62.8/99.6/1.2     |
> | AIDE          | 50.3/57.9/99.6/1.0  | 77.9/73.7/65.3/90.5  |    50.0/53.0/99.7/0.3     |
> | SAFE          | 50.0/50.8/100.0/0.0 | 71.34/82.1/89.0/53.7 |    50.0/49.9/100.0/0.0    |
>
> **Q3. Computational costs.**
>
> We appreciate the reviewer’s suggestion. To improve transparency and reproducibility, we have added detailed information regarding the computational setup in the revised manuscript. Specifically, all experiments were conducted on NVIDIA A6000 40GB GPUs. Table II (see below) summarizes the training parameters (i.e., number of trainable parameters), inference time, and detection performance (mean R.Acc., F.Acc., Acc., and A.P.) for some key evaluated AIGI detectors under Setting-II—i.e., training on 144K images generated by both SD-v1.4 and ProGAN. We will further supplement the complete computational cost information for all models in the revised version.
>
> The results highlight the trade-offs between model complexity and performance. For instance, CLIPDetection has the largest parameter size (427.62M) and the longest inference time (51 minutes), yet only achieves moderate accuracy (72.5%) and precision (75.6%). In contrast, SAFE, with just 1.44M parameters and a 23-minute inference time, achieves the highest overall accuracy (79.9%) and competitive average precision (82.6%), demonstrating strong efficiency-performance balance. CNNDetection is the most lightweight (1.27M parameters) but suffers from low fake accuracy (11.6%) and overall accuracy (54.9%).
>
> **Table II. The mean results for different AIGI detectors, where the training dataset settings is Setting-II: Training on 144K images generated by both SD-v1.4 and ProGAN.**
> | Detector      | Training Parameters | Inference Time | R.Acc. Mean | F.Acc. Mean| Acc. Mean| A.P. Mean|
> | ------------- | :-----------------: | :------------: | :---------: | :-----------------: | :------------: | :---------: |
> | CNNDetection  |        1.27M        |     16min      |   98.2| 11.6| 54.9| 67.0|
> | CLIPDetection |       427.62M       |     51min      | 73.3| 71.5| 72.5| 75.6|
> | DFFreq        |        1.34M        |     17min      |    89.6| 51.9| 71.1| 75.7|
> | Ladeda        |       13.60M        |     27min      |    91.7| 54.9| 73.4| 79.3|
> | AIDE          |       54.43M        |    2h31min     |  88.1| 67.0| 77.6| 82.7|
> | SAFE          |        1.44M        |     23min      |    96.8 |63.0| 79.9| 82.6|
>
> **Q4. Testing on existing benchmarks.**
>
> Thank you for your valuable feedback. One of the key contributions of our work is the construction of a new evaluation dataset specifically designed to reflect real-world deployment scenarios. Compared to existing benchmarks such as GenImage, our dataset—AIGIBench—offers several notable advantages:
> (1) AIGIBench comprises 23 diverse subsets of fake images, spanning classic GANs, modern diffusion models, DeepFake techniques, and personalized generation methods (e.g., InstantID, PhotoMaker, IP-Adapter). Notably, over half of these models were released in 2024 or later, ensuring that the dataset captures the latest developments in generative technology.
> (2) In addition to synthesized images, AIGIBench incorporates real-world fake content from a wide range of platforms, including ArtStation, Liblib, and social media sources such as X (formerly Twitter) and Reddit. This broader sourcing leads to greater diversity and realism, better simulating the complexities of AI-generated content encountered in practice.
>
> To directly address the reviewer’s suggestion, we have further evaluated several representative detectors on the GenImage dataset. As shown in Table III, detectors consistently achieve higher performance on GenImage than on AIGIBench. This performance gap underscores the increased difficulty and real-world relevance of our benchmark, and further highlights its value in advancing the development and evaluation of robust detection methods.
>
> **Table III. The generalization results (ACC. and A.P.) for different AIGI detectors on GenImage dataset, where the training dataset settings is Setting-II: Training on 144K images generated by both SD-v1.4 and ProGAN.**
> |    Detector | Acc. Mean | A.P. Mean | R.Acc Mean | F.Acc Mean |
> | :----------- | :--: | :--: | :---: | :---: |
> | CNNDetection  | 61.0 | 83.3 | 99.6  | 22.3  |
> | CLIPDetection | 83.1 | 92.4 | 91.9  | 74.3  |
> |    DFFreq     | 87.2 | 96.4 | 98.0  | 76.0  |
> |    Ladeda     | 79.7 | 94.5 | 99.8  | 59.7  |
> |     AIDE      | 91.3 | 96.3 | 92.6  | 89.9  |
> |     SAFE      | 91.4 | 97.8 | 99.2  | 83.6  |

---

> > ### Comment · Reviewer_PjrR · 2025-08-01
> > **The rebuttal has solved all my concerns**
> >
> > Once again, I apologize to the author for the incorrect rating.
> > I am glad that the author was able to patiently and clearly address my concerns despite the emotional impact this situation may have had.
> > Based on the author's detailed response, **I have decided to raise my rating from 5 to 6.**

---

> > > ### Comment · Area_Chair_QmuH · 2025-08-01
> > >
> > > Got it. I will deal with the problem.
> > >
> > > Best,
> > > AC

---

> > > > ### Comment · Reviewer_PjrR · 2025-08-01
> > > >
> > > > Thanks for your help！

---

> > > ### Author Response · Authors · 2025-08-01
> > > **Thanks for your feedback**
> > >
> > > Dear Reviewer PjrR:
> > >
> > > We sincerely thank the reviewer for the thoughtful and constructive feedback, as well as for the time and effort dedicated to thoroughly engaging with our work. We are particularly grateful for your recognition of the revisions we made and your considerate acknowledgment of the emotional labor that can accompany the review process. We deeply appreciate your fair reassessment and the subsequent increase in rating. Your recognition of our contributions is both encouraging and motivating.

---

> ### Comment · Reviewer_PjrR · 2025-08-01
> **Explanation of the incorrect rating**
>
> Dear PC, SAC, AC, other reviewers, and authors,
>
> I sincerely apologize for the misunderstanding.
> The rating of 2 for this work was a mistake.
> I originally gave it a rating of **5 (Accept)** not 2.
> When reviewing the **Revisions**, I noticed that it was still 5 on June 22, but it changed to 2 on July 20.
> On July 20, the review had already concluded, so why did the rating change to 2?
> I am not concerned about the reason why it is.
> However, I feel compelled to correct this outcome, even though it was not caused by me.
> Especially for the authors, this was a negative experience, and I would like to express my apologies to them.

---

### Official Review · Reviewer_nDdE · 2025-07-01

**Ethics Flags:** Safety and security
**Rating:** 5
**Confidence:** 4

**Summary:**

The authors introduce a new benchmark fro evaluating detectors of AI-generated images. This benchmark contatins 23 syntehtic images subsets including GANs, diffusion models, deepfake face swaps and in the wild web collection of AI art and social media fakes.
The authors evaluate the detectors on multi-source generalization, robustness to JPEG/noise/up sampling degradation , sensitivity to common data-augmentations and impact of test-time preprocessing. They have provided eleven detectors (CNNS, CLIP-based, frequency domain and etc)  retained under two training settings(Progan only or ProGAN+ SD v1.4) and benchmarked on all tasks.
Main findings. Despite >95 % accuracy claimed in controlled papers, every detector’s fake-image accuracy (F.Acc.) collapses on real-world data or mild perturbations (often <5 %) while real-image accuracy stays high
. Simple crops boost R.Acc. but rarely improve F.Acc.
. Overall, the paper argues that AIGI detection is far from “solved”.

**Dataset Code Accessibility:**

Yes

**Ethical Comments:**

Adversaries might exploit benchmark gaps (e.g., detectors’ fragility to JPEG) to train more evasive generators.

**Ethical Considerations:**

No, there are no or only very minor ethics concerns

**Final Justification:**

After reading the rebuttal and updated experiments, I am maintaining my recommendation to accept. The authors have responded thoroughly to the main concerns raised in my initial review.

Training scope: The authors clarified their rationale for the ProGAN + SD-v1.4 setup as a representative generalization test, and further added experiments training on all subsets. These additional results substantiate the claim that broader training coverage can sometimes reduce generalization, and the analysis provides valuable nuance.

Purely diagnostic contribution: While no new detectors are proposed, the authors convincingly argue that diagnostic benchmarking is a necessary step, and they highlight actionable insights (augmentation sensitivity, architectural traits, etc.) that can guide future method development. This justification aligns with the benchmark’s role as an infrastructure paper.

Metrics: The expanded evaluation (AUC, EER, TPR at 10%/1% FPR) resolves my concern about limited metrics and strengthens the empirical rigor of the study.

Ethical and bias analysis: The authors acknowledged the limitation and committed to releasing demographic annotations and subgroup analysis in the revised version. While a full audit is still pending, I am satisfied with the direction and transparency shown.

Overall weighting:

All major weaknesses from my review have been addressed, either with new experiments, expanded analyses, or clear justification.

The benchmark remains highly timely and impactful for the community, offering breadth, rigor, and actionable insights into detector robustness.

The lack of methodological novelty is expected given the benchmark nature of the work and does not diminish its value.

Conclusion: With the rebuttal improvements, the paper makes a strong and well-justified contribution as a benchmark paper. I recommend accept.

**Limitations Weaknesses:**

1. Narrow Training Scope
   Only ProGAN and SD-v1.4 are used for detector training (Sec. 2.1), omitting newer generators such as SD-XL, DALLE-3, Midjourney-V6, InstantID, etc.

2. Purely Diagnostic—No Remedies Proposed
   The paper documents detector failures but does not introduce any new training strategies or model improvements to address them.

3. Limited Metric Suite
   Evaluation relies on R.Acc., F.Acc., and A.P.; calibration-aware or balanced metrics (e.g., AUROC, EER, Brier score) are not reported.

4. Sparse Ethical & Bias Analysis
   Board-impact discussion is brief; there is no demographic bias audit of the scraped real/fake face images or a detailed risk-mitigation plan.

**Strengths Contributions:**

1-Timely & Relevant. The benchmark meets an urgent need for realistic AIGI evaluation given the 2024–25 diffusion boom.
2-Breadth & Diversity. No existing benchmark covers such a wide generator spectrum or social-media fakes (Table 1 vs. others).
3-Clear, multi-facet task design. The four tasks map cleanly to real deployment pain-points (unknown models, compression, augmentation pitfalls, preprocessing)
4-Comprehensive baselines. Eleven detectors—including 2025 releases—are re-trained and evaluated under uniform protocols, exposing nuanced failure modes
5-Presentation quality. Writing is crisp, tables/figures are dense yet readable; experimental methodology is explicit.

---

> ### Author Rebuttal · Authors · 2025-07-26
>
> **Q1: Narrow Training Scope.**
>
> We appreciate the reviewer’s thoughtful comment regarding the training scope of our evaluation. We would like to clarify our rationale and provide additional experimental results to strengthen our argument.
>
> (1) Focus on Generalization:
>
> The core challenge in AIGI detection lies in its generalization capability—that is, the ability to detect synthetic images without prior knowledge of the specific generation method. This problem is fundamentally different from within-distribution detection and is increasingly critical as generative models evolve rapidly.
>
> In this context, our manuscript adopts the widely accepted evaluation setup in the community, using ProGAN and Stable Diffusion v1.4 as representative training sources. These two models were deliberately chosen to represent distinct families of generation paradigms—GANs and diffusion models—thereby offering a strong baseline for evaluating the generalization capacity of detectors. This setting aligns with established practices in related work and serves to emulate a realistic open-world detection scenario.
>
> (2) Additional Experiment – Training on All Subsets:
>
> To further address the reviewer’s concern and evaluate detector performance when exposed to a broader range of generation methods, we have added an experiment where all available subsets (covering multiple generators) are used for training and testing. This setting examines how detectors perform when trained with diverse synthetic distributions. Table I presents the performance of various AIGI detectors under two training settings: (1) training on all 25 subsets (main score), and (2) training only on SD-v1.4 and ProGAN (difference shown as subscript). The main values represent the mean performance across all test sets when trained on the full dataset. The subscripts indicate the performance gain or drop compared to the SD-v1.4 + ProGAN training setting, highlighting the impact of broader training coverage on generalization.
>
> Overall, most detectors benefit significantly from exposure to a more diverse set of generators during training. Notably, CNNDetection shows the largest improvement in fake accuracy (+48.8%) and average precision (+35.6%), indicating its limited generalization ability under the narrow training setup. In contrast, Ladeda and DFFreq exhibit more stable performance, with relatively smaller gaps, suggesting better inherent generalization.
>
> Interestingly, the state-of-the-art method SAFE yields the weakest performance when trained on the full dataset. Its overall accuracy even declines slightly (−1.3%) compared to the narrow training case, with more pronounced drops in robustness accuracy. This counterintuitive result points to a potential trade-off between generalization and fitting capacity.
>
> These findings suggest that training on all available datasets may not always be optimal for generalization assessment. In fact, such settings risk underestimating detectors with broader adaptability while overestimating those that rely on specific or narrow generative priors. Therefore, our results support the use of strategically curated training subsets that strike a balance between distributional coverage and discriminative clarity, rather than indiscriminate aggregation.
>
> We hope this analysis clarifies the motivation behind our training design and reinforces our commitment to comprehensive and meaningful evaluation. The revised manuscript will include this experiment and the accompanying discussion.
>
> **Table I. The performance for different AIGI detectors training on the subset of the all 25 datasets (main score) and training on both SD-v1.4 and ProGAN (subscript indicates the difference).**
> | Detector      |   Acc. Mean    |   A.P. Mean    |   R.Acc Mean   |   F.Acc Mean   |
> | :------------ | :------------: | :------------: | :------------: | :------------: |
> | CNNDetection  | $78.5_{+23.6}$ | $90.2_{+35.6}$ | $96.7_{-1.5}$  | $60.4_{+48.8}$ |
> | CLIPDetection | $93.3_{+20.8}$ | $98.4_{+22.8}$ | $94.7_{+21.2}$ | $91.9_{+20.4}$ |
> | DFFreq        | $89.9_{+1.88}$ | $95.5_{+19.8}$ | $89.8_{+0.2}$  | $89.9_{+38.0}$ |
> | Ladeda        | $92.6_{+19.2}$ | $98.8_{+19.5}$ | $98.4_{+0.9}$  | $92.6_{+37.7}$ |
> | AIDE          | $89.6_{+12.0}$ | $95.4_{+12.7}$ | $89.4_{+1.3}$  | $89.8_{+22.8}$ |
> | SAFE          | $78.6_{-1.3}$  | $86.7_{+4.1}$  | $92.7_{-4.1}$  | $64.4_{+1.4}$  |
>
> **Q2: Purely Diagnostic.**
>
> We thank the reviewer for this insightful comment. Our primary objective in this work is to systematically identify and analyze the critical limitations of existing AIGI detectors under realistic, real-world conditions. As acknowledged, the focus of our study lies in benchmark and dataset design, with the goal of establishing AIGIBench as a rigorous and comprehensive evaluation framework to support future method development and comparison.
> We argue that this diagnostic perspective is a necessary prerequisite for designing effective remedies. Without a clear understanding of where and why current detectors fail—whether in generalization, robustness, augmentation sensitivity, or test-time preprocessing—it is difficult to propose broadly effective and targeted improvements.
>
> While our study does not introduce new model architectures or training strategies, it implicitly offers actionable insights that can guide future advancements. For example:
> (a) Our analysis of data augmentation and test-time preprocessing strategies reveals their limited or inconsistent benefits, suggesting the need for augmentation-aware training paradigms or preprocessing-invariant architectures.
> (b) The comparative evaluation of frequency-based and CLIP-based detectors highlights specific architectural traits that contribute to improved robustness, offering valuable directions for detector design.
>
> Looking ahead, we plan to build upon these findings by exploring and evaluating remedial strategies in future work. Additionally, we aim to extend AIGIBench with a public leaderboard, fostering continual progress in the development of reliable and generalizable AIGI detection models.
>
> **Q3: Limited Metric Suite Evaluation relies on R.Acc., F.Acc., and A.P.**
>
> We thank the reviewer for this valuable suggestion. In addition to AP, Acc, F-Acc, and R-Acc, we now report AUC, EER, and TPRs at 10% and 1% FPR in Table II below.
> Due to space limitations, we provide the mean performance across the entire dataset here. The complete results, including per-subset breakdowns, will be included in the revised manuscript.
>
> Among the compared methods, SAFE achieves the highest TPRs at both 10% FPR (69.6%) and 1% FPR (56.3%), indicating strong robustness in scenarios requiring low false-positive rates. AIDE also performs competitively with a high AUC of 84.0 and a TPR of 65.2% at 10% FPR. In contrast, traditional methods like CNNDetection show significantly lower performance, especially at 1% FPR, where the TPR drops to just 9.5%.
>
> **Table II. The generalization results (AUC, EER, TPR at 10% FPR, and TPR at 1% FPR) for different AIGI detectors, where the
> training dataset settings is Setting-II: Training on 144K images generated by both SD-v1.4 and ProGAN.**
> |    Detector     | AUC Mean | EER Mean | T-10 Mean | T-1 Mean |
> | :----------- | :--: | :--: | :--: | :--: |
> | CNNDetection  | 68.7 | 35.5 | 31.2 | 9.5  |
> | CLIPDetection | 77.5 | 27.6 | 48.2 | 15.9 |
> |    DFFreq     | 78.0 | 26.9 | 51.4 | 24.2 |
> |    Ladeda     | 76.9 | 26.2 | 57.3 | 32.6 |
> |     AIDE      | 84.0 | 19.8 | 65.2 | 32.4 |
> |     SAFE      | 82.9 | 18.8 | 69.6 | 56.3 |
>
> **Q4: Sparse Ethical & Bias Analysis Board-impact discussion is brief.**
>
> We appreciate the reviewer’s concern regarding ethical considerations and bias analysis. We fully acknowledge that a more comprehensive treatment of demographic bias and risk mitigation is crucial—particularly for benchmarks involving facial data and synthetic media. While our current submission includes only a brief discussion due to space constraints, we have taken initial steps toward ethical dataset construction and diversity-aware design.
>
> Specifically, real face images in our dataset are sourced from publicly available datasets such as FFHQ and CelebA-HQ, both of which are curated to include a wide range of demographic attributes (e.g., age, gender, and ethnicity). For synthetic content, we collect data from a wide variety of generative models and platforms (e.g., X, Reddit, ArtStation, Civitai), aiming to maximize diversity in content style, generation method, and application domain. **For text-to-image generation**, we  leverage the Gemini API with carefully designed prompts to synthesize diverse and high-quality image descriptions. For instance, a prompt for generating realistic human descriptions might read: “To create varied descriptions of people in photo or realistic style, I need 1000 distinct sentences, each 20–25 words, please help me. Example: ‘woman wearing a red dress in the park, Disney cartoon style.’”. We generate 1500 sentences for each of four categories—people, animals, objects, and landscapes—to ensure broad coverage and content diversity. **For image-to-image generation**, we random select reference images from public datasets such as FFHQ and CelebA-HQ, which are curated to include diverse demographic attributes (e.g., age, gender, ethnicity).
>
> That said, we agree that an explicit demographic audit and bias evaluation would significantly strengthen the transparency and fairness of our benchmark. In this response, we conduct a demographic attribute prediction for all facial images using an off-the-shelf API to estimate age, gender, and other key attributes. In the revised version, we will publicly release the annotations and provide an initial analysis of demographic distributions across real and synthetic images. This will offer insights into potential dataset imbalance and serve as a foundation for subgroup performance evaluation in future work.

---

> > ### Comment · Reviewer_nDdE · 2025-08-04
> >
> > The authors have provided a complete response to my concerns. I highly recommend the authors include the revisions in the final version.

---

> > > ### Author Response · Authors · 2025-08-05
> > > **Thanks for your comments**
> > >
> > > We sincerely thanks for your thoughtful feedback and positive recommendation. We are pleased to hear that our revisions addressed your concerns. We will incorporate the suggested changes into the final version as recommended.

---

### Official Review · Reviewer_4Tx5 · 2025-07-02

**Rating:** 5
**Confidence:** 4

**Summary:**

This paper presents a new benchmark for detecting AI-generated images. It introduces four key aspects that aim to make the benchmark more realistic and challenging: (1) collecting synthetic images from over 20 different generative models, (2) evaluating detection performance under multiple levels of image degradation, (3) analyzing the effects of different training-time data augmentations, and (4) assessing the impact of common test-time processing steps such as cropping and resizing. The authors conduct extensive experiments and find that most existing detectors perform poorly under these realistic conditions, highlighting the need for more robust detection methods.

**Additional Feedback:**

The paper makes a valuable contribution by proposing a challenging and diverse benchmark for AI-generated image detection. However, I have several concerns that should be addressed.

First, there seems to be considerable overlap with latest works. For example, [1] already introduces a dataset with over 20 generative models, and [2] extends this with more realistic generation settings. Both works also include in-the-wild test sets and robustness evaluations. The authors may need to clarify how their benchmark differs from or improves upon these existing efforts.

Second, the evaluation setup includes some questionable design choices. The JPEG robustness test uses a quality factor of 50, which is extremely low and not representative of typical real-world usage. Most users would reject images of such low quality, making this setting less meaningful. Additionally, robustness may need to be evaluated on samples where detectors perform well under clean conditions. Testing on already difficult examples makes it hard to isolate the impact of added perturbations.

Third, the paper does not report TPR at low FPR, a metric that is now standard in the literature [1][2]. This metric is especially important for deployment scenarios where false positives must be minimized. Including it would significantly strengthen the evaluation.

---
References:

[1] Cheng, Siyuan, et al. "CO-SPY: Combining Semantic and Pixel Features to Detect Synthetic Images by AI." CVPR 2025.

[2] Chen, Baoying, et al. "DRCT: Diffusion Reconstruction Contrastive Training Towards Universal Detection of Diffusion-Generated Images." ICML 2024.

**Dataset Code Accessibility:**

Yes

**Dataset Code Comments:**

The dataset code is well-structured.

**Ethical Considerations:**

No, there are no or only very minor ethics concerns

**Final Justification:**

I appreciate the authors' efforts to address my comments and concerns.
I recognize the strengths of the proposed benchmark, particularly its scale and comprehensiveness compared to existing ones. Accordingly, I have raised my score, and I highly recommend that the authors include these justifications over recent works, as well as the additional robustness evaluation in the final version.

**Limitations Weaknesses:**

The benchmark appears to overlap with some recent work in terms of both dataset composition and evaluation strategy. There are also concerns about the evaluation setup, including the choice of unrealistic parameters in robustness testing. In addition, the evaluation is missing an important metric, i.e., TPR at low FPR, which is widely adopted in recent studies and critical for assessing real-world performance.

**Strengths Contributions:**

The paper is clearly written and easy to follow. The proposed benchmark is comprehensive in terms of both data diversity and evaluation scenarios. The experiments are thorough and effectively support the main claim that detecting AI-generated images remains an open and challenging problem.

---

> ### Author Rebuttal · Authors · 2025-07-26
>
> **Q1: There seems to be considerable overlap with latest works.**
>
> We thank the reviewer for pointing out the relevance of prior benchmarks and appreciate the opportunity to clarify the distinct contributions of our work. While previous efforts such as CO-SPYBENCH [1] and DRCT-2M [2] have indeed introduced large-scale datasets and emphasized in-the-wild scenarios, our proposed benchmark AIGIBench advances the field in the following key aspects:
>
> **(1) Broader and More Recent Coverage of Generative Models.** Although CO-SPYBENCH [1] and DRCT-2M [2] contains over 20 generative models, their datasets are exclusively composed of images generated by diffusion models. More importantly, the majority of these models are variants of Stable Diffusion, resulting in relatively limited diversity in generation techniques and content. In contrast, AIGIBench encompasses 23 diverse fake image subsets, integrating both classic GANs, modern diffusion models, DeepFake techniques, and personalized generation methods (e.g., InstantID, PhotoMaker, IP-Adapter). Over half of these models are from 2024 or later, reflecting the latest advancements. Furthermore, CO-SPYBENCH [1] and DRCT-2M [2] generate images using the corresponding real image captions, while our proposed  AIGIBench leverage the Gemini API with carefully designed prompts to synthesize diverse and high-quality image descriptions. For instance, a prompt for generating realistic human descriptions might read: “To create varied descriptions of people in photo or realistic style, I need 1000 distinct sentences, each 20–25 words, please help me. Example: ‘woman wearing a red dress in the park, Disney cartoon style.’”. We generate 1500 sentences for each of four categories—people, animals, objects, and landscapes—to ensure broad coverage and content diversity.
>
> **(2) Inclusion of Real-World Content from Social Media and AI Art Communities.**
> Although CO-SPYBENCH [1] and DRCT-2M [2] include in-the-wild test sets, both benchmarks primarily collect images from CIVITAI. In contrast, AIGIBench extends the scope of data sources by additionally incorporating real-world fake images from a broader range of platforms, including ArtStation, Liblib, and notably, social media networks such as X (formerly Twitter) and Reddit. This wider coverage contributes to greater diversity and realism, better reflecting the complexities of AI-generated content encountered in practical scenarios.
>
> **(3) Comprehensive Evaluation Framework Covering Four Critical Tasks.**
> While prior benchmarks such as CO-SPYBENCH [1] and DRCT-2M [2] primarily focus on two evaluation aspects—namely, generalization to unseen generators and robustness to image degradations—AIGIBench is the first to propose a comprehensive evaluation framework encompassing four critical tasks that reflect practical challenges in real-world AIGI detection. These include:
> a) Generalization to unseen generative models,
> b) Robustness to multiple image degradations (e.g., compression, noise, and resampling),
> c) Sensitivity to different data augmentation strategies, and
> d) Impact of test-time pre-processing methods (e.g., crop vs. resize).
> This holistic design enables a more realistic and thorough assessment of detector performance, addressing limitations often overlooked in earlier benchmarks.
>
> **(4) Consistent Re-Evaluation of 11 Recent Detectors Under Uniform Protocols.**
> Our benchmark evaluates 11 state-of-the-art AIGI detectors, including SAFE, AIDE, CLIPDetection, and LaDeDa, many of which were proposed in 2024 or 2025. All detectors are retrained and tested under standardized protocols, ensuring a fair and transparent comparison. In contrast, prior benchmarks exhibit more limited and outdated detector coverage. For example, DRCT-2M includes 8 detectors, all of which were proposed before 2024; similarly, CO-SPYBENCH [1] evaluates 10 detectors, 7 of which were proposed before 2024.
> By incorporating more recent methods and unified experimental settings, AIGIBench offers a more timely and rigorous evaluation of current detection capabilities.
>
> **Q2: The evaluation setup includes some questionable design choices.**
>
> We appreciate the reviewer’s insightful feedback regarding the evaluation setup. We acknowledge that using a JPEG quality factor of 50 represents a strong degradation that may not align with typical real-world usage scenarios. Our original motivation was to stress-test detector robustness under extreme compression artifacts. However, to better reflect realistic conditions, we have expanded our evaluation in the revised manuscript to include more moderate settings, specifically: (1) JPEG compression with a quality factor of 90, (2) Gaussian noise with a moderate intensity (σ = 1), and (3) A composite setting combining JPEG-95 and σ = 1 noise.
>
> These additions allow for a more practical assessment of robustness under conditions likely to occur in real-world image sharing or editing pipelines. The updated results are summarized in Table I below, where all detectors are trained under Setting-II and evaluated across 25 diverse test datasets.The metrics reported include Accuracy (Acc.), Average Precision (A.P.), Real Accuracy (R.Acc.), and Fake Accuracy (F.Acc.). Key observations include:
> (1) CLIPDetection consistently outperforms other detectors across all perturbation settings, demonstrating strong robustness in both clean and noisy conditions.
> (2)CNNDetection, DFFreq, Ladeda, AIDE, and SAFE suffer significant performance degradation, particularly in terms of Fake Accuracy, indicating a high false negative rate under perturbations.
> (3) The composite perturbation setting (JPEG-95 + noise) further amplifies these challenges, exposing the vulnerability of frequency- and CNN-based detectors.
> (4) Interestingly, AIDE and SAFE maintain near-perfect Real Accuracy, but this comes at the cost of very poor Fake Accuracy, revealing a strong bias toward classifying images as real under noise.
>
> These results reinforce the importance of evaluating detectors under more moderate yet realistic perturbation levels and highlight the varying robustness characteristics of different detection paradigms.
>
> **Table I. The overall robust performance of AI-generated image detectors, where the training dataset follows Setting-II: training on 144K images generated by both SD-v1.4 and ProGAN. Notably, all reported results represent average values computed across 25 diverse test datasets.**
> | Detector      |       JPEG-90 （Acc./A.P./R.Acc./F.Acc.）      |  Noise -$\sigma=1$   | Noise -$\sigma=1$ & JPEG-95 |
> | :------------ | :-----------------: | :------------------: | :-----------------------: |
> | CNNDetection  | 54.6/66.3/97.9/11.4 | 53.9/57.4/96.1/11.7  |    53.8/57.7/96.4/11.1    |
> | CLIPDetection | 69.7/73.2/74.5/64.8 | 71.9/74.4/75.3/68.4  |    72.0/75.6/76.3/67.6    |
> | DFFreq        | 51.1/54.9/99.5/1.8  | 64.9/73.3/80.8/48.6  |    52.3/63.5/99.1/4.7     |
> | Ladeda        | 50.1/57.0/99.9/0.3  | 64.2/74.2/83.8/44.6  |    50.5/62.8/99.6/1.2     |
> | AIDE          | 50.3/57.9/99.6/1.0  | 77.9/73.7/65.3/90.5  |    50.0/53.0/99.7/0.3     |
> | SAFE          | 50.0/50.8/100.0/0.0 | 71.34/82.1/89.0/53.7 |    50.0/49.9/100.0/0.0    |
>
>
>
> **Q3: The paper does not report TPR at low FPR.**
>
> We thank the reviewer for pointing out the importance of including TPR at low FPR, a metric especially relevant for real-world deployment scenarios where minimizing false positives is critical. In addition to AP, Acc, F-Acc, and R-Acc, we now report ROC-AUC, EER, and TPRs at 10% and 1% FPR in Table II below.
> Due to space limitations, we provide the mean performance across the entire dataset here. The complete results, including per-subset breakdowns, will be included in the revised manuscript.
>
> Among the compared methods, SAFE achieves the highest TPRs at both 10% FPR (69.6%) and 1% FPR (56.3%), indicating strong robustness in scenarios requiring low false-positive rates. AIDE also performs competitively with a high AUC of 84.0 and a TPR of 65.2% at 10% FPR. In contrast, traditional methods like CNNDetection show significantly lower performance, especially at 1% FPR, where the TPR drops to just 9.5%.
>
> **Table II. The generalization results (AUC, EER, TPR at 10% FPR, and TPR at 1% FPR) for different AIGI detectors, where the
> training dataset settings is Setting-II: Training on 144K images generated by both SD-v1.4 and ProGAN.**
> |    Detector     | AUC Mean | EER Mean | T-10 Mean | T-1 Mean |
> | :----------- | :--: | :--: | :--: | :--: |
> | CNNDetection  | 68.7 | 35.5 | 31.2 | 9.5  |
> | CLIPDetection | 77.5 | 27.6 | 48.2 | 15.9 |
> |    DFFreq     | 78.0 | 26.9 | 51.4 | 24.2 |
> |    Ladeda     | 76.9 | 26.2 | 57.3 | 32.6 |
> |     AIDE      | 84.0 | 19.8 | 65.2 | 32.4 |
> |     SAFE      | 82.9 | 18.8 | 69.6 | 56.3 |

---

### Official Review · Reviewer_wEqj · 2025-07-02

**Rating:** 5
**Confidence:** 2

**Summary:**

This paper introduces AIGIBench, a comprehensive benchmark for evaluating the robustness and generalization of AI-generated image (AIGI) detectors. The authors curate a diverse dataset of 23 subsets, including images from state-of-the-art GAN and diffusion models, as well as real-world examples from social media. Through four evaluation tasks—multi-source generalization, robustness to degradation, sensitivity to data augmentation, and impact of test-time pre-processing—the paper demonstrates that current state-of-the-art detectors exhibit significant performance drops in real-world scenarios, challenging the notion that AIGI detection is a solved problem.

**Dataset Code Accessibility:**

Yes

**Ethical Considerations:**

No, there are no or only very minor ethics concerns

**Limitations Weaknesses:**

* Lack of Statistical Significance Reporting: The paper's main claims rest on the performance differences reported in Tables 3-6. However, these tables only report mean performance scores without any measure of variance (e.g., standard deviation across multiple runs) or statistical significance tests. For a benchmark paper, demonstrating the stability of these results is crucial. Could the authors report error bars or conduct significance testing to confirm that the observed performance differences, especially between top-performing models like SAFE and AIDE, are statistically significant?

* Insufficient Detail on Curation Process: In Section 2.2, the paper mentions using CLIP to remove similar instances and an aesthetic score predictor to filter images. This is a critical step that could introduce bias. The paper would be strengthened by providing more specific details: What similarity threshold was used for CLIP? What was the cutoff for the aesthetic score? A discussion of how these choices might affect the final dataset distribution and downstream evaluation is needed for full transparency.

**Strengths Contributions:**

* High-Quality and Comprehensive Asset: The paper introduces a significant and well-curated benchmark, AIGIBench. Its value lies in its breadth (25 generative methods, including in-the-wild data from social media) and its systematic evaluation structure. This is a timely contribution that will be of immediate use to the community.

* Thorough and Insightful Analysis: The experimental design is rigorous. The paper's core strength is its multi-faceted analysis across the four defined tasks. For example, the breakdown of accuracy into R.Acc and F.Acc (Table 3) provides a much more nuanced view than a single accuracy score, revealing that many detectors achieve high R.Acc at the cost of near-zero F.Acc on challenging subsets like SocialRF. Similarly, the analysis in Table 4, showing that F.Acc drops to nearly 0% for most detectors under JPEG compression, is a stark and valuable finding.

* Clear Presentation and Reproducibility: The paper is well-written and the results are clearly presented. The authors have also made the data and code publicly available, which is a crucial requirement for the Datasets and Benchmarks track and greatly enhances the work's impact and utility.

---

> ### Author Rebuttal · Authors · 2025-07-26
>
> **Q1: Lack of Statistical Significance Reporting**
>
> We sincerely thank the reviewer for highlighting the importance of reporting statistical significance in benchmarking studies. We fully agree that demonstrating the stability and reliability of performance differences is essential, particularly when comparing top-performing models.
>
> To address this concern, we report both the mean and standard deviation of performance metrics in Tables I–IV. These results highlight not only the average performance but also the stability of each detector. Notably, methods such as SAFE and AIDE exhibit both high mean accuracy and relatively low variance, indicating consistent performance, whereas other detectors display larger fluctuations that may compromise their reliability in practical scenarios. In addition, we conducted paired t-tests to assess the statistical significance of key model comparisons—for instance, SAFE vs. AIDE. As shown in Table 3 of the manuscript, the paired t-test yields a t-statistic of 2.53 with a corresponding p-value of 0.0147, indicating a statistically significant difference between AIDE and SAFE (p < 0.05) and rejecting the null hypothesis of equal performance. In contrast, Table 4 of the manuscript presents a t-statistic of –0.0948 and a p-value of 0.926, suggesting no significant difference in robustness between the two detectors. We believe these enhancements substantially strengthen the benchmark’s credibility.
>
> **Table I. Mean and standard deviation of Table 3 on the manuscript.**
> | Detector       |Acc Mean|Acc Std|AP Mean|AP Std|R.Acc  Mean|R.Acc Std|F.Acc Mean|F.Acc Std|
> |----------------|--------------|-------------|-------------|------------|----------------|---------------|----------------|---------------|
> | Resnet-50      | 61.9         | 15.5        | 69.3        | 20.9       | 95.7           | 2.12          | 27.9           | 31.09         |
> | CNNDetection   | 54.9         | 14.2        | 67.0        | 16.8       | 98.2           | 0.97          | 11.6           | 18.15         |
> | Gram-net       | 58.6         | 16.3        | 62.4        | 19.5       | 90.5           | 4.35          | 26.6           | 29.45         |
> | LGrad          | 62.9         | 16.3        | 66.6        | 19.5       | 85.8           | 4.06          | 39.6           | 28.12         |
> | CLIPDetection  | 72.5         | 14.8        | 75.6        | 18.2       | 73.3           | 8.27          | 71.5           | 26.17         |
> | FreqNet        | 66.2         | 18.6        | 70.1        | 22.4       | 65.9           | 10.26         | 66.4           | 30.75         |
> | NPR            | 67.9         | 17.2        | 73.9        | 21.0       | 93.8           | 2.35          | 41.9           | 33.33         |
> | DFFreq         | 71.1         | 16.9        | 75.7        | 20.5       | 89.6           | 3.39          | 51.9           | 31.72         |
> | LaDeDa         | 73.4         | 15.0        | 79.3        | 19.8       | 91.7           | 3.92          | 54.9           | 36.79         |
> | AIDE           | 77.6         | 12.4        | 82.7        | 15.6       | 88.1           | 4.74          | 67.0           | 36.81         |
> | SAFE           | 79.9         | 20.1        | 82.6        | 24.3       | 96.8           | 1.41          | 63.0           | 41.21         |
>
> **Table II. Mean and standard deviation of Table 4 on the manuscript.**
> | Detector        | R.Acc. Mean | R.Acc. Std | F.Acc. Mean | F.Acc. Std | A.P. Mean | A.P. Std |
> |-----------------|-------------|------------|-------------|------------|-----------|----------|
> | Resnet-50       | 97.7        | 2.04       | 14.7        | 14.57      | 66.8      | 4.96     |
> | CNNDetection    | 97.5        | 2.31       | 8.3         | 7.41       | 58.6      | 8.85     |
> | Gram-net        | 94.2        | 4.22       | 15.9        | 12.16      | 60.7      | 3.52     |
> | LGrad           | 90.0        | 4.77       | 30.4        | 22.38      | 65.4      | 11.09    |
> | CLIPDetection   | 79.9        | 7.75       | 57.4        | 17.13      | 73.6      | 1.99     |
> | FreqNet         | 78.5        | 14.57      | 44.9        | 29.99      | 65.6      | 8.89     |
> | NPR      | 96.8        | 2.95       | 20.7        | 20.53      | 70.7      | 9.19     |
> | DFFreq   | 91.9        | 5.84       | 31.5        | 22.44      | 69.7      | 7.89     |
> | LaDeDa   | 95.7        | 4.33       | 26.0        | 28.77      | 73.5      | 10.35    |
> | AIDE     | 88.7        | 10.27      | 29.6        | 27.36      | 65.2      | 15.09    |
> | SAFE     | 99.2        | 1.60       | 20.1        | 29.54      | 62.9      | 17.87    |
>
> **Table III. Mean and standard deviation of Table 5 on the manuscript.**
> | Detector       |Acc Mean|Acc Std|AP Mean|AP Std|R.Acc  Mean|R.Acc Std|F.Acc Mean|F.Acc Std|
> |----------------|--------------|-------------|-------------|------------|----------------|---------------|----------------|---------------|
> |CLIPDetection|70.78 | 1.38   |75.47 | 1.04   | 78.58 | 4.96   |62.95 | 5.35   |
> |FreqNet|65.20 | 2.17   |69.08 | 2.44   | 74.28 | 9.60   |55.97 | 10.55  |
> |NPR| 66.63 | 1.40   |73.03 | 1.97   | 94.77 | 2.07   | 38.45 | 4.07   |
> |DFFreq|69.03 | 3.06   |73.90 | 2.89   |89.00 | 2.35   |48.25 | 4.91   |
> |SAFE|78.05 | 2.09   |83.32 | 1.45   |93.85 | 5.84   |62.18 | 6.57   |
>
> **Table IV. Mean and standard deviation of Table 6 on the manuscript.**
> | Detector       |Acc Mean|Acc Std|AP Mean|AP Std|R.Acc  Mean|R.Acc Std|F.Acc Mean|F.Acc Std|
> |----------------|--------------|-------------|-------------|------------|----------------|---------------|----------------|---------------|
> |CLIPDetection|69.50 | 3.00    |72.00 | 3.60    |75.10 | 1.80    |63.80 | 7.70    |
> |FreqNet| 70.20 | 4.00    | 75.05 | 4.95    |75.25 | 9.35    |64.95 | 1.45    |
> |NPR| 68.05 | 0.15    |77.90 | 4.00    |96.55 | 2.75    |39.40 | 2.50    |
> |DFFreq|72.75 | 1.65    |78.40 | 2.70    | 92.85 | 3.25    | 51.80 | 0.10    |
> |SAFE|72.40 | 7.50    |75.60 | 7.00    |80.05 | 16.75   | 64.75 | 1.75    |
>
> **Q2: Insufficient Detail on Curation Process**
>
> We thank the reviewer for highlighting the importance of transparency in our data curation process. In our revision, we will add detailed descriptions of the filtering steps. Specifically, we used CLIP image embeddings with a cosine similarity threshold of 0.98 to identify and remove near-duplicate instances. For aesthetic filtering, we employed an off-the-shelf aesthetic quality predictor and retained only 4K images with the most aesthetic  scores.
>
> The primary motivation behind this filtering process is to increase the difficulty and discriminative power of the detection benchmark. Low-quality or visually homogeneous images often reduce the evaluation challenge and may artificially inflate the performance of detectors. By removing redundant and low-aesthetic images, we ensure the dataset better reflects real-world, high-fidelity scenarios and allows for a more rigorous and meaningful evaluation of detection capabilities.
>
> To address concerns about potential bias, we have also added a discussion in the revised manuscript analyzing how these filtering steps impact dataset diversity. To further support this analysis, Table V presents the F.Acc (Fake Accuracy) results of six representative detectors on both the curated dataset and the original unfiltered dataset. The main value in each cell corresponds to the detector's performance on the original unfiltered dataset. The subscript indicates the absolute performance difference between the unfiltered and curated  datasets. A positive subscript (e.g., +2.9) indicates that the detector performed worse on the curated dataset, implying that the curated dataset is more challenging. Conversely, a negative subscript (e.g., −1.0) suggests improved performance. Across most models and subsets, we observe a consistent decline in F.Acc following curation, particularly for **state-of-the-art detectors such as AIDE and SAFE**. These results underscore the increased challenge presented by the curated dataset and validate our filtering approach as a means of constructing a more robust benchmark.
>
> Finally, to promote transparency and reproducibility, we will release both the curated dataset and the original unfiltered data. This will enable the community to assess the effects of our filtering decisions and apply alternative selection strategies as needed.
>
> **Table V. F.Acc results on the original unfiltered dataset (main score) and the curated dataset (subscript indicates the difference). A positive subscript (e.g., +2.9) indicates that the detector performed worse on the curated dataset, implying that the curated dataset is more challenging.**
> |   Detector    |    FaceSwap    |  IP-Adapter   |  Midjourney   |        SD3        |   StyleSwim   |
> | :----------- | :------------: | :-----------: | :-----------: | :---------------: | :-----------: |
> | CNNDetection  |  $1.8_{+0.4}$  | $5.0_{-1.0}$  | $6.1_{+0.3}$  |   $16.2_{+2.9}$   | $8.7_{+1.8}$  |
> | CLIPDetection | $29.8_{+2.5}$  | $94.3_{+2.3}$ | $78.5_{-2.1}$ |   $92.1_{+1.5}$   | $97.9_{-0.2}$ |
> |    DFFreq     |  $0.9_{+0.6}$  | $76.8_{-1.3}$ | $55.2_{+1.2}$ |   $75.7_{+2.3}$   | $87.9_{+7.1}$ |
> |    Ladeda     |  $0.7_{+0.7}$  | $92.6_{+2.0}$ | $84.6_{+1.2}$ |   $98.5_{-0.5}$   | $96.3_{-1.0}$ |
> |     AIDE      | $12.0_{-2.3}$  | $94.1_{+0.6}$ | $80.1_{+0.3}$ | $99.3_{+0.0}$ | $91.4_{+9.4}$ |
> |     SAFE      | $22.9_{+19.6}$ | $99.3_{+9.5}$ | $98.5_{+1.3}$ |   $86.3_{-5.4}$   | $99.9_{+0.6}$ |

---

> ### Comment · Area_Chair_QmuH · 2025-08-09
>
> Dear Reviewer wEqj,
>
> The reviewer-author discussion will end in 24 hours. Please respond to authors' rebuttal and participate in the discussion before it closes.
>
> Best,
>
> AC

---

### Comment · Area_Chair_QmuH · 2025-08-06
**Please engage in the reviewer-author discussion.**

Dear Reviewers,

Thank you for providing the initial reviews. Please respond to the authors' rebuttal and engage in the reviewer-author discussion if you haven't done so.

Best,

AC

---

### Note · Authors · 2025-08-15

We sincerely thank all reviewers for their constructive feedback, which has significantly improved the quality and clarity of our work. In response to the main concerns:

**1. Statistical Significance & Metric Expansion** – We now report mean and standard deviation for all performance metrics and conduct paired t-tests to assess significance between key detectors. Additional evaluation metrics, including ROC-AUC, EER, and TPR at 10%/1% FPR, have been incorporated to better reflect deployment-oriented performance.

**2. Data Curation Transparency** – We have detailed our duplicate removal (CLIP embeddings, cosine similarity 0.98) and aesthetic filtering (off-the-shelf predictor, high-resolution retention) steps. We analyze the impact on dataset diversity and show that curation increases benchmark difficulty. Both curated and unfiltered datasets will be released for reproducibility.

**3. Benchmark Novelty & Scope** – We clarify that AIGIBench advances prior benchmarks by (a) integrating 23 diverse generators (GAN, diffusion, DeepFake, personalization), over half from 2024+, (b) incorporating broader in-the-wild sources beyond CIVITAI, (c) covering four realistic evaluation tasks beyond generalization and degradation, and (d) re-evaluating 11 recent detectors under unified protocols.

**4. Evaluation Design Choices** – In addition to extreme JPEG-50 tests, we include more realistic degradations (JPEG-90, moderate noise, compound perturbations), revealing distinct robustness characteristics across detection paradigms. We further report results from training on all 25 subsets, showing that broader training does not always improve generalization and may even harm certain detectors.

**5. Ethical & Bias Considerations** – We describe dataset sourcing to ensure demographic diversity and, in the revision, will release predicted demographic attributes for all facial images to support subgroup analysis and bias evaluation.

Overall, these revisions enhance the statistical rigor, transparency, and comprehensiveness of AIGIBench, reinforcing its value as a robust and reproducible benchmark for AI-generated image detection research. We believe the improved manuscript addresses all major concerns and offers meaningful contributions to the community.

---

### Decision · Program_Chairs · 2025-09-18

**Decision:**

Accept (poster)

**Comment:**

This paper introduces AIGIBench, a comprehensive and well-curated benchmark for evaluating AI-generated image detectors. Reviewers unanimously agree that the work is timely, rigorous, and impactful, with strong methodological design, broad coverage of generative models, and a realistic evaluation setup. The analysis yields important insights into detector weaknesses, such as robustness failures under compression and distribution shifts, and provides valuable guidance for future research. The presentation is clear, reproducibility is ensured, and the contribution is well-suited to the Datasets and Benchmarks track. Most of the reviewers' concerns have been addressed during the discussion. This work stands out for its comprehensive evaluation and timely contribution to the AIGI community. Overall, I recommend it for Spotlight paper.

===== FINAL UPDATE FROM DB Track PCs ====

The final decision for this paper has been taken by the program chairs after consultation with the SACs. All Senior Area Chairs have ranked papers according to the feedback from the AC during the review process. We decided to leave the original meta-review to reflect the opinion of the AC in light of the initial discussions with reviewers and SAC.